# Environmental drivers of soil phosphorus composition in natural ecosystems

Leonardo Deiss[1], Anibal de Moraes[1], Vincent Maire[2]

[1] Federal University of Parana, Curitiba, Rua dos Funcionários 1.540 CEP80035-050, Brazil.
[2] University of Quebec in Trois-Rivières, Trois-Rivières, Quebec, QC G9A 5H7, Canada.

*Correspondence to*: Leonardo Deiss (leonardodeiss@gmail.com)

**Abstract.** Soil organic and inorganic phosphorus (P) compounds can be influenced by distinctive environmental properties. This study aims to analyze soil P composition in natural ecosystems, relating organic (inositol hexakisphosphate, DNA and phosphonates) and inorganic (orthophosphate, polyphosphate and pyrophosphate) compounds with major temporal (weathering), edaphic and climatic characteristics. A dataset including 88 sites was assembled from published papers that determined soil P composition using one-dimensional liquid state $^{31}$P nuclear magnetic resonance spectroscopy of NaOH-EDTA extracts of soils. Bivariate and multivariate regression models were used to better understand the environmental properties influencing soil P. In bivariate relationships, trends for soil P compounds were similar for mineral and organic layers but with different slopes. Independent and combined effects of weathering, edaphic and climatic properties of ecosystems explained up to 78% (inositol hexakisphosphates) and 89% (orthophosphate) of variations in organic and inorganic P compounds across the ecosystems, likely deriving from parent material differences. Soil properties, particularly pH, total carbon and carbon-to-phosphorus ratios, over climate and weathering mainly explained the P variation. We conclude that edaphic and climatic drivers regulate key ecological processes that determine the soil P composition in natural ecosystems. These processes are related to the source of P inputs, primarily determined by the parent material and soil forming factors, plant and microbe P cycling, the bio-physico-chemical properties governing soil phosphatase activity, soil solid surface specific reactivity and P losses through leaching, and finally the P persistence induced by the increasing complexity of organic and inorganic P compounds as the pedogenesis evolves. Soil organic and inorganic P compounds respond differently to combinations of environmental drivers, which likely indicates that each P compound has specific factors governing its presence in natural ecosystems.

# 1 INTRODUCTION

Phosphorus (P) is a key nutrient in animal, microbial and plant nutrition and 'bears light' to terrestrial ecosystem functioning, regulating primary and secondary productivities (Walker and Adams 1958; Vitousek et al. 2010). Phosphorus input into a young ecosystem derives predominantly from the weathering of parent material, with some systems receiving P input from eolian deposits (Chadwick et al. 1999). Once P has been dissolved from primary minerals, plants and microorganisms access it from the soil solution. This P is then recycled through soil as organic and inorganic P ($P_i$ and $P_o$, respectively) compounds (Noack et al. 2012; Damon et al. 2014), which are similarly subjected to a new cycle of physico-chemical and biological reactions. Each iteration of this cycle alters the form and bioavailability of the P, leading to decreasing levels of bioavailable P compounds (McDowell et al. 2007). In the absence of 'fresh' P inputs, this results in severe P limitations to ecosystem productivity (Walker and Syers 1976). The five state factors of soil formation (time, parent material, climate, topography and biota) determine the rate at which the cycle is completed (Jenny 1941). Therefore, a better understanding of the role of the five state factors as drivers of soil P composition is crucial to quantifying the relative abundance and form of both $P_i$ and $P_o$ pools.

In soils, $P_i$ and $P_o$ pools are each composed of specific P compounds (species) (Newman and Tate 1980; Tate and Newman 1982). The main $P_o$ compound categories are: i) orthophosphate monoesters (single ester linkage to orthophosphate) such as inositol hexakisphosphates, ii) orthophosphate diesters (two ester linkages to orthophosphate) such as ribonucleic acid, deoxyribonucleic acid, lipoteichoic acid, phospholipid fatty acids, and iii) phosphonates. Inorganic P compounds include orthophosphate, polyphosphate and pyrophosphate. Specific phosphatase enzymes are required to transform the different $P_o$ and $P_i$ forms into orthophosphate, which is the P compound directly taken up by plants and microbes (Jackman and Black, 1952). As with most enzymes, the activities of soil phosphatases are very sensitive to the hydrogen potential (pH) with specific enzyme optima (Frankenberger and Johanson, 1982). However, other soil variables are also involved in regulating $P_i$ and $P_o$ transformations. For example, inositol hexakisphosphates bind strongly to metal oxides and other soil components, which strongly limits their bioavailability (Turner et al. 2007). Similarly, in pH lower than 5 (the isoelectric point of DNA), amino group protonation of adenine, guanine and cytosine bases in the DNA molecule can cause adsorption of positively charged DNA to the negatively charged clay surface (Cai et al. 2006). As a result, many soil properties regulate soil P composition but their relative importance across contexts is unclear.

The absolute and relative abundances of $P_o$ and $P_i$ forms and compounds are likely related to ecosystem development and soil weathering (McDowell et al. 2007; Turner et al. 2007), as conceptualized by the Walker and Syers model for P fractions (e.g., Walker and Syers 1976; Yang and Post 2011) (Figure 1, upper panel). As soils undergo pedogenesis, ecosystem productivity progresses from nitrogen (N) to P limitation with ecosystem productivity peaking at the N-P colimitation intermediate stage of pedogenesis (Walker and Syers 1976; Turner and Condron, 2013). Parallel changes occur in soil properties including a decrease in total exchangeable bases, soil acidification, and an increase in Al and Fe oxide concentration (Albrecht 1957; Walker 1965). As a result, some $P_o$ and $P_i$ compounds increasingly react with the

mineral surface and progressively become occluded P (Yang and Post 2011). Subsequently, the complexity in $P_o$ and $P_i$ composition increases during ecosystem development (McDowell et al. 2007, Figure 1, bottom panel). The degree of soil weathering is inherently linked to the state factor 'time'—as demonstrated along many chronosequences (e.g. Turner and Laliberté 2015)—but it can be altered through other state factors (Albrecht 1957), such as along climosequences (e.g. Feng et al. 2016) or toposequences (e.g. Agbenin and Tiessen 1995). Along a climosequence, precipitation increased both base cation leaching and the degree of soil weathering, whereas potential evapotranspiration decreased these processes (Feng et al. 2016). While the Feng et al. (2016) study evaluated the mineral-P associations described by the Hedley P fractions, rather than P speciation, it illustrates the opposing effects of various climatic factors on edaphic factors of interest.

The parent material has distinct effects on soil properties, all other state factors otherwise being equal. Some of these effects are direct effects, such as the total P concentration of the initial geologic material. However, other factors may be more indirect. Parent materials can differ in total exchangeable base concentration and mineral composition. Variations in mineral composition can lead to differences in soil pH, soil texture, and Al and Fe oxides, all of which influence soil P cycling and P composition. For instance, soil P retention potential is influenced by differential absorption of $P_i$ and $P_o$ to clays, soluble Ca content, as well as Al and Fe oxyhydroxides (Batjes 2011). As such, the parent material state factor is an essential consideration in describing soil P cycling. Most importantly, we need to investigate the hierarchical nature of causal effects between state factors, soil weathering, soil properties, and $P_o$ and $P_i$ composition.

Nuclear magnetic resonance spectroscopy (NMR) is a widely-used method to study $P_o$ and Pi compounds in ecosystems around the world (Kizewski et al. 2011). This technique can be used for both qualitative and quantitative estimates of P compounds in soil (Cade-Menun and Preston 1996). The most effective extractant for NMR analysis has been NaOH combined with the chelating agent EDTA (Cade-Menun and Liu 2014). This does not imply that NaOH-EDTA is the best extractant for $^{31}$P NMR; however, because of its widespread use, it is a good baseline for comparison (Cade-Menun and Liu 2014). According to Cade-Menun and Preston (1996), NaOH can solubilize $P_i$ and $P_o$ compounds while EDTA chelates metallic cations to increase P extraction efficiency from the soil. The NaOH-EDTA extraction method is widely recognized to quantitatively extract P compounds from the soil (Turner and Blackwell 2013, Cade-Menun and Liu 2014). However, there are drawbacks of using NaOH-EDTA extractant for $^{31}$P NMR analysis. NaOH-EDTA does not extract all soil P and the highly alkaline environment can potentially degrade some P compounds (Cade-Menun et al. 2006; Cade-Menun and Liu 2014). Additionally, the high pH of the NaOH-EDTA extraction separates P species from the cations (e.g. Al, Fe, Ca) with which they were associated in soil. There are other methods to study P dynamics (Frossard et al. 2011), and soil P composition (Kruse et al. 2015), but none of these methods is perfect individually. For example, while X-ray absorption near edge structure (XANES), is a more preferred method for looking at orthophosphate speciation (Hesterberg 2010), and is a solid-state technique that does not require extraction, P concentrations are often below the detection limit. Therefore, XANES can only detect broad P species groups (e.g. Fe-P, Ca-P), but cannot, for example, determine if DNA is sorbed to Fe or Al. The most thorough studies of soil P use a combination of techniques together, and not any single technique (e.g. Liu et al. 2013; Liu et al. 2015).

There is a lack of broader understanding of how soil P composition is affected by different state factors of soil formation. Using a large-scale comparative geographical approach, we aim to determine the causal paths through which climate, parent material and time influence soil properties, as well as their impact on $P_i$ and $P_o$ pools and specific P compounds. Combining the soil $P_i$ and $P_o$ results obtained with $^{31}P$ NMR using NaOH-EDTA from different studies, allows us to describe the effect of state factors on soil P composition in natural ecosystems. We hypothesize that the compounds comprising soil $P_i$ and $P_o$ will be modified by distinctive edaphic and climatic properties due to different key ecological processes coupled with soil P cycling.

## 2 METHODS

### 2.1 Dataset

A database search was conducted until November 17, 2017, to identify published papers that accurately determined soil P compounds through one-dimensional liquid state $^{31}P$ NMR on NaOH-EDTA extracts. According to McDowell et al. (2006) and Cade-Menun and Liu (2014), we consider as accurate the papers that used an adequate delay time prior to the NMR analysis, therefore enabling the production of quantitative data on the NMR instrument. We used two platforms and specific search terms for each one. The first platform was the Web of Knowledge. The following terms were used: "soil* phosphorus or P or 31P* nuclear-magnetic-resonance or NMR* naoh or sodium hydroxide* edta or ethylenediaminetetraacetic" from which 129 results were obtained. The second platform was Google Scholar. The following terms were used: "soil* phosphorus* "nuclear magnetic resonance"* naoh* edta", which yielded 2,190 results (excluding patents and citations).

We followed pre-defined eligibility criteria to consider the papers, and then to select or reject them. The first criteria was that only native growth media were considered (manure, pot soil, soil leachate and sediment samples were excluded). In studies focusing on changing natural conditions, only the control (unchanged) samples were used, (e.g., litter removal in Vincent et al. 2010 was excluded). Next we only considered studies where the one-dimensional liquid state $^{31}P$ NMR method was used with the following features: 1) NaOH-EDTA extractor without pretreatment (0.5 or 0.25 M NaOH and 0.1 or 0.05 M EDTA), 2) delay times > 2 s (i.e., quantitative data, see Cade-Menun and Liu 2014); 3) NMR features or explanations according to $^{31}P$ NMR principles (see Cade-Menun and Liu 2014); and 4) total NaOH-EDTA extracted P and total P. Both top mineral and organic layers were considered. From selected papers, we compiled: total P, total NaOH-EDTA P, and NaOH-EDTA organic P, as well as the $P_o$ compounds inositol hexakisphosphates (*myo*-, *scyllo*-, *neo*-, and D-*chiro*-IHP, when available), deoxyribonucleic acid (DNA), phosphonates, NaOH-EDTA inorganic P, and the $P_i$ compounds orthophosphate, pyrophosphate and polyphosphate. No duplicity was found in the selected papers.

### 2.2 Site environmental properties

Soil texture, total C, total N and pH, short range ordered Al and Fe minerals (poorly crystalline) estimated with oxalate extraction, climate characteristics (mean annual precipitation and mean annual temperatures), as well as soil age, when

available, were also collected from the papers. When the total C was unavailable, the organic C was assumed to be the total C. This assumption only occurred for non-calcareous soils. Some variables were unavailable for some results, and the number of experimental used for each analysis is presented in the results section. Missing texture and total C data (representing 12 sites and one site, respectively) were extracted from a global soil dataset, SoilGrids, which is now at 250 m resolution (0-20 cm topsoil, Hengl et al. 2017). The resulting dataset is available in Appendix S1. We used the Whittaker's diagram (Whittaker 1975) and the "BIOMEplot" package (Kunstler 2014) to determine the biomes of our sites (Appendix S2).

Soil weathering stages were derived from the soil type according to Cross and Schlesinger (1995) and Yang and Post (2011) as well as from chronosequence positions. A low weathering stage was attributed to Entisol, Mollisols and Inceptisols forming the first stages of chronosequences and gleyed Acrisols. An intermediate weathering stage was attributed to Alfisol, Aridisol, Mollisols and Inceptisols forming the intermediate stages of chronosequences and orthic Acrisol. Finally, a high weathering stage was attributed to Oxisol, Spodosol, Ultisol, and humic Acrisol.

## 2.3 Data analysis

Statistical analyses were conducted on R Version 3.1.0 (© 2014 The R Foundation for Statistical Computing) using mixed-regression models including edaphic and climatic variables as continuous and categorical fixed effects. The latitude, the percentage of P extracted with NaOH-EDTA and the soil sampling depth were considered as random effects. Latitude was used to control for the spatial auto-correlation (Maestre et al 2005). The percentage of P extracted with NaOH-EDTA was used because the NaOH-EDTA extraction process varies according to soil characteristics and experimental conditions (i.e., pretreatment, soil-to-solution ratio and soil characteristics) (Cade-Menun and Liu 2014, see Fig. S5). Sampling depth was used because of potential differences in organic matter concentration along the soil profile. The bivariate effects of latitude, percentage of P extracted and sampling depth on the soil P composition are presented in Appendices S3-S5.

We used variation partitioning and Venn diagrams (Legendre and Legendre 2012) (the '*vegan*' package) to partition the total variation explained uniquely by the matrix of either soil variables, climate variables or soil weathering stages as well as the variation explained by the combined effect of these matrices. The unique effect of soil, climate or soil weathering stages was calculated as the adjusted $r^2$ value ($r_a^2$) difference between the full model and unique model. The joint effect of these matrices was calculated as the difference between the summed $r_a^2$ of unique models and the $r_a^2$ of the full model.

Structural equation modeling (path analysis, the '*lavaan*' package) was used to explore how variations in $P_i$ and $P_o$ compounds are driven by both direct and indirect effects of key environmental drivers (soil, climate and parent material). We first established an *a-priori* model that is based on our knowledge and is presented in Appendix 10. Then, we tested for the most parsimonious model among many alternative ones, i.e. the one that differed the least from the observations and presented the greater *P*-value. Parent material, which was unknown for our sites, was considered as a latent variable in the model to explain the remaining coordinated variations in total P, clay and pH variables that were not explained by climate and weathering. As mainly determined by the biota, total C was not considered as being constrained by the parent material.

Different units were used across statistics to analyze soil P composition. The bivariate relationships (Figures 2, 3, and 5, and Appendices S3-S8) considered: i) total $P_i$ or $P_o$ concentration in NaOH-EDTA extracts (mg kg$^{-1}$ soil), ii) proportion of total $P_i$ or $P_o$ as percentage of total NaOH-EDTA P (% of NaOH-EDTA P), and iii) proportion of soil P compounds as percentage of their respective pools (% of NaOH-EDTA $P_i$ or $P_o$). In contrast, in both Venn diagrams (Figure 6) and structural equation modeling (Figure 7) soil P compounds were in mg kg$^{-1}$. In bivariate relationships, we used percentages was to compare the relative composition of P along environmental variables that are linked with the weathering of soils. In both the Venn diagram and path analysis, the objective was to explain soil P composition either partitioning the variation among state factors or accounting for the causal structure of environment. For that, we used the mg kg$^{-1}$ unit so that the distribution of our variables was not constrained as a proportion.

## 3 RESULTS

Our search resulted in 100 native vegetation outcomes from 13 references (Appendix S1) (Backnäs et al. 2012, n=1; Celi et al. 2013, n=4; Doolette et al. 2017, n=5; Li et al. 2015, n=1; McDowell and Stewart 2006, n=4; McDowell et al. 2007, n=26, Turner and Engelbrecht 2011, n=19; Turner et al. 2003, n=1, Turner et al. 2007, n=8; Turner 2008b, n=1; Turner et al. 2014, n=20; Vincent et al. 2013, n=8; Vincent et al. 2010, n=1). Most of the papers were excluded (from more than 2,000 papers found during the search) because they failed to meet the eligibility criteria including land use (e.g., crop, pasture, planted forest or wetlands) and $^{31}$P NMR features. The results selected were from the following countries: Australia (n=5), Finland (n=1), Italy (n=1), New Zealand (n=59), Republic of Panama (n=21), Russia (n=4), Sweden (n=8) and the United States of America (n=1). These results comprised most of the global biomes classified according to Whittaker's diagram (Whittaker 1975), except for the subtropical desert, tundra and temperate rain forest. The six chronosequences studies (5 in New Zealand, 1 in Sweden; 5 on A layer, 2 on O layer) were the most important contributors to the data (45/74 sites on A layer, 18/20 sites on O layer).

In the compiled data, 80% of results were from mineral layers and the remaining 20% from organic layers; 39% did not contain inositol hexakisphosphates results (including all tropical regions), and 12% of DNA results were absent (including both non-tropical and tropical regions). The P extracted with NaOH-EDTA on mineral layers averaged 55% (2 to 98% range), and on organic layers averaged 73% (57 to 94% range) of total soil P. The average sample depth was 12.2 cm (2 to 42 cm range) for mineral layers and 11.0 cm (1 to 28 cm range) for organic layers.

### 3.1 Edaphic properties

All surveyed edaphic properties affected soil $P_i$ and $P_o$ pools and compounds. These results are summarized in Figures 2 and 3. Both total $P_i$ (Figure 2A) and $P_o$ (Figure 3A) concentrations in NaOH-EDTA extracts (mg kg$^{-1}$ soil) had a quadratic response to soil pH, with higher values occurring at an intermediate pH, although this effect was constrained to the mineral layers. In mineral layers, the proportion of NaOH-EDTA P in the form of $P_i$ increased with pH (Figure 2B), whereas the

proportion in the form of $P_o$ decreased with pH (Figure 3B). However, there was no pH effect on either pool in the organic layers. The distribution of compounds in both $P_i$ (Figure 2C-E) and $P_o$ (Figure 3C-D) pools responded dynamically to the pH. In the $P_i$ pool (% of $P_i$) of mineral layers, orthophosphate decreased, and pyrophosphate accounted for the remaining $P_i$ as the pH decreased. The pH had no effect on these $P_i$ compounds in the organic layer (even though there is an apparent trend, these relationships became insignificant after including sampling depth as a random effect on models; Appendix S3 shows the sampling depth effect over the soil P composition). In the $P_o$ pool (% of $P_o$), both inositol hexakisphosphates (mineral layer) and DNA (mineral and organic layers) proportions increased as the pH decreased. Phosphonates response to edaphic properties (insignificant) is presented in Appendix S6.

Both total $P_i$ and $P_o$ concentrations in NaOH-EDTA extracts (mg kg$^{-1}$ soil) responded quadratically to the clay concentration, with higher values occurring at intermediate textural classes (Figures 2F and 3E). Clay impacted on neither $P_i$ and $P_o$, nor on the proportions of P compounds (% of NaOH-EDTA P) (Figures 2G-J and 3F-H).

Total $P_i$ and $P_o$ concentrations in NaOH-EDTA extracts (mg kg$^{-1}$ soil) increased as the soil C concentration increased in mineral layers, whereas in organic layers there was no C concentration effect on $P_i$ and $P_o$ concentrations (Figures 2K and 3I). As a percentage of NaOH-EDTA P, $P_i$ decreased and $P_o$ increased (% of NaOH-EDTA P) as the soil C concentration increased in mineral layers, and there was no C concentration effect on either $P_i$ and $P_o$ proportions in organic layers (Figures 2L and 3J). In the $P_i$ pool (% of $P_i$) of mineral layers, orthophosphate and pyrophosphate proportions decreased and increased, respectively, as the soil C concentration increased (Figure 2M-O). As the soil C concentration increased in the organic layer, orthophosphate decreased, at a greater extent when compared to the mineral layer, pyrophosphate decreased (in contrast to the mineral layer, in which it increased), while the polyphosphate proportion increased, and gradually dominated the $P_i$ pool at greater soil C concentrations. In the $P_o$ pool (% of $P_o$), there was no C concentration effect on the soil $P_o$ composition (phosphonates, inositol hexakisphosphates and DNA) in either mineral or organic layers (Figure 3K-L; Appendix S6).

Both total $P_i$ and $P_o$ concentrations in NaOH-EDTA extracts (mg kg$^{-1}$ soil) from both mineral and organic layers increased as the total soil P concentration increased (Figures 2P and 3M). Only the DNA compound from the $P_o$ pool (% of $P_o$) in the mineral layer was affected by the total soil P concentration (Figure 3P). As the total soil P concentration increased, the DNA proportion in the $P_o$ pool decreased. It is important to note that the reported total P (x axis on Figures 2 and 3) is the one obtained by digestion and it includes both the extracted P and the residual P. The recovery of the total P by NaOH-EDTA extraction varies depending on soil characteristics and laboratory procedures (Cade-Menun and Liu 2014).

Total $P_i$ and $P_o$ concentrations in NaOH-EDTA extracts (mg kg$^{-1}$ soil) were only affected by the soil CP ratio in organic layers. Both total $P_i$ and $P_o$ concentrations decreased as the soil CP ratio increased (Figures 2U and 3Q). As a percentage of NaOH-EDTA extract (% of NaOH-EDTA P), $P_i$ decreased while $P_o$ increased, both exponentially, as the soil CP ratio increased in mineral layers (Figures 2V and 3R). As proportions in the $P_i$ pool (% of $P_i$) of mineral layers, orthophosphate decreased and pyrophosphate increased as the soil CP ratio increased (Figure 2W-X). In the $P_i$ pool (% of $P_i$) of organic layers, proportions of orthophosphate decreased and polyphosphate increased, gradually dominating the $P_i$ pool as

the soil CP ratio increased (Figure 2W-Y). In the $P_o$ pool (% of $P_o$), the DNA proportion increased as the soil CP ratio increased, only in the mineral layer (Figure 3T).

## 3.2 Climatic properties

Climatic properties affected soil $P_i$ and $P_o$ pools and their composition only through the mean annual precipitation. These results are summarized in Appendices S7 and S8. The mean annual temperature, ranging from -0.4 to 27 Cº, did not promote any change in the soil P composition in natural ecosystems. There was no effect of climatic variables on total $P_i$ and $P_o$ concentrations in NaOH-EDTA extracts (mg kg$^{-1}$ soil) (Appendices S7A and S8A). As a fraction of the NaOH-EDTA extract (% of NaOH-EDTA P), $P_i$ decreased and $P_o$ increased as the precipitation increased (Appendices S7B and S8B). As the precipitation increased, proportions of orthophosphate decreased and pyrophosphate increased as compounds of the $P_i$ pool (% of $P_i$) (Appendix S7C-D).

## 3.3 Soil weathering stages

Soil weathering stages determined from the soil type and chronosequence positions affected soil age and CP ratios following an expected effect of pedogenesis (Figure 4). As soil weathering stages increased, the soil age and CP ratio also increased. Both $P_i$ and $P_o$ pools were affected by the soil weathering stage (Figure 5). Total $P_i$ and $P_o$ in NaOH-EDTA extracts (mg kg$^{-1}$ soil) were more concentrated in soils at moderate weathering stages when compared to low and high weathering stages (n=79, Figure 5A, F). As percentages in the $P_i$ pool (% of $P_i$), orthophosphate decreased and pyrophosphate increased as the soil weathering stage increased (n=79 for all $P_i$ compounds, Figure 5C-D). In the $P_o$ pool (% of $P_o$), the DNA (n=64) proportion was greater in more weathered stages, and there was no effect of weathering stages on phosphonates (n=79) and inositol hexakisphosphates (n=52) proportions (Figure 5H-J). Using available data (n=49), we observed no effect of soil weathering stages on short range ordered (poorly crystalline) Al and Fe minerals estimated with oxalate extraction (p>0.1, Appendix S9).

## 3.4 Variation partitioning among edaphic, climatic and weathering on the soil P composition

The variation partitioning of ecosystem properties governing the soil P composition (in mg kg$^{-1}$ soil) was generally more pronounced for soil variables (pH, clay concentration and total P and C concentrations) than climatic variables (precipitation and temperature) and soil weathering (Figure 6). For the total $P_i$ concentration and its compounds orthophosphate and pyrophosphate, the total variation explained by models ranged from 46% to 89%, and they were mostly explained by soil variables and combined effects of soil and weathering. Polyphosphates had a poorly-defined response to the variation partitioning of ecosystem properties (<0.01% of the total variation explained).

In the $P_o$ pool, the total variation explained by models ranged from 41 to 86% (Figure 6). The total $P_o$, inositol hexakisphosphates and DNA had their total variation mostly explained by soil variables, and to a lower degree, but more

pronounced for the DNA compound, by combined effects of soil variables and weathering. In contrast, most of the variation in phosphonates was explained by combined effects of climate and soil variables, followed by uniquely soil variables.

### 3.5 Interdependences between environmental variables and soil P compounds

We used path analyses to explore the interdependences between edaphic and climatic variables and how they relate to the soil $P_i$ and $P_o$ compounds (Figure 7; Appendix S10). The parent material was used as a latent variable (set by the pH) in both models ($P_i$ and $P_o$). Climate and soil weathering drivers were independently related to soil variables (total P, pH, clay and total soil C), and soil variables were considered direct effects in the models. The most parsimonious path analysis model explained up to 78% of $P_o$ compounds variation and 89% of $P_i$ compounds variation.

Following an expected effect of pedogenesis, the path analysis indicated that the parent material (latent variable) was positively related to the soil total P, clay and pH. Greater mean annual precipitation was negatively related to the soil total P, pH, and it positively influenced soil total C, while clay was negatively influenced by precipitation in the $P_o$ model only. In the $P_o$ model, precipitation promoted soil weathering, whereas in the $P_i$ model, soil weathering was positively affected by temperature. The mean annual temperature positively affected the clay and pH. Soil weathering was negatively related to the soil pH, and positively related to the soil clay and total C. In the $P_i$ model only, soil weathering negatively affected soil total P. There were also significant direct and positive effects between soil total C and clay, and total P, in both $P_i$ and $P_o$ models, and there was a positive relationship between soil total C and pH in the $P_o$ model only.

In the $P_i$ model, orthophosphate was negatively related to precipitation, and it was positively influenced by soil total P and total C. Pyrophosphate was positively influenced by precipitation, soil total P and total C. Polyphosphate was negatively influenced by temperature, and it was positively related to soil pH. In the $P_o$ model, inositol hexakisphosphates were negatively affected by precipitation and temperature, but positively affected by the soil total P, total C and pH. In contrast, total P and total C positively affected DNA, and there were no effects of climatic variables over DNA. Phosphonates were negatively affected by temperature and weathering, but positively affected by precipitation and soil total C.

### 4 DISCUSSION

Our results showed how soil $P_i$ and $P_o$ compounds responded to edaphic variables (Figures 2 and 3), climatic variables (Appendices S7 and S8), and soil weathering stages as a proxy for pedogenesis (Figures 5) on a wide geographical scale, including a variety of natural ecosystems. While the soil P composition was primarily directly influenced by soil properties, the impact of climate and weathering stage occurred mainly through indirect paths and their influence on soil properties (Figures 6 and 7). In addition, soil $P_i$ and $P_o$ compounds responded to different combinations of explicative variables, which likely indicates that each P compound has specific factors governing its presence, transformation and persistence in ecosystems. This could be due to many factors including: i) the source of P inputs, primarily by minerals, and then altogether

with plant and microbe P cycling; ii) the presence of specific phosphatase enzymes that are required to transform $P_i$ and $P_o$ compounds into orthophosphate; iii) the soil specific reactivity and P losses governed by physico-chemical properties (e.g., clay, short-range ordered oxides and pH); and iv) the P persistence induced by the increasing complexity of $P_i$ and $P_o$ compounds as pedogenesis evolves.

As time passes after the onset of pedogenesis, the ecosystem accumulates organic matter up to a maximum, and then starts to decline. Along with this decline, there are also changes in the chemical composition of organic matter, in which the decaying degree (i.e. decomposition) of C element is lower than the P, and concomitantly there is an increasingly acidic soil environment (Walker 1965). In addition, parent material supplies cations and orthophosphate to young soils, whereas more weathered soils are substantially changed from the parent material. Consequently, highly weathered soils generally

have higher CP ratios, a lower pH and greater clay concentration. The soil total P content depends on both weathering stages and parent material, but generally decreases with increasingly weathered soil orders (Yang and Post 2011). Our data included soil orders ranging from all three stages of soil weathering (low, intermediate and high), according to Cross and Schlesinger (1995) and Yang and Post (2011). The soil weathering stage classification also takes into account changes in the soil P fractions, and generally follows the Walker and Syers (1976) conceptual model: there is gradual decrease and eventual

depletion of primary mineral P (mainly apatite P), decrease of total P, increase and then decrease of total $P_o$ and increase and eventual dominance of occluded P during the soil development (Yang and Post 2011). In highly weathered soils, occluded P increases through the encapsulation of the $P_i$ and $P_o$ compounds inside of Fe and Al minerals (McDowell et al. 2007; Turner et al. 2007).

        Even though most results were from New Zealand and Panama, our dataset comprised several biomes according to

the Whittaker's diagram (Whittaker 1975), including the temperate grassland desert, woodland shrubland, temperate forest, boreal forest, tropical rain forest, tropical forest savanna and intermediates between the temperate rain forest and boreal forest, and tropical rain forest and temperate rain forest (Appendix S2); however, quantitative data on the feedback between P compounds and biological communities during pedogenesis is still incipient to conclusions drawn from the influence of vegetation and organisms on the soil P composition (Huang et al. 2017), especially for [31]P NMR results. What is clearer is

how soil P availability shapes the ecosystem's overall primary productivity, and to a lesser extent, soil food webs. In a global analysis, Maire et al. (2015) demonstrated that the soil available P is a key environmental dimension increasing leaf P content along with species' maximum photosynthetic rates and lower stomatal conductance. However, this trend is expected to gradually decline in more weathered soils due to a lower P availability. Conducted at a narrower scale, Laliberté et al.'s study (2017) showed that soil fertility (including P availability) strongly shaped underground food webs, promoting changes

such as a shift in dominance from bacterial to fungal energy channels with increasing soil age.

### 4.1 Soil properties and the soil P composition

As soils aged, pyrophosphate and polyphosphate may have accumulated because of the incorporation and stabilization of these compounds (biological origin) into soil organic matter (Turner et al. 2007). The soil pH, total carbon and CP ratio, as

well as weathering stage had a major influence on the soil $P_i$ pool composition. As the orthophosphate proportion decreased in more weathered, acidic, organic-rich, and P-limited soil environments (Figures 2C, M, W and 5C), pyrophosphate and polyphosphate proportions increased and dominated the $P_i$ pool (Figures 2D, N, O, X and Y and 5D).

Even though pyrophosphate and polyphosphate are $P_i$ compounds, they have a biological origin (Turner and Engelbrecht 2011). Condensed forms of P (including pyrophosphate and polyphosphates) are found in every bacterial, archaeal and eukaryotic cell, but in highly variable amounts (Kornberg et al. 1999). Bünemann et al. (2008) found a positive relationship between the proportion of fungi and the amount of pyrophosphate, and Reitzel and Turner (2014) found a positive link between the pyrophosphate proportion and soil microbial P. Polyphosphate can originate from ectomycorrhizal fungi (Koukol et al. 2008), and there are some ectomycorrhizal fungi specialized for P uptake in low P, acidified soil conditions (Wang and Qiu 2006). Ectomycorrhizal fungi convert the orthophosphate that they take up from the soil into polyphosphates, and translocate the polyphosphate along fungal hyphae, sometimes at a great distance from where the orthophosphate is taken up (Bücking and Heyser 1999; Plassard and Dell 2010). Therefore, we believe that pyrophosphate and polyphosphate dominated the $P_i$ pool in acidic, P-limiting (CP ratio), and high organic matter (total C) soils because of the microbial origin of these P sources, but much information is still needed in regard to plant and microbial communities characterization in studies of P forms. These organisms could have helped to deplete and transform the bioavailable orthophosphate, turning it into more microbial biomass derived P compounds as pedogenesis progressed in these environments.

Moreover, polyphosphates tend to occur in abundance only in soils where decomposition is slowed, such as acidified soil conditions, or cold and wet soils high in organic matter (e.g., Cade-Menun et al. 2000; Turner et al. 2004). Studying wetland soils, Cheesman et al. (2014) found that polyphosphates played a preeminent role in P-limited systems, predominantly in acidic, high-organic-matter systems. Adding to that, pyrophosphate hydrolysis was found to be more rapid with greater biological activity and higher agricultural soil pH (Sutton and Larsen 1964), and this may have contributed to reducing the pyrophosphate proportion at a higher pH in mineral soils (Figure 2D). As the C concentration increased in organic layers, polyphosphate dominated the soil $P_i$ pool (Figure 2M-O) possibly because of its lesser lability when compared to orthophosphate and pyrophosphate. Pyrophosphate is less polymerized and potentially more susceptible to hydrolysis than polyphosphate. According to Savant and Racz (1972), Subbarao et al. (1977) and Dick (1985), pyrophosphate is hydrolyzed more rapidly than polyphosphate because pyrophosphate is an intermediate product of polyphosphate hydrolysis until the final orthophosphate is produced.  These are also conditions under which ectomycorrhizal fungi are found. These fungi produce  hyphal mats in the forest floor, so an increase in polyphosphates could reflect an increase in ectomycorrhizal  hyphal mats. However, caution must be taken when interpreting pyrophosphate and polyphosphate hydrolyzation results from [31]P NMR analysis on NaOH-EDTA extracts. Polyphosphates can potentially degrade to pyrophosphates during extraction and NMR analysis of P (Cade-Menun et al. 2006), so they cannot be considered as fully distinct P forms. This potential degradation could be one explanation to why the polyphosphates results were poorly explained by the variation partitioning and structural equation models.

As time passes after the onset of pedogenesis, modifications in the soil $P_o$ composition were possibly related to the acidifying environment in soils, which may increase the charge of some $P_o$ compounds, and thus increase sorption. Soil pH affects the sorption of $P_i$ and $P_o$ compounds by soils, but different P compounds respond differently to pH changes (Shang et al. 1992). Shang et al. (1992) verified that sorption of both orthophosphate and inositol hexakisphosphate by Al and Fe precipitates generally decreased as pH increased, whereas there was little pH effect on the adsorption of glucose 6-phosphate by both precipitates. Sorption of inositol hexakisphosphate to minerals surface is often stronger than orthophosphate, but both tend to be less sorbed to those minerals in neutral to alkaline soils (Berg and Joern 2006; Xu et al. 2017). The presence of humic acids may affect the amount of inositol hexakisphosphate that sorbs on the minerals surface at lower pH values, but it cannot displace inositol hexakisphosphate from that surface (Ruyter-Hooley et al. 2016). Moreover, another study founds that inositol hexakisphosphate sorption in soils was unaffected by the presence of orthophosphate, β-D-glucose-6-phosphate or adenosine 5′-triphosphate (Berg and Joern 2006). Following a similar pattern, the amount of DNA bound on clay minerals such as montmorillonite, kaolinite, hydroxyl aluminum species and variable charge soil colloids also increased by lowering pH of solution (Khanna and Stotzky, 1992; Cai et al. 2006; Cai et al. 2008; Wang et al. 2009; Saeki et al. 2010). At pH < 5, protonation of the amino groups of adenine, guanine and cytosine occurs and causes the increase of net positive charge of DNA and electrostatic attraction between negatively charged tetrahedral silica layer on the clay surface and DNA (Cai et al. 2006). Therefore, the increasingly acidic pH could have increased sorption, and therefore facilitated inositol hexakisphosphates (Figures 3C) and DNA (Figure 3D, Figure 7) accumulation in those soils.

In our study, we found that there was an increasing proportion of inositol hexakisphosphates and DNA in the $P_o$ pool (% of NaOH-EDTA P) as pH decreased, and there was predominance of inositol hexakisphosphates in acidic, more weathered soils (Figures 3C-D, 5J). The hierarchy of investment for P acquisition through enzymatic activity may also be a factor that contributed to modifications in the soil $P_o$ composition as time passes since the onset of pedogenesis. According to Turner et al. (2018), Turner and Haygarth (2005) and Kunito et al. (2012), P limitation may stimulate increased phosphoesterase synthesis as a way to increase bioavailable P by the mineralization of $P_o$. Fungi are well known for their capacity to secrete acid phosphatases (Rosling et al. 2016), and are usually the predominant microorganisms in acidic natural soils; while alkaline phosphatase and phytase genes are distributed across a broad phylogenetic range and display a high level of microdiversity (Zimmerman et al. 2013). Plants and microorganisms that breakdown orthophosphate diesters need a higher investment for the P acquisition (see Turner 2008a for the conceptual model) than orthophosphate monoesters, since they require hydrolysis by both phosphodiesterase and phosphomonoesterase to release available phosphate, whereas orthophosphate monoesters require only the last one (Halsted 1964; Skujins 1967; Tabatabai and Bremner 1969). Although inositol hexakisphosphates are also classified as part of orthophosphate monoesters, they need a higher investment in $P_o$ acquisition than other orthophosphate monoesters and diesters because they can be strongly bounded by metal oxides, clays and organic matter (Turner et al. 2007). Organic acids are required to desorb inositol phosphates, so that phytases can hydrolyze them to release a free orthophosphate. This would suggest that as soil gets more weathered, inositol hexakisphosphates may accumulate more than DNA, and the latter more than other $P_o$ compounds.

Nonetheless, inositol hexakisphosphates can persist as a main $P_o$ compound up to a certain point of pedogenesis, and then decline in more weathered soils. Some authors described that inositol hexakisphosphates declined to lower concentrations in older soils of non-tropical regions (Turner et al. 2014; Turner et al. 2007), and most tropical soils had negligible inositol hexakisphosphates contribution (e.g., Turner and Engelbrecht 2011). In our results, soil weathering had no

effect on inositol hexakisphosphates concentrations in mg kg$^{-1}$ in non-tropical environments (Figure 7), but in fact, all compiled results from tropical soils had no inositol hexakisphosphates. Inositol hexakisphosphates have been found in very weathered soils (e.g., Oxisols), but under agricultural management that included well-known sources for that P compound such as plant seeds (Turner 2006; Smernik and Dougherty 2007; Deiss et al. 2016). These results suggest that inositol hexakisphosphates could occur in tropical soils under native vegetation, but they are either being rapidly turned into

bioavailable compounds (by plants and microorganisms) or inputs of inositol hexakisphosphates, which are abundant in seeds and pollen (Raboy 2007), are lower in lowland tropical forests compared to temperate ecosystems (Turner and Engelbrecht 2011). In P limited soil environments, the acquisition of inositol hexakisphosphates may be strongly improved by root exudates, which may increase the solubility of these compounds in soil (Gerke 2015). In addition, mineralization of myo-inositol hexakisphosphate by ectomycorrhizal fungi (Chen et al. 2004) may also have contributed to its decline in more

weathered, acidic soils, due to fungi predominance in these environments. Finally we believe that other P compounds such as DNA (Figures 3D, 5J) and pyrophosphate (Figures 2D, 5D) will possibly prevail in more weathered systems from tropical regions because they are intrinsic components of the microbial biomass (Kornberg et al. 1999). Mature soils are known for having the microbial P as the main component of the P pool (Turner et al. 2013).

Soil clay concentration affected both soil total $P_o$ and $P_i$ concentrations (Figures 2F and 3E), but had a minor

association with soil P compounds. Recent investigations have found that organic compounds stabilization may be mainly driven by factors that are typically minor constituents of the clay-sized fraction by mass, but highly reactive components. Vogel et al. (2014) showed that organic matter is preferentially stabilized in certain hot-spot zones (i.e., rough surfaces), and that only a limited portion of clay-sized surfaces contributed to soil organic matter stabilization. This concept was further tested for P by Werner et al. (2017) who found that microscale spatial heterogeneity influences P accessibility and

bioavailability in soil aggregates, depending on soil substrate and depth. They also found that in P-rich areas of soil aggregates, the P was predominantly co-located with Al and Fe oxides, while in low-P topsoil aggregates, most of the P was organically bound (Werner et al. 2017). Yang et al. (2016) also showed that only limited portions of fine mineral surfaces contributed to soil organic matter stabilization. So, these facts could justify the minor association between bulk soil clay concentration and soil P compounds as observed in our study.

Soil organic matter stabilization could be facilitated in more weathered soils by the potential increase in amorphous Al and Fe oxides (Albrecht 1957; Walker 1965), and consequently more reactive surface area availability to absorb and stabilize soil organic matter; however, we did not observe a significant overall soil weathering effect on Al and Fe oxide concentrations (Appendix S9), suggesting that it may depend on other factors such as the parent material or specific soil orders. Moreover, contrasting soil organic matter responses to short-range-ordered (amorphous) Fe or Al oxides have been

found in literature. Some investigations found a pronounced role promoted by Al oxides (Heister 2016; Kaiser et al. 2016), whereas others found Fe oxides as the main soil organic matter stabilizing mechanism (Wilson et al. 2013; Catoni et al. 2016; Deiss et al. 2017). Investigations also found no apparent relationship between soil organic matter and both Al and Fe oxides (Cloy et al. 2014; Vogel et al. 2015; Rumpel et al. 2015). Therefore, we could not confirm the role of Al and Fe oxides as influenced by soil weathering stages over the soil P composition.

## 4.2 Climate and the soil P composition

Climatic variables exerted an important role on the soil P composition but to a lesser extent when compared to soil variables (Figure 6). Contradicting what we expected, our results showed that temperatures ranging from -0.4 to 27 C° had no effect on both $P_i$ and $P_o$ pools and their compounds (Appendices S7 and S8). It was expected that the soil $P_o$ concentration would decrease with increasing temperatures because higher temperatures are optimal for the breakdown of the soil $P_o$ compounds by the microbial biomass through phosphatase enzymes release (Hui et al. 2013). Hui et al. 2013 confirmed that greater maximum phosphatase activity occurred at incubation temperatures $>25^oC$ when compared to $20^oC$, but no differences were observed among temperatures greater than $25^oC$ (Hui et al. 2013). Therefore, phosphatase activity may depend on the range and magnitude of temperatures; and our results covered a greater range of markedly lower temperatures, which may reduce microbial activity variability even more due to a slowdown in the microorganisms' metabolism.

In contrast, precipitation affected several variables in the soil $P_i$ pool. This result was also expected based on the classic paper of Walker and Syers (1976), which suggested that pedogenesis depends predominantly on the volume of water leached through soil. In our results, the soil total P concentration, pH (Figure 7), and orthophosphate proportion (Appendix S7C) were negatively related to precipitation. As precipitation increased (Appendix 7C) and the soil was in a higher weathering stage (Figure 5C), the orthophosphate proportion (% of NaOH-EDTA P) possibly decreased because of increased leaching. However, caution needs to be used when discussing changes in orthophosphate extracted by NaOH-EDTA for $^{31}$P NMR, because NaOH-EDTA will preferentially extract $P_o$ rather than orthophosphate. As such, studies that analyzed the residual P after NaOH-EDTA extraction have shown that it is mainly composed by orthophosphate (e.g. Cade-Menun et al 2005). Feng et al. (2016) evaluated P fractions along a climosequence and observed that greater precipitation (in soils with no impeded drainage) reduced the inorganic concentration of P linked to Ca, corresponding to a marked decline in soil exchangeable Ca and suggesting an enhanced leaching of P along with weatherable cations. Moreover, greater soil water availability, and consequent greater primary productivity, may have increased the demand for P in its bioavailable form and contributed to the orthophosphate depletion.

As the orthophosphate percentage and concentration (Appendix 7C and Figure 7, respectively) decreased following greater precipitation, the pyrophosphate percentage and concentration (Appendix 7D and Figure 7, respectively) increased suggesting that this compound predominates under these environmental conditions. As previously described, this may be due to the incorporation of these compounds into recalcitrant soil organic matter (Turner et al. 2007). Moreover, given the microbial origin of pyrophosphate and its association with the microbial P biomass (Koukol et al. 2008; Turner and

Engelbrecht 2011; Reitzel and Turner 2014), pyrophosphate (Figure 7, Appendix 7D) possibly mirrored the response of the total $P_o$ (Appendix 8B) to climatic variables, which may have resulted from greater soil organic matter accumulation following greater productivity (i.e., plants and organisms) in these ecosystems with greater water availability. Evaluating the P budget of the whole ecosystem, Turner et al. (2013) demonstrated the dominance of microbial P in mature soils. Wang et al. (2014) found that greater $P_o$ concentrations were associated with increasing biomass production (i.e., primary production and microbial biomass) because plants and microbes incorporate P into biomass and return it to the soil. However, it is important to note that the majority of P in plant biomass is as orthophosphate (e.g. Noack et al. 2012) and not as $P_o$ compounds. However, we believe that with higher orthophosphate inputs through plant biomass, soil $P_o$ concentrations would increase altogether with orthophosphate P concentrations, and also at expense of soil orthophosphate due to the greater bioavailability to plants and organisms of that latter P compound.

Changes in vegetation are expected to occur during pedogenesis, and climatic variables may govern magnitudes of these alterations along with soil changes. Vitousek et al. (1995) showed that as ecosystems develop, the pattern of P concentration in plants leaves follows a non-linear response to time, in which lower concentrations occur at either early or late stages of pedogenesis, and a maximum is reached at an intermediate stage of pedogenesis. In addition, precipitation can affect the magnitude of that maximum response (intermediate stage), where the P concentration in plant leaves is higher in mesic gradients when compared to more wet gradients (Vitousek et al. 1995). Moreover, as described earlier, the soil available P, along with other climatic variables, governs maximum photosynthetic rates, but a trend that is expected to gradually decline in more weathered soils, due to a lower P availability (Maire et al. 2015). Phosphorus limitation can become sufficiently intense in the late stages of ecosystem development (also known as the retrogressive phase) to cause a decline in forest biomass, and productivity (Wardle et al. 2004). The exception seems to be tropical forests (Turner et al. 2007), which exhibit very diverse tree communities on old, infertile soils (Losos and Leigh 2004). Moreover, Turner et al. (2018) showed that in lowland tropical ecosystems, P limitation affects individual species, but species-specific P limitation does not translate into a community-wide response, because some species grow rapidly on infertile soils despite extremely low P availability.

## 4.3 Future research priorities

Many efforts have been made to explain soil P composition during pedogenesis; however, a clear picture on how specific plant species, plant functional traits, and their communities can influence the soil P composition is still lacking, especially with results obtained with [31]P NMR. For example, why are inositol hexakisphosphates not found in tropical soils under native vegetation, i.e., is it because the rapid turnover promoted by plants and, or organisms (which one?), or exclusively due to lack of inputs from plants? Does the changes in forest biomass and plant species diversity as soil P turns scarcer contribute to soil P composition in non-tropical environments, either by inputs or P compounds consumption, or the soil *per se* governs both the soil P composition and vegetation dynamics? Therefore, we point out that studies aiming to disentangle

confounding effects among soil biotic and abiotic components, climate and vegetation are required to enable a better understanding of soil P composition in natural ecosystems.

Moreover, studies coupling $^{31}$P NMR with other important techniques (e.g. Liu et al. 2013; Liu et al. 2015) could contribute to a better understanding of P cycling and composition in terrestrial ecosystems. A clear understanding of how orthophosphate species such as Fe-, Al- and Ca-phosphate (Hesterberg 2010; Kizewski et al. 2011) respond to pedogenesis could be elucidated with XANES (see Prietzel et al. 2013; Hashimoto and Watanabe 2014). Studies that quantify specific-P-related enzymes activity (see Turner et al. 2018) in native vegetation soils could help understand if the hierarchy of investment for the P acquisition actually contributes to different degrees of accumulation of inositol hexakisphosphates and DNA as pedogenesis progresses in non-tropical environments, and if phosphatases are leading to a rapid turnover of inositol hexakisphosphates in tropical environments. This can be achieved through determining the presence and abundance of microorganisms and enzymes, and how these changes affect soil P composition. Turnover, exchange kinetics, mineralization rates could be assessed using isotopic techniques (see Frossard et al. 2011), and enable a separation between different sources of P compounds, and their dynamics in soils, organisms and plants.

Finally, we expect future research to provide results of as many soil P compounds as they can find rather than broad compound classes only (i.e. orthophosphate diesters and monoesters), even when compound concentrations are low (and describe when main soil compounds are not detected), which may enable future analyses to avoid possible confounding effects of P compounds inside functional groups (e.g., inositol hexakisphosphates and orthophosphate monoesters) and to make a more precise correction for potential degraded peaks occurring during the alkaline extraction and reading process. We also urge researchers to determine variances or standard errors for soils with distinctive properties. Then, as stated by Stewart (2010), future analyses could use the different information provided by studies of different scopes and quality in a meta-analytical approach.

**5 CONCLUSION**

We conclude that edaphic and climatic properties are important factors in determining soil $P_i$ and $P_o$ pools as well as their compounds, since they regulate key ecological processes governing their presence, transformation and persistence in soils. These processes are related to the source of P inputs, primarily determined by parent material and then altogether through plant and microbe P cycling, the bio-physico-chemical properties governing soil phosphatase activity, soil solid surface specific reactivity and P losses through leaching, and finally the P persistence induced by increasing complexity of $P_i$ and $P_o$ compounds as pedogenesis evolves. Soil drivers that played a preeminent role were soil acidification, C concentration, P limitation determined as CP ratio, soil weathering as the temporal variable, while precipitation was the climatic variables that most influenced soil P composition. Soil P composition was more influenced by soil variables than either climatic variables or weathering stages. However, combined effects among these factors also contributed to explain considerable soil P variability in these ecosystems. Soil $P_i$ and $P_o$ compounds responded differently to combinations of environmental drivers,

which likely indicates that each P compound has specific factors governing its existence in natural ecosystems. Therefore, knowing how environmental drivers affect soil P composition enabled a comprehensive understanding of soil P in natural ecosystems.

## 6 APPENDICES (SUPPLEMENTARY FILES)

**Appendix S1** – Dataset: The dataset is attached as a supplementary Excel file.

**Appendix S2** – Global biomes comprised in our dataset according to the Whittaker' diagram.

**Appendix S3** – Soil depth effect on soil P composition.

**Appendix S4** – Latitude effect on soil P composition.

**Appendix S5** – Percentage of P extracted with NaOH-EDTA effect on soil P composition.

**Appendix S6** – Soil properties and soil organic phosphonates.

**Appendix S7** – Climatic properties and soil inorganic phosphorus.

**Appendix S8** – Climatic properties and soil organic phosphorus.

**Appendix S9** – Soil weathering stages and poorly crystalline Al and Fe concentration.

**Appendix S10** – Models tested to explore the interdependences between edaphic and climatic variables (path analysis) as the main environmental predictors of soil inorganic and organic P compounds.

## 7 ACKNOWLEDGEMENTS

The authors gratefully acknowledge the Coordination for the Improvement of Higher Education Personnel (CAPES-Brazil) and the National Council for Scientific and Technological Development (CNPq-Brazil) for funding this research. Maire is funded by the NSERC Discovery (2016-05716) grant. We sincerely thank the anonymous reviewers for the comprehensive evaluations of our manuscript. We also acknowledge the effort made by all the authors from whose studies we compiled our results. Our study would not be materialized without their work. We truly acknowledge the great effort made by Jordon Wade for his language review.

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

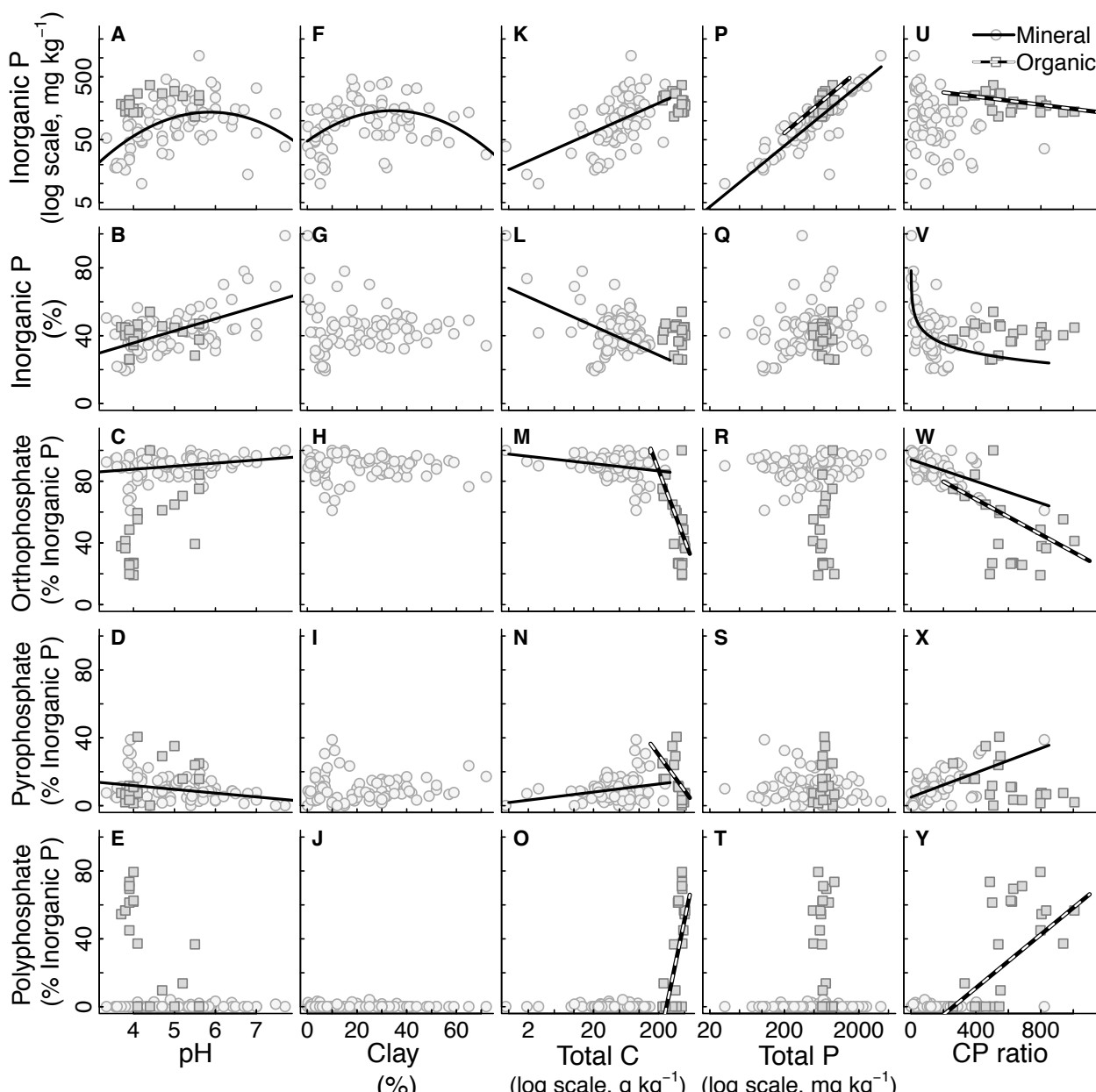

**Figure 2: Relationship between edaphic properties and soil inorganic phosphorus (P) composition in NaOH-EDTA extract from soil mineral and organic layers on natural ecosystems. Note that the reported total P is the one obtained by digestion and usually comprise the residual P non-recovered by the NaOH-EDTA extractant. Regression models (n = 80 mineral layer and n = 20 mineral layer): mineral layer, log(total $P_i$ mg $kg^{-1}$) = -1.62 + 1.28 pH – 0.11 $pH^2$, $r^2$ = 0.33; mineral layer, total $P_i$ (%) = 7.21 + 7.12 pH, $r^2$ = 0.34; mineral layer, orthophosphate = 79.7 + 2.00 pH, $r^2$ = 0.11; mineral layer, pyrophosphate = 20.8 – 2.23 pH, $r^2$ = 0.11; mineral layer, log(total $P_i$ mg $kg^{-1}$) = 1.68 + 0.028 – 0.00041 $clay^2$, $r^2$ = 0.23; mineral layer, log(total $P_i$ mg $kg^{-1}$) = 1.22 + 0.46 log(total C), $r^2$ = 0.32; mineral layer, total $P_i$ (%) = 68.0 - 17.2 log(total C), $r^2$ = 0.14; mineral layer, orthophosphate = 97.6 – 4.74 log(total C), $r^2$ = 0.08; organic layer, orthophosphate = 348.0 – 113.4 log(total C), $r^2$ = 0.30; mineral layer, pyrophosphate = 1.85 + 4.74 log(total C), $r^2$ = 0.08; organic layer, pyrophosphate = 151.7 – 53.0 * log(total C), $r^2$ = 0.34; organic layer, polyphosphate = -446.4 + 184.4 log(total C), $r^2$ = 0.45; mineral layer, log(total $P_i$ mg $kg^{-1}$) = -0.63 + 0.97 log(total P), $r^2$ = 0.73; organic layer, log(total**

$P_i$ mg kg$^{-1}$) = -0.50 + 1.00 log(total P), $r^2$ =0.27; organic layer, log(total $P_i$ mg kg$^{-1}$) = 2.51 – 0.00033 CP ratio, $r^2$ = 0.26; mineral layer, total $P_i$ (%) = 77.0 – 7.88 * log(CP ratio), $r^2$ = 0.33; mineral layer, orthophosphate = 94.0 – 0.0353 CP ratio, $r^2$ = 0.37; organic layer, orthophosphate = 91.1 – 0.00057 CP ratio, $r^2$ = 0.19; mineral layer, pyrophosphate = 5.04+ 0.0359 CP ratio, $r^2$ = 0.37; organic layer, polyphosphate = -20.5 + 0.079 CP ratio, $r^2$ = 0.31.

$P_i$ mg kg$^{-1}$) = -0.50 + 1.00 log(total P), $r^2$ =0.27; organic layer, log(total $P_i$ mg kg$^{-1}$) = 2.51 – 0.00033 CP ratio, $r^2$ = 0.26; mineral layer, orthophosphate = 91.1 – 0.00057 CP ratio, $r^2$ = 0.19; mineral layer, pyrophosphate = 5.04+ 0.0359 CP ratio, $r^2$ = 0.37; organic layer, polyphosphate = -20.5 + 0.079 CP ratio, $r^2$ = 0.31.

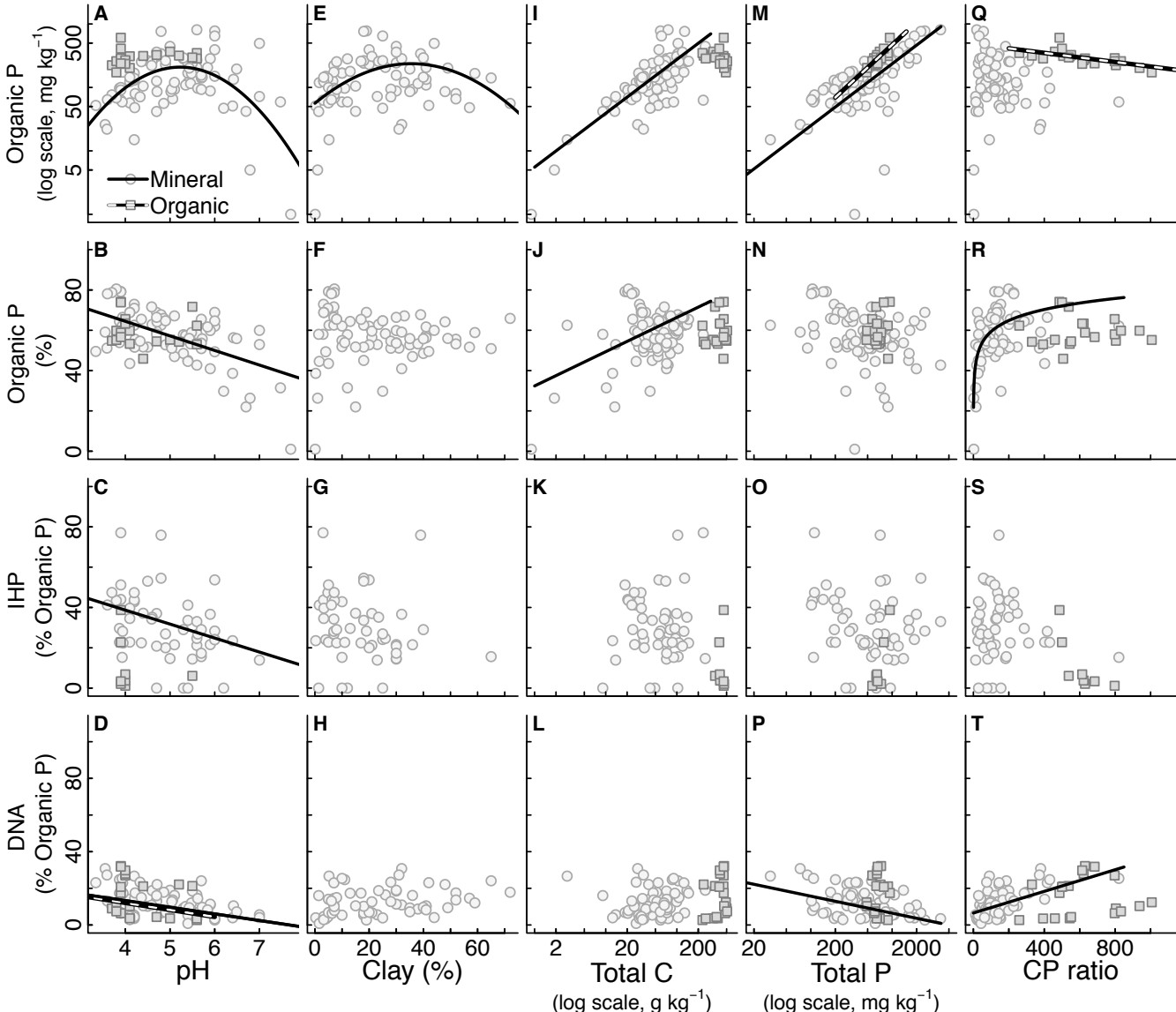

**Figure 3: Relationship between edaphic properties and soil organic phosphorus (P) composition in NaOH-EDTA extract from soil mineral and organic layers on natural ecosystems. Note that the reported total P (x axis) is the one obtained by digestion and usually comprise the residual P non-recovered by the NaOH-EDTA extractant. Regression models: mineral layer, $\log(\text{total P}_o \text{ mg kg}^{-1})$ = -3.61 + 2.27 pH – 0.22 pH $^2$, $r^2$ = 0.34 (n=80); mineral layer, total $P_o$ (%) = 93.4 – 7.24 pH, $r^2$ = 0.35 (n=80); mineral layer, inositol hexakisphosphate (IHP) =66.4 - 6.94 pH, $r^2$ = 0.16 (n=52); mineral layer, DNA = 27.8 – 3.63 pH, $r^2$ = 0.19 (n=64); organic layer, DNA = 27.4 – 3.83 pH, $r^2$ =0.10 (n=20); mineral layer, $\log(\text{total P}_o \text{ mg kg}^{-1})$ = 1.75 + 0.035 clay – 0.00049 clay $^2$, $r^2$ = 0.16 (n=80); mineral layer, $\log(\text{total P}_o \text{ mg kg}^{-1})$ = 31.2 + 120.5 log(total C), $r^2$ = 0.60 (n=80); mineral layer, total $P_o$ (%) = 32.4 + 17.0 log(total C), $r^2$ = 0.12 (n=80); mineral layer, $\log(\text{total P}_o \text{ mg kg}^{-1})$ = -0.55 + 0.97 log(total P), $r^2$ = 0.29 (n=80); organic layer, $\log(\text{total P}_o \text{ mg kg}^{-1})$ = -0.89 + 1.19 log(total P), $r^2$ = 0.68 (n=20); mineral layer, DNA = 34.2 – 9.27 log(total P), $r^2$ = 0.18 (n=64); organic layer, $\log(\text{total P}_o \text{ mg kg}^{-1})$ = 2.69 – 3.58*10$^{-4}$ CP ratio, $r^2$ =0.48 (n=20); mineral layer, total $P_o$ (%) =23.0 + 7.90 * log(CP ratio), $r^2$ = 0.33 (n=80); mineral layer, DNA = 6.53 + 0.029 CP ratio, $r^2$ = 0.34 (n=64).**

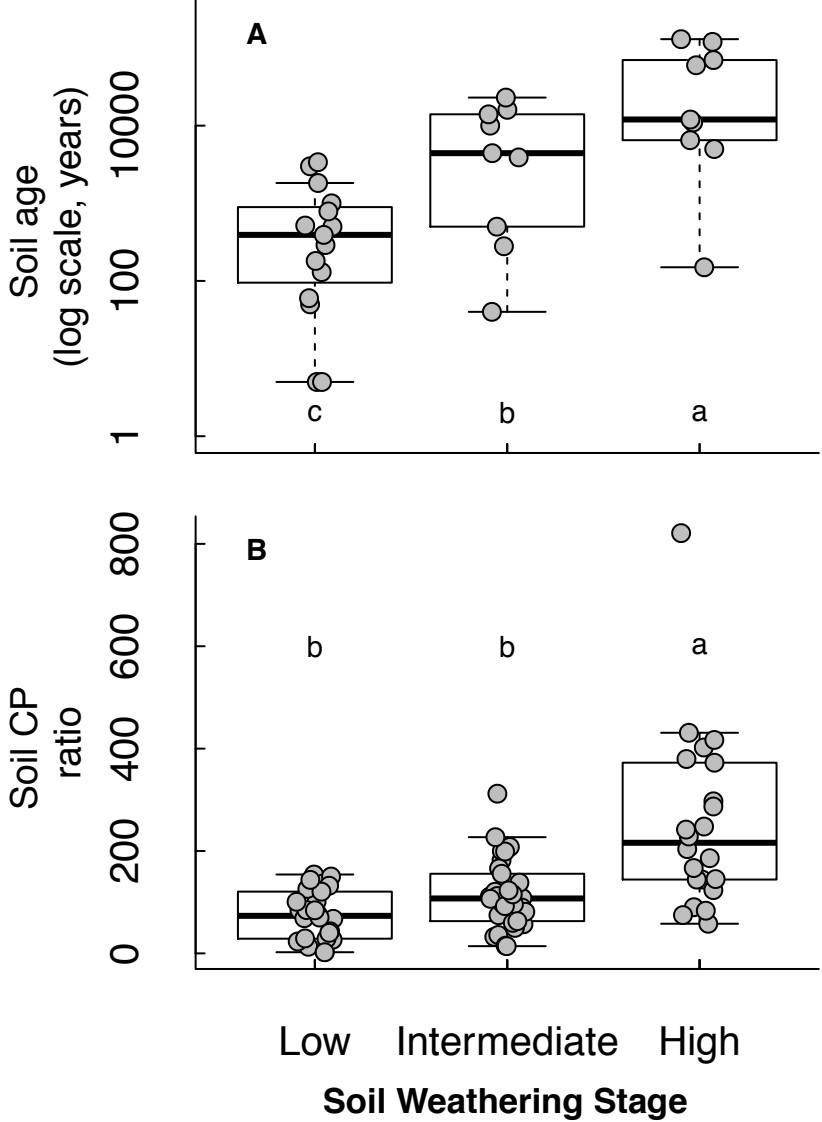

5    **Figure 4: Soil weathering stage relationship with soil age (n = 33) and CP ratio (n = 78) on natural ecosystems.**

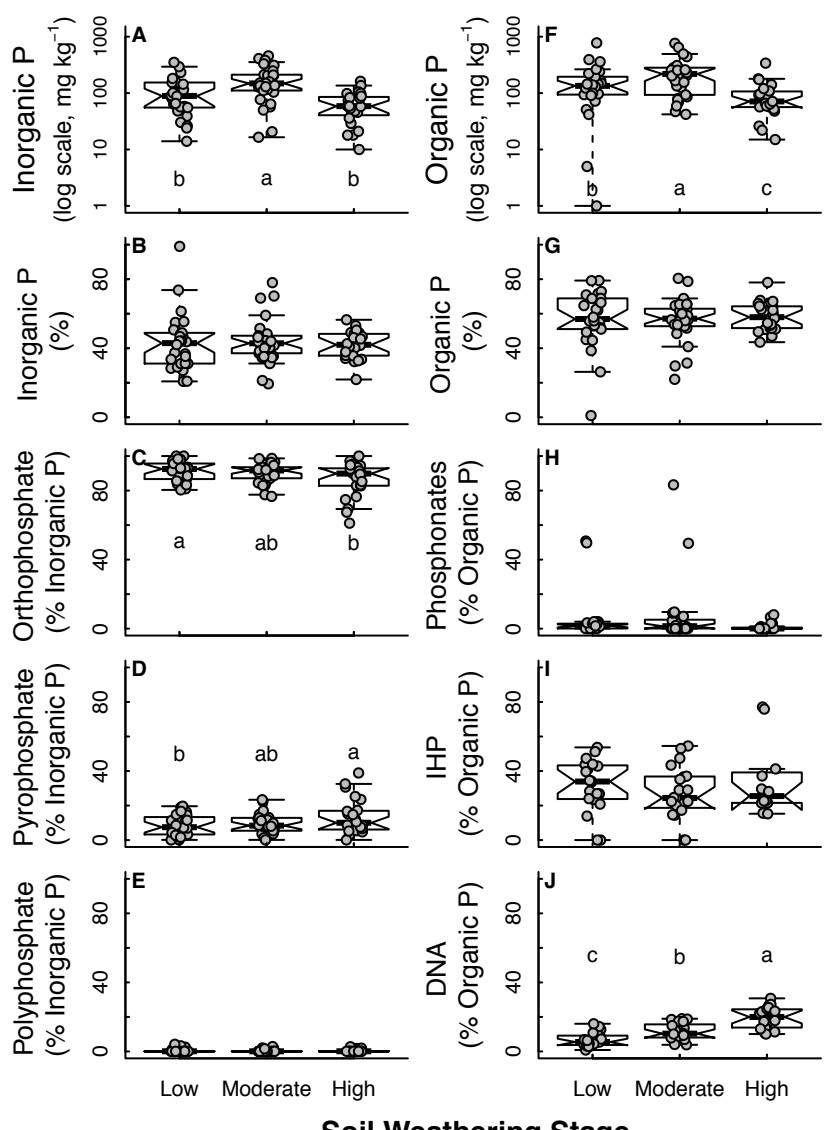

**Figure 5: Soil inorganic and organic phosphorus (P) composition in NaOH-EDTA extract as influenced by weathering stages on natural ecosystems.**

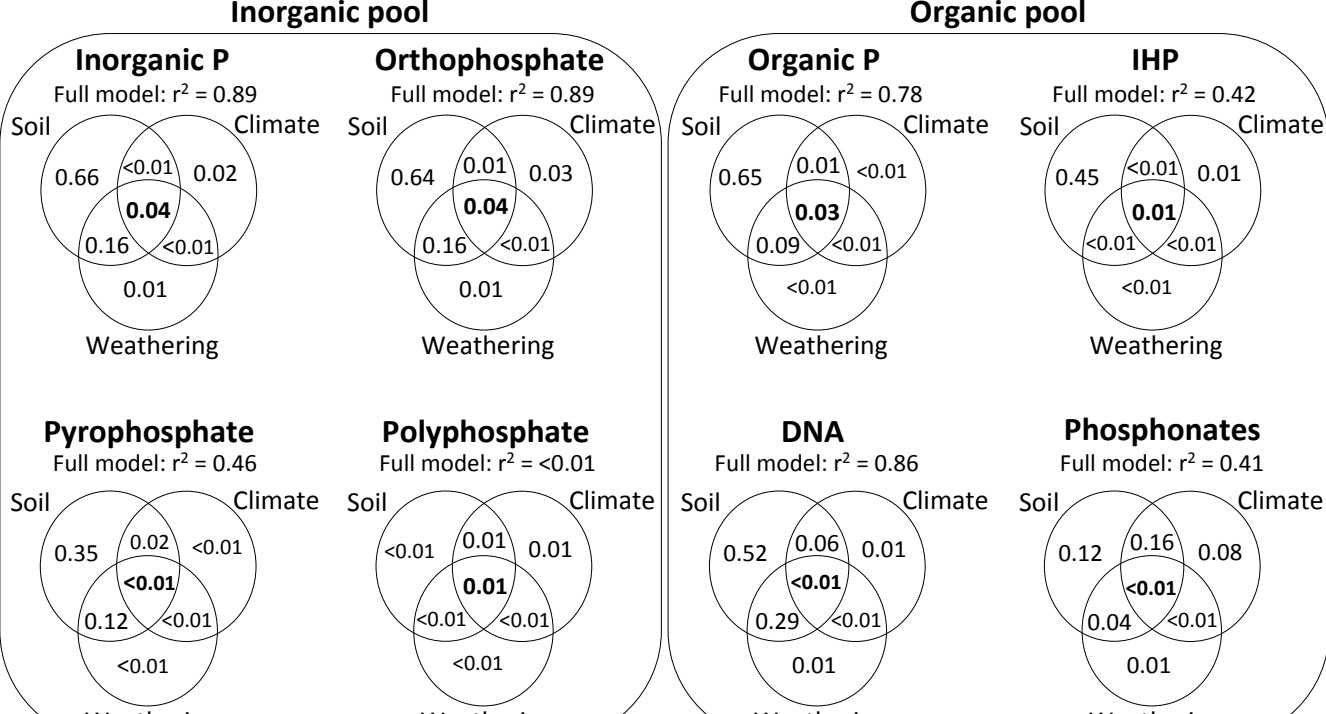

**Figure 6: Variation partitioning among edaphic, climatic, and weathering stages on soil inorganic and organic P composition in NaOH-EDTA extract on natural ecosystems. Soil organic and inorganic P forms and compounds were in mg kg$^{-1}$, and the other variables followed units described on Figures 2 and 3.**

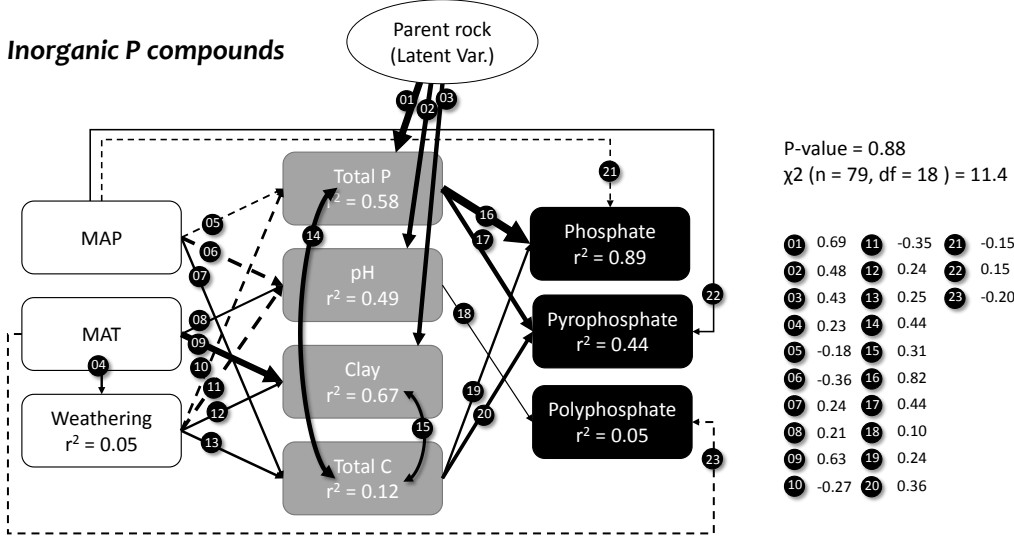

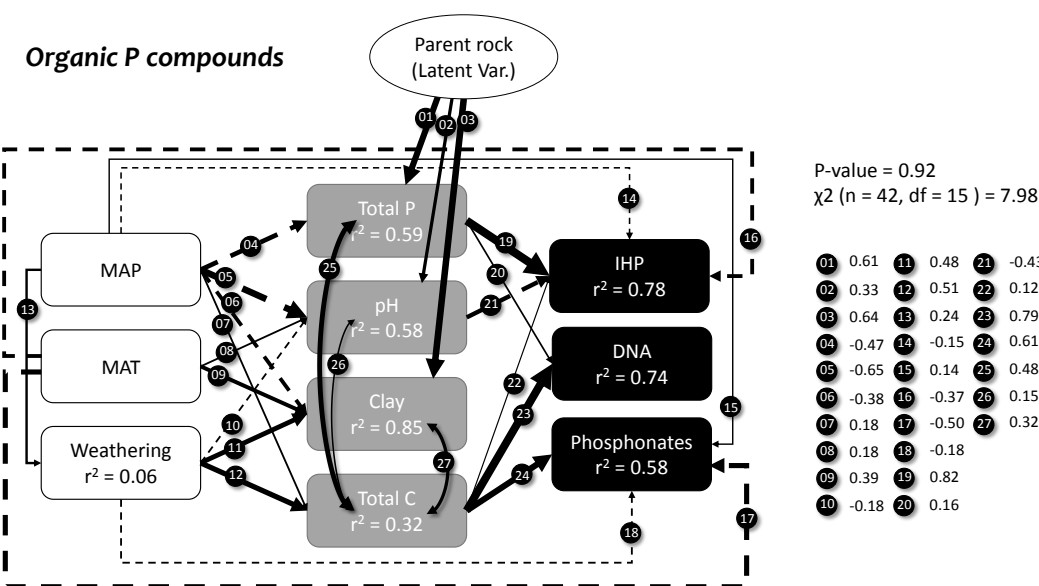

**Figure 7: Path analysis describing the direct and indirect effects of the main environmental predictors of soil inorganic and organic P compounds (mg kg$^{-1}$) in NaOH-EDTA extract as influenced by edaphic and climatic drivers on natural ecosystems. Solid and dashed lines represent positive and negative relationships, respectively. Soil organic and inorganic P compounds were in mg kg$^{-1}$, and the other variables followed units described on Figures 2 and 3.**

