# Peer review of "Environmental drivers of soil phosphorus composition in natural ecosystems"

_Biogeosciences, 2017_

## Referee Comment (RC1) · Anonymous Referee #1 · 8 Sep 2017

Organic phosphorus (P) cycling in soils is a topic that has received attention in recent years. As more papers are published, meta-analyses that link the data from these papers together to identify trends in organic P cycling become possible, at least in theory, and a paper presenting novel findings could be of interest to readers. However, deriving meaningful interpretations from a meta-analysis of soil P-NMR studies requires a clear understanding of the P-NMR method and its limitations, in order to correct for known for known artifacts of analysis. This was not done for this manuscript. As such, it cannot be published in its present form, and will require a major revision, including reanalysis of data, to make it publishable.

1. Writing quality: a) The quality of English in the manuscript is poor in many places. If the authors revise this manuscript, I suggest they have it read by someone more

familiar with English, who also understands the research field. b) Please check that you are using the correct spelling of the names of authors whose papers are cited. For example, "Vincent" is repeatedly cited as "Vicent", including in the supplemental files. c) Be specific with terminology. The term "P" is an abbreviation for the element for phosphorus. However, the authors use it interchangeably for phosphate, which is incorrect.

2. As P-NMR has become more widely used to characterize soil P forms, enough data has become available to indicate the possibility of using these data in meta-analyses to look at soil factors controlling P forms, especially organic P. However, those of us who use this technique the most also recognize its limitations. Although the use of P-NMR has advanced our understanding of soil organic P cycling more than almost any other method to date, the technique is not perfect. It is important to understand the artifacts of the method. It is also important to separate P-NMR results on a soil extract from the P forms that would have been present in the original soil sample prior to extraction. After all, isn't that the objective of a soil science study? Unfortunately, it isn't clear to me that the authors of this manuscript are familiar enough with the soil P-NMR technique to understand its limitations and address them. This has produced a study that clearly involved a lot of work by the authors, but which ultimately has not produced any new insights with respect to soil P. Some specific areas of concern are:

a) Concentration: It is not possible to determine absolute concentrations of P forms or compound classes using NMR; only relative percentages can be determined, because it is a compositional analysis in which the total must be 100%. Concentrations of P forms are then determined by multiplying by the total extracted P concentration by the percentage of each P form, which is still based on the compositional analysis. This is why the proportions and concentrations of total organic P and total inorganic P (Figs. 2 and 3) show inverse relationships to one another – together they have to add to 100%. This is exactly what would be expected, so it is strange to me that the authors would comment on this (p. 6, lines 13-16). The authors also do not seem

to understand the relationship between total P in the soils and P extraction in NaOH-EDTA. In natural (non-tilled) samples, P is stratified, such that concentrations are higher at the soil surface and lower with depth. There will also be an increase in organic P at the soil surface from inputs of plant material, which will decrease with depth – especially in forests with limited mixing and with greater fungal activity in mats in the forest floor (as is typical for temperate forests, where the majority of these studies were conducted). This needs to be accounted for somehow.

b) Extraction efficiency and soil pH: It has been very well established that the recovery of total P from soil samples with NaOH-EDTA extraction is never 100%, and is higher from samples with lower pH. The extraction seems to favor samples high in iron and aluminum, with generally poor P recovery from samples high in calcium; the reasons for this are unclear. As such, any meta-analysis comparing across a range of sample must take into account differences in P recovery among studies, and even among depths within the same soil profile or at different points along a soil chronosequence. For example, the recovery of total P in the samples for the Turner et al. (2003) paper ranged from 14-45%, in the Turner et al. (2007) paper 63-91%, and in the McDowell et al. (2007) paper 11-75%. If the purpose of this meta-analysis is to look at factors controlling soil P, then these differences in recovery must be factored in. Is it even possible to compare the results for a soil where only 11% of the total P was extracted to one with 91% extraction? What about the 89% of total P that wasn't extracted? The authors of this manuscript don't even mention this as a factor, let alone correct for it. And that, unfortunately, undermines their results.

c) Degradation: As noted, it is important for any soil study to ensure that the forms discussed, or the ratios of compound classes such as orthophosphate monoesters and diesters, are based on what was in the original soil sample, and not what was produced during extraction and analysis. It is well established that some orthophosphate diesters such as RNA and phospholipids can degrade to the orthophosphate monoesters $\alpha$- and $\beta$-glycerophosphates (phospholipids) and various monophosphates (RNA) when

analyzed at the high pH required for good peak separation in P-NMR spectra [e.g. Turner et al. 2003; Doolette et al. 2009; He et al. 2011, Vincent et al. 2013; Schneider et al. 2016. The degree of degradation will vary depending on the length of NMR experiment and other factors [see Cade-Menun and Liu (2014) and Cade-Menun (2015) for more details]. It is essential that these degradation peaks are identified and quantified in order to determine the correct concentrations of orthophosphate monoesters and diesters that were in the original soil sample; doing so improves any comparison of these P forms to other soil properties (e.g. Young et al., 2013; Liu et al. 2013 J. Environ. Qual. 42:1763-1770). Unfortunately, most studies before 2010 did not identify these compounds and correct for degradation. The authors of this manuscript acknowledge that degradation can occur (p. 4), but for some reason have chosen to ignore it, which is a major problem. The issue of degradation MUST be addressed for any study of edaphic and climatic characteristics to have any meaning. If the concentrations of orthophosphate monoesters and diesters were not corrected in the original study, then the authors of this manuscript could have applied some correction factor to compensate. For example, Vincent et al. (2013) note that most non-inositol phosphate monoesters were diester breakdown products (p. 160). The studies used by the authors here all included some measurement of inositol phosphates (at least myo-IHP and scyllo-IHP). As such, the authors could have assumed that those were the only true monoesters, and corrected the remaining proportion of monoesters to diesters. It would have at least been more meaningful that what they did, which was to ignore degradation but then reach the conclusion that the ratio of diesters to monoesters was a significant factor in the study.

3. Selection of studies: The authors indicate in the methods that they were careful in their selection of papers to include in their meta-analysis, such as native vegetation. As such, I am puzzled as to why the Turner et al. 2003 paper was included as the only study from the USA, because it used agricultural soils. And while the abstract and elsewhere in the text indicate a "dataset including 88 sites", these are overwhelmingly biased to sites in New Zealand (59) and Panama (21), which does not cover a range

of "temporal, edaphic and climatic characteristics". The sites selected are also mainly from chronosequence studies, which may also have affected the P forms and their relationship to soil properties.

4. Introduction:

a) Please include references for all statements of fact, and make sure those facts are correct. For example, p. 1, lines 24-25: "Once P has been dissolved as free orthophosphate" It isn't possible for free orthophosphate to exist in the soil solution; it will still be associated with cations, although as more soluble forms.

b) Be careful with terminology. Page 2, line 1: "inorganic and organic P pools are each composed by fractions or functional groups". No, they are composed of specific P compounds. The term "functional group" is used elsewhere in the introduction. Please indicate what is meant by this term, which isn't one used for soil P chemistry. And note that fractionation measures operationally-defined P pools, rather than specific P forms.

c) Page 2, line 10: Turner 2007 is not cited in the references.

5. Methods:

a) See comments above about site selection.

b) Page 4, lines 14-23: This discussion about degradation belongs in the Discussion section, not the methods section.

c) The authors have made a lot of assumptions here, particularly for soil classification. Please justify these assumptions in the Discussion section of the manuscript.

6. Results: a) I am puzzled by the phrase "concentration (% of total NaOH EDTA P)", page 6 line 30. Do you mean % or concentration in mg/kg? They are not the same thing, although they are derived from the same data (% of P forms multiplied by extract concentration).

b) As noted above, any results related to total concentrations or percentages of orthophosphate monoesters, orthophosphate diesters and the diester:monoester ratio are meaningless if not corrected for degradation. The authors must remove all reference to uncorrected concentrations and ratios. They could correct them as suggested above, or they could focus on specific P forms (e.g. DNA or IHP).

7. Discussion: Given the issues noted above, I am not sure there is anything meaningful in the discussion section, which as written is a review of the temporal, edaphic and climatic characteristics affecting P forms in NaOH-EDTA extracts, rather than in the original soils themselves. This is really unfortunate given the amount of work the authors put into this study. I hope the authors will address these issues. When they do, I expect much of the discussion section to change.

8. Figures:

a) The two figures used for Figure 1 were both published elsewhere, and thus are covered by copyright. However, the authors do not indicate anywhere that they have permission to use these figures in their manuscript, which must be obtained from the publishers of the original papers.

b) All figures containing references to total orthophosphate monoesters and diesters, and the diester:monoester ratio (e.g. 3, 5, 7, 8, 9, S4.1, S4.2, S4.3, S4.4) must be corrected for degradation. And all figures will likely change when the authors have normalized the data used in this study for P recovery.

---

## Referee Comment (RC2) · Anonymous Referee #2 · 20 Oct 2017

Phosphorus is an essential nutrient for life that limits productivity in many terrestrial ecosystems. However a big knowledge gap still needs to be filled concerning on P cycling dynamics in the soil, as a complex interaction of mineral, biological, and climate factors. This paper addresses this question analyzing a wide dataset, trying to extract general or global patterns that could help to complement the current knowledge of this important biogeochemical cycle. Despite the paper presents an interesting dataset and a significant goal, I could not recommend publication without major revisions. The writing should be improved, paying attention to shorten and simplify phrases. It also requires a better organization of the main ideas or messages the authors want to transmit. It is not clear the take-home message of the paper; especially the conclusions should be improved. I also have some important comments on the focus of the discus-

sion. However, the data compilation proves that behind this publication there is already a large work, for this reason I would encourage the authors to work to improve the manuscript.

General comments

- The writing of the paper needs to be improved. The paper is dense and hard to read like it is. - The main messages to take home are not clear, these must be highlighted. - There are to many bivariate graphs that distract to understand the main messages. I would suggest to add most of them for the supplementary material and keep in the main text the ones that are significant and are used to describe main processes in the text - The authors present the patterns shown as global, but there is no reference on the role of different biomes and plant communities, which are in turn related to soil properties. Ecological implications for the relations seen are missing. - Because of the distribution of the dataset, where most of the samples are from New Zealand, the authors should address the associated bias that the data could have. - The authors consider the weathering status as a temporal proxy (as it is said in the abstract) to be crossed with soil and climate properties. However, weathering status in this paper is defined by soil type, which makes this classification at certain point redundant with soil properties and climate. The authors should clarify this decision. - To assume organic C as total C is only acceptable in organic soils. This assumption can lead to large errors in calcareous soils. - Why the path analysis is used to explain exclusively diester/monoester ratio and not other P-form? Is this ratio providing specific information on nutrient state of the ecosystem? Is significant for understanding P-limitation or inorganic control over the P cycle? This should be argued. - I miss a clear explanation on the role of the basement/parent material.

Specific comments

- The last sentence of the abstract is not telling anything new "organic and inorganic P pools as well as their functional groups composition are determined by distinctive

drivers that regulate key ecological governing their presence..." - Pag 2, line 22, which 5 factors? - Pag 4, line 27, starts a list with "a) " but no more items are listed - Pag 7, line 25, the no effect of many climatic variables can be related to the geographic bias of the dataset. Should be argued. - Pag 8, line 10. Is obvious that poorly crystalline Fe and Al, do not correspond to weathering status if we consider the classification status than the authors have used. However, the presence of these oxides can deeply influence the P pools and cycles in Oxisols and Ultisols but also Andosols. - Pag 9, line 10. This is a too ambitious sentence. There is no information presented in this study about the variability among communities or different biomes. It is not explained neither how some edaphic variables depend on climate. - Precipitation and moisture index give similar bivariate relations, maybe with one of both variables would be enough.
* * *

---

## Author Comment (AC1) · 18 Nov 2017

We sincerely thank the editor and reviewers for evaluating our manuscript. We have responded to each comment. As requested by the editorial board, we are providing in this document responses to comments only, and not the revised manuscript, even though we already made several changes on it following the reviewers' suggestions, and specific details are presented on this document. Comments made by the reviewer are identified by "R1", and responses from authors are identified with "Response to R1".

R1 General comment.

Organic phosphorus (P) cycling in soils is a topic that has received attention in recent

years. As more papers are published, meta-analyses that link the data from these papers together to identify trends in organic P cycling become possible, at least in theory, and a paper presenting novel findings could be of interest to readers. However, deriving meaningful interpretations from a meta-analysis of soil P-NMR studies requires a clear understanding of the P-NMR method and its limitations, in order to correct for known artifacts of analysis. This was not done for this manuscript. As such, it cannot be published in its present form, and will require a major revision, including reanalysis of data, to make it publishable.

Response to R1 General comment.

We understand the point the reviewer is making about using the correction for potentially degraded peaks (of diesters converted to monoesters). Just to clarify, we did not use the correction previously because 39% of inositol phosphate (comprehending all tropical results and other locates) and 12% of DNA results were absent from the compiled data. We knew that correction was possible through adding to the total diesters concentration, the $\alpha$- and $\beta$-glycerophosphate concentrations (potentialy degraded peaks), but the reviewer also provided additional details that could improve our analysis. To address the issue, we will follow the reviewer suggestion. Using the available data, we will focus on specific organic P compounds (i.e. DNA and IHP) instead of its respective functional groups (diester and monoester). Given the huge proportions of potentially degraded peaks (non-inositol monoesters), and the uncertain about which compounds were present in this potentially degraded fraction, we choose to not to work with the corrected di-to-mono ratio, focusing on DNA and IHP compounds instead.

R1 Comment 1.

Writing quality: a) The quality of English in the manuscript is poor in many places. If the authors revise this manuscript, I suggest they have it read by someone more familiar with English, who also understands the research field. b) Please check that you are using the correct spelling of the names of authors whose papers are cited.

For example, "Vincent" is repeatedly cited as "Vicent", including in the supplemental files. c) Be specific with terminology. The term "P" is an abbreviation for the element for phosphorus. However, the authors use it interchangeably for phosphate, which is incorrect.

Response to R1 Comment 1.

a) In the new manuscript version, a native language specialist will revise the English.

b) "Vicent" will be replaced by "Vincent", and we will check all the names of the other authors whose papers are cited.

c) In the new manuscript version, the terminology will be revised regarding the proper use of abbreviations. "P" will be used as an abbreviation for the element phosphorus, Po and Pi will be used for the respective organic and inorganic pools, and the other P compounds will be described by their proper names.

R1 Comment 2.

As P-NMR has become more widely used to characterize soil P forms, enough data has become available to indicate the possibility of using these data in meta-analyses to look at soil factors controlling P forms, especially organic P. However, those of us who use this technique the most also recognize its limitations. Although the use of P-NMR has advanced our understanding of soil organic P cycling more than almost any other method to date, the technique is not perfect. It is important to understand the artifacts of the method. It is also important to separate P-NMR results on a soil extract from the P forms that would have been present in the original soil sample prior to extraction. After all, isn't that the objective of a soil science study? Unfortunately, it isn't clear to me that the authors of this manuscript are familiar enough with the soil P-NMR technique to understand its limitations and address them. This has produced a study that clearly involved a lot of work by the authors, but which ultimately has not produced any new insights with respect to soil P.
Response to R1 Comment 2.

We recognize that P-NMR can have limitations, and we have addressed them in specific parts of the manuscript. We will emphasize those limitations according to the suggested comments. Regarding the separations of P-NMR results from other P forms present in the original soil sample, we worked with P-NMR results obtained from NaOH-EDTA extracts only (Y axis on figures 2, 3, 4, 5, and 7, which do not include the residual P, i.e. difference between soil total P and NaOH-EDTA P). The total P of NaOH-EDTA extracts could be obtained by adding Organic P (e.g. figure 3A) to Inorganic P (e.g. figure 2A), but it does not correspond to the soil total P. The total P (obtained with other method – not P-NMR, e.g. digestion) was also presented in the manuscript, but acknowledging that it was obtained by a different method. In the new manuscript version, we will add more information in the figure captions to avoid misunderstandings, i.e., results in the Y axis are from NaOH-EDTA P-NMR results.

R1 Comment 2a.

a) Concentration: It is not possible to determine absolute concentrations of P forms or compound classes using NMR; only relative percentages can be determined, because it is a compositional analysis in which the total must be 100%. Concentrations of P forms are then determined by multiplying by the total extracted P concentration by the percentage of each P form, which is still based on the compositional analysis. This is why the proportions and concentrations of total organic P and total inorganic P (Figs. 2 and 3) show inverse relationships to one another – together they have to add to 100%. This is exactly what would be expected, so it is strange to me that the authors would comment on this (p. 6, lines 13-16). The authors also do not seem to understand the relationship between total P in the soils and P extraction in NaOH-EDTA. In natural (non-tilled) samples, P is stratified, such that concentrations are higher at the soil surface and lower with depth. There will also be an increase in organic P at the soil surface from inputs of plant material, which will decrease with depth – especially in forests with limited mixing and with greater fungal activity in mats in the
forest floor (as is typical for temperate forests, where the majority of these studies were conducted). This needs to be accounted for somehow.

Response to R1 Comment 2a.

Our total organic and inorganic P results, on mg kg-1 basis, are from NaOH–EDTA extracts only (do not include residual P, i.e. difference between soil total P and total P of NaOH-EDTA extracts). Based on our understanding, the results on mg kg-1 basis were determined from the proportion (%) of each P compound or functional group on spectra (determined by integration of peaks area or deconvolution) multiplied by the total P extracted with NaOH–EDTA. Most authors have presented their P-NMR results (forms and compounds) on both % and mg kg-1 basis (from P-NMR results of NaOH–EDTA extracts), including most of the ones we compiled data from. In the new manuscript version, we will add more information on figure captions to state that results on Y axis are from NaOH–EDTA extracts only. Usually, P-NMR results from NaOH-EDTA soil extracts are presented in both ways: (a) on mg kg$-1$ basis (non-including residual P), and (b) relative distribution of P (%). We followed the same criteria used by those papers to present our results. We do understand that results are based on a compositional analysis (i.e. P forms are determined by multiplying the total P extracted with NaOH-EDTA by the percentage of each P form), but the description of the inverse relation (obviously a inverse relation) between organic and inorganic concentration (% of total NaOH EDTA P) meant to explore the phenomena of pH or other variable impacting these forms. It was the way we found to describe our results. In the new manuscript version, we will reformulate the text avoiding the obviousness on describing results from percentages. In the specific case, the sentence containing "they showed a contrasting behavior" will be excluded. We do understand that soil total P is different than soil P extracted with NaOH- EDTA. We have mentioned that on Page 7 lines 10-13 "It's important to note that the reported total P is the one obtained by digestion and usually comprise the residual P non-recovered by the NaOH EDTA extractant. The recovery of total P by NaOH EDTA extraction is variable depending on soil characteristics and

laboratory procedures (Cade-Menun and Liu, 2014)." Moreover, knowing that there is a potential effect of soil conditions and laboratory procedures, we used the P recovery (percentage of P extracted with NaOH EDTA from soil total P) as a random factor in all bivariate regression models. We agree that natural (non-tilled) samples have stratified nutrient distributions. Our supplemental Figure S2 presented the results obtained regarding this effect. But contrary to what was expected, we found no effect of sampling depth over organic P concentration in mg kg−1, neither for both organic and inorganic on % basis (even though functional groups of organic and inorganic P responded dynamically to soil depth, even having contrasting responses for organic and mineral soil layers). We did find a sampling depth effect for inorganic P concentration in mg kg−1. Therefore, knowing that there is a potential effect of sampling depth, we used it as a random factor in all bivariate regression models.

R1 Comment 2b.

b) Extraction efficiency and soil pH: It has been very well established that the recovery of total P from soil samples with NaOH-EDTA extraction is never 100%, and is higher from samples with lower pH. The extraction seems to favor samples high in iron and aluminum, with generally poor P recovery from samples high in calcium; the reasons for this are unclear. As such, any meta-analysis comparing across a range of sample must take into account differences in P recovery among studies, and even among depths within the same soil profile or at different points along a soil chronosequence. For example, the recovery of total P in the samples for the Turner et al. (2003) paper ranged from 14-45%, in the Turner et al. (2007) paper 63-91%, and in the McDowell et al. (2007) paper 11-75%. If the purpose of this meta-analysis is to look at factors controlling soil P, then these differences in recovery must be factored in. Is it even possible to compare the results for a soil where only 11% of the total P was extracted to one with 91% extraction? What about the 89% of total P that wasn't extracted? The authors of this manuscript don't even mention this as a factor, let alone correct for it. And that, unfortunately, undermines their results.

Response to R1 Comment 2b.

We do understand that soil chemical characteristics can impact the recovery of P with a NaOH-EDTA extraction. We also agree that an "analysis comparing across a range of sample must take into account differences in P recovery among studies, and even among depths within the same soil profile". We have already addressed that using the 1) P recovery, and 2) sampling depths as random factors (and also latitude for other purpose not directly associated with the comment) in the analysis (which are described in the methods section Page 5 lines 7-15). An example of the impact of a random factor is described in the Page 6 lines 18-20: "There was no pH effect over this inorganic compounds in the organic layer (even though there is an apparent trend, these relationships became non-significant after including sampling depth as random effect on models; Supplementary Appendix S2 shows the sampling depth effect over soil P composition)."

R1 Comment 2c.

c) Degradation: As noted, it is important for any soil study to ensure that the forms discussed, or the ratios of compound classes such as orthophosphate monoesters and diesters, are based on what was in the original soil sample, and not what was produced during extraction and analysis. It is well established that some orthophosphate diesters such as RNA and phospholipids can degrade to the orthophosphate monoesters $\alpha$- and $\beta$-glycerophosphates (phospholipids) and various monophosphates (RNA) when analyzed at the high pH required for good peak separation in P-NMR spectra [e.g. Turner et al. 2003; Doolette et al. 2009; He et al. 2011, Vincent et al. 2013; Schneider et al. 2016. The degree of degradation will vary depending on the length of NMR experiment and other factors [see Cade-Menun and Liu (2014) and Cade-Menun (2015) for more details]. It is essential that these degradation peaks are identified and quantified in order to determine the correct concentrations of orthophosphate monoesters and diesters that were in the original soil sample; doing so improves any comparison of these P forms to other soil properties (e.g. Young et al., 2013; Liu et al. 2013

J. Environ. Qual. 42:1763-1770). Unfortunately, most studies before 2010 did not identify these compounds and correct for degradation. The authors of this manuscript acknowledge that degradation can occur (p. 4), but for some reason have chosen to ignore it, which is a major problem. The issue of degradation MUST be addressed for any study of edaphic and climatic characteristics to have any meaning. If the concentrations of orthophosphate monoesters and diesters were not corrected in the original study, then the authors of this manuscript could have applied some correction factor to compensate. For example, Vincent et al. (2013) note that most non-inositol phosphate monoesters were diester breakdown products (p. 160). The studies used by the authors here all included some measurement of inositol phosphates (at least myo-IHP and scyllo-IHP). As such, the authors could have assumed that those were the only true monoesters, and corrected the remaining proportion of monoesters to diesters. It would have at least been more meaningful that what they did, which was to ignore degradation but then reach the conclusion that the ratio of diesters to monoesters was a significant factor in the study.

Response to R1 Comment 2c.

We understand and agree with the reviewer's comment. But, as described in the methods section Page 4 lines 14-16: "We know that it is possible to correct degraded peaks of diesters converted to monoesters (e.g., Young et al., 2013 and Cade-Menun et al., 2010), but since some papers only showed functional groups like monoesters and diesters, and not species (specific P compounds) inside these functional groups, this correction was not done." Not all studies used in this manuscript included some measurement of inositol phosphates (at least myo-IHP and scyllo-IHP). Specifically, the following papers did not present P species (including myo-IHP and scyllo-IHP) inside these functional groups (monoesters and diesters) are: Celi et al., 2013, n=4; Vincent et al. 2010, n=1; Turner, 2008b, n=1; Turner et al 2003 (native soil sample), n=1; Turner and Engelbrecht, 2011, n=19; Turner et al 2014, n=10; and therefore correction was not possible to be addressed properly based on our previous knowledge. Some

of these authors acknowledge that there is a small contribution of inositol phosphates (most tropical soils) while others have provided no explanation about with they did not present specific P compounds results. Therefore, we thought it will still be biased to assume something that we were not certain of (i.e. amount of inositol phosphates). As described earlier, to address the issue, we will follow the reviewer suggestion. Using the available data, we will focus on specific organic P compounds (i.e. DNA and IHP) instead of its respective functional groups (diester and monoester). According to the gathered data, non-inositol monoesters (potentially degraded peaks, as suggested by the reviewer) corresponded to 66.76 % in average of the total amount of non-corrected monoesters (ranging from 7.8 to 100%), previously reported as total monoesters content, from papers that presented IHP results (n=61). The same non-inositol monoesters (potentially degraded peaks) corresponded to 53.94 % in average of the total NaOH EDTA organic P amount (ranging from 6.47 to 100%) from papers that presented IHP results (n=61). Based of the results presented by the authors we could not calculate how much of the potentially degraded peaks were: $\alpha$- and $\beta$-glycerophosphate (Doolette et al., 2009), nor RNA and phospholipid (which includes glycerophosphates) (Vincent et al., 2013); which were determined as degraded peaks by those authors. Therefore, given the proportions, correcting for potentially degraded peaks has a huge impact on the results, and it is a not completely unbiased calculation, since we don't know if all potentially degraded peaks were $\alpha$- and $\beta$-glycerophosphate (Doolette et al., 2009), or RNA and phospholipid (Vincent et al., 2013), so we choose to not work with the di-to-mono ratio. Inositol plus DNA represented 59.20% in average of total NaOH EDTA organic P (n=51) from papers that presented both DNA and IHP results. Therefore, it is also a huge proportion and could be an unbiased approach for those results. The reported proportions are not closing exactly due to the different datasets (n = 51 and n=61). To re-analyze data, IHP will not be considered for tropical soil results because they have non-detected concentrations of this compound (but tropical results will be maintained for the other variables). The following two paragraphs were written just to clarify why we have done the analysis in the previous way. We tried to be as

clear as possible about this issue, as it is written in the Page 4 lines 18-20: "We expect for future researches to provide results of as much soil P species they can find rather than functional groups only, even when species concentrations are low (and describe when species are not detected), what may enable future analysis to avoid possible confounding effects of organic P species inside functional groups (e.g., inositol and monoesters)." So, we believe that some questions will still remain to be addressed regarding soil P composition in terrestrial natural ecosystems, but our manuscript will provide significant and robust information using currently available results from literature. We understand the importance of what the reviewer is asking for, and recognize that in the manuscript, but as described we could not reach that level of detail due to absence of data (all specific P compounds). We have used an approach used by other authors. The same approach of not correcting for potentially degraded peaks was used in another recent paper, for example, that combined results from pasture soils using P-NMR results of NaOH–EDTA extracts (Nash et al., 2014). Essentially, they did not corrected for any degraded peak to determine the diester-to-monoester ratio, and described that this was out of their scope, but we agree that their approach is also not optimal.

R1 Comment 3.

Selection of studies: The authors indicate in the methods that they were careful in their selection of papers to include in their meta-analysis, such as native vegetation. As such, I am puzzled as to why the Turner et al. 2003 paper was included as the only study from the USA, because it used agricultural soils. And while the abstract and elsewhere in the text indicate a "dataset including 88 sites", these are overwhelmingly biased to sites in New Zealand (59) and Panama (21), which does not cover a range of "temporal, edaphic and climatic characteristics". The sites selected are also mainly from chronosequence studies, which may also have affected the P forms and their relationship to soil properties.

Response to R1 Comment 3.

The Turner et al. 2003 paper included most soils under arable cropping, although there was a native site, and this was the one we included in our analysis. We understand that we were not able to cover a vast representative sample, at global level, but we included as much as we could, given the data availability on the literature. This compilation made this study to have the wider geographical coverage on the topic (terrestrial environments with native vegetation - P-NMR results of NaOH–EDTA extracts).

R1 Comment 4.

Introduction: a) Please include references for all statements of fact, and make sure those facts are correct. For example, p. 1, lines 24-25: "Once P has been dissolved as free orthophosphate" It isn't possible for free orthophosphate to exist in the soil solution; it will still be associated with cations, although as more soluble forms. b) Be careful with terminology. Page 2, line 1: "inorganic and organic P pools are each composed by fractions or functional groups". No, they are composed of specific P compounds. The term "functional group" is used elsewhere in the introduction. Please indicate what is meant by this term, which isn't one used for soil P chemistry. And note that fractionation measures operationally-defined P pools, rather than specific P forms. c) Page 2, line 10: Turner 2007 is not cited in the references.

Response to R1 Comment 4.

a) In the new manuscript version, all statements of fact will be referenced, and it was make sure that those facts were correct. Specifically, "as free orthophosphate" will be excluded from the sentence. In other occurrence we will use "available" instead of "free" when referring to P that could be potentially taken up by plants.

b) In the new manuscript version, the statement will be reviewed clarifying that inorganic and organic P pools are composed of specific P compounds. "Functional groups" were changed to compounds in the whole manuscript when describing P compounds.

c) It will be corrected in the new manuscript version. The correction is "Turner et al.,

2007", which was previously cited in other parts of the manuscript.

R1 Comment 5.

Methods: a) See comments above about site selection. b) Page 4, lines 14-23: This discussion about degradation belongs in the Discussion section, not the methods section. c) The authors have made a lot of assumptions here, particularly for soil classification. Please justify these assumptions in the Discussion section of the manuscript.

Response to R1 Comment 5.

a) It was answered on Authors' response to comment 3.

b) In the new manuscript version, we will move the part about degradation to the Discussion section.

c) In the new manuscript version, the assumptions about soil classification will be justified in the discussion section. The assumptions include: The soil total P content depends on both weathering stages and parent material, but generally decreases with increasingly weathered soil orders (Yang and Post, 2011). The soil weathering stages classification also takes into account changes in soil P composition, and generally follows the Walker and Syers (1976) conceptual model: there is a gradual decrease and eventual depletion of primary mineral P (mainly apatite P), a decrease of total P, an increase and then decrease of total organic P, and a increase and eventual dominance of occluded P during soil development (Yang and Post, 2011). In highly weathered soils, occluded P increases at the expense of organic P through by encapsulation of mineralized P inside of Fe and Al minerals (Crews et al., 1995).

R1 Comment 6.

Results: a) I am puzzled by the phrase "concentration (% of total NaOH EDTA P)", page 6 line 30. Do you mean % or concentration in mg/kg? They are not the same thing, although they are derived from the same data (% of P forms multiplied by extract concentration). b) As noted above, any results related to total concentrations

or percentages of orthophosphate monoesters, orthophosphate diesters and the diester:monoester ratio are meaningless if not corrected for degradation. The authors must remove all reference to uncorrected concentrations and ratios. They could correct them as suggested above, or they could focus on specific P forms (e.g. DNA or IHP).

Response to R1 Comment 6.

a) We meant % of NaOH EDTA P in %.

b) We consider to be the same response of "Authors' response to comment 2c)". We did not correct them for degradation in the previous manuscript version. As described earlier, to address the issue, we will follow the reviewer suggestion. Using the available data, we will focus on specific organic P compounds (i.e. DNA and IHP) instead of its respective functional groups (diester and monoester).

R1 Comment 7.

Discussion: Given the issues noted above, I am not sure there is anything meaningful in the discussion section, which as written is a review of the temporal, edaphic and climatic characteristics affecting P forms in NaOH-EDTA extracts, rather than in the original soils themselves. This is really unfortunate given the amount of work the authors put into this study. I hope the authors will address these issues. When they do, I expect much of the discussion section to change.

Response to R1 Comment 7.

As described above, we will focus on specific organic P compounds (i.e. DNA and IHP). Specifically, we deleted discussion about the mechanisms that prompted the inverse response of monoesters and diesters as P limitation increased (since those functional groups results were excluded from the manuscript). Discussion was added about why DNA concentration increased as both P limitation and soil acidity increased in older, more weathered soil systems. Discussion was also added about the increase in inositol

phosphates concentrations at more acidic soil environments.

R1 Comment 8.

Figures: a) The two figures used for Figure 1 were both published elsewhere, and thus are covered by copyright. However, the authors do not indicate anywhere that they have permission to use these figures in their manuscript, which must be obtained from the publishers of the original papers. b) All figures containing references to total orthophosphate monoesters and diesters, and the diester:monoester ratio (e.g. 3, 5, 7, 8, 9, S4.1, S4.2, S4.3, S4.4) must be corrected for degradation. And all figures will likely change when the authors have normalized the data used in this study for P recovery.

Response to R1 Comment 8.

a) In the new manuscript version, we will provide the coverage by copyright. License Numbers: 4210920836823 (Elsevier) and 4210930550479 (John Wiley and Sons).

b) The response about the correction for degradation is on "Authors' response to comment 2c)", and regarding the normalization for P recovery is addressed on the "Authors' response to comment 2a)".

References.

Cade-Menun, B., and Liu, C.W.: Solution phosphorus-31 nuclear magnetic resonance spectroscopy of soils from 2005 to 2013: A review of sample preparation and experimental parameters, Soil Sci. Soc. Am. J., 78, 19–37, doi:10.2136/sssaj2013.05.0187dgs, 2014.

Cade -Menun, B.J., Carter, M.R., James, D.C., and Liu, C.W.: Phosphorus forms and chemistry in the soil profile under long-term conservation tillage: a phosphorus-31 nuclear magnetic resonance study, J. Environ. Qual., 39, 1647–1656, doi : doi:10.2134/jeq2009.0491, 2010.

Celi, L., Cerli, C., Turner, B.L., Santoni, S., and Bonifacio, E.: Biogeochemical cycling of soil phosphorus during natural revegetation of Pinus sylvestris on disused sand quarries in Northwestern Russia, Plant Soil, 367, 121–134, doi : 10.1007/s11104-013-1627-y, 2013.

Crews, T., Kitayama, K., Fownes, J., Riley, R., Herbert, D., Mueller-Dombois, D., and Vitousek, P.: Changes in soil phosphorus fractions and ecosystem dynamics across a long chronosequence in Hawaii, Ecology, 76, 1407–1424, doi:10.2307/1938144, 1995.

Doolette, A. L., Smernik, R.J., and Dougherty, W.J.: Spiking Improved Solution Phosphorus-31 Nuclear Magnetic Resonance Identification of Soil Phosphorus Compounds, Soil Sci. Soc. Am. J., 73, 919-927, doi: 10.2136/sssaj2008.0192, 2009.

Nash, D.M., Haygarth, P.M., Turner, B.L., Condron, L.M., McDowell, R.W., Richardson, A.E., Watkins, M., and Heaven, M.W.: Using organic phosphorus to sustain pasture productivity: A perspective, Geoderma, 221-222, 11–19, doi: 10.2136/sssaj2013.2105.0187dgs, 2014.

Turner, B.L.: Soil organic phosphorus in tropical forests: An assessment of the NaOH-EDTA extraction procedure for quantitative analysis by solution 31P NMR spectroscopy, Eur. J. Soil Sci., 59, 453–466, doi: 10.1111/j.1365-2389.2007.00994.x, 2008b.

Turner, B.L., Wells, A., and Condron, L.M.: Soil organic phosphorus transformations along a coastal dune chronosequence under New Zealand temperate rain forest, Biogeochemistry, 121, 595–611, doi: 10.1007/s10533-014-0025-8, 2014.

Turner, B.L. and Engelbrecht, B.M.J.: Soil organic phosphorus in lowland tropical rain forests, Biogeochemistry, 103, 297–315, doi: 10.1007/s10533-010- 9466-x., 2011.

Turner, B.L., Condron, L.M., Richardson, S.J., Peltzer, D. A., and Allison, V.J.: Soil organic phosphorus transformations during pedogenesis, Ecosystems, 10, 1166–1181, doi: 10.1007/s10021-007-9086-z, 2007.

Turner, B.L., Cade-Menun, B.J., and Westermann, D.T.: Organic Phosphorus Composition and Potential Bioavailability in Semi-Arid Arable Soils of the Western United States, Soil Sci. Soc. Am. J., 67, 1168-1179, doi: 10.2136/sssaj2003.1168, 2003.

Vincent, A.G., Vestergren, J., Gröbner, G., Persson, P., Schleucher, J., and Giesler, R.: Soil organic phosphorus transformations in a boreal forest chronosequence, Plant Soil, 367, 149–162, doi: 10.1007/s11104-013-1731-z, 2013.

Vincent, A.G., Turner, B.L., and Tanner, E.V.J.: Soil organic phosphorus dynamics following perturbation of litter cycling in a tropical moist forest, Eur. J. Soil Sci., 61, 48–57, doi: 10.1111/j.1365-2389.2009.01200.x, 2010.

Young, E.O., Ross, S.S., Cade-Menun, B.J., and Liu, C.W.: Phosphorus speciation in riparian soils: a phosphorus-31 nuclear magnetic resonance spectroscopy and enzyme hydrolysis study, Soil Sci. Soc. Am. J., 77, 1636–1647, doi: 10.2136/sssaj2012.0313, 2013.

Yang, X. and Post, W.M.: Phosphorus transformations as a function of pedogenesis: A synthesis of soil phosphorus data using Hedley fractionation method, Biogeosciences, 8, 2907-2916, doi: 10.5194/bg-8-2907-2011, 2011.

Walker, T. and Syers, J.: The fate of phosphorus during pedogenesis, Geoderma, 15, 1–19, doi: 10.1016/0016-7061(76)90066-5, 1976.

―――――――――――――――――――

---

## Author Comment (AC2) · 18 Nov 2017

We sincerely thank the editor and reviewers for evaluating our manuscript. We have responded to each comment. As requested by the editorial board, we are providing in this document responses to comments only, and not the revised manuscript, even though we already made several changes on it following the reviewers' suggestions, and specific details are presented on this document. Comments made by the reviewer are identified by "R2", and responses from authors are identified with "Response to R2".

R2 General comments.

- The writing of the paper needs to be improved. The paper is dense and hard to read

like it is.

- The main messages to take home are not clear, these must be highlighted.

- There are to many bivariate graphs that distract to understand the main messages. I would suggest to add most of them for the supplementary material and keep in the main text the ones that are significant and are used to describe main processes in the text.

- The authors present the patterns shown as global, but there is no reference on the role of different biomes and plant communities, which are in turn related to soil properties. Ecological implications for the relations seen are missing.

- Because of the distribution of the dataset, where most of the samples are from New Zealand, the authors should address the associated bias that the data could have.

- The authors consider the weathering status as a temporal proxy (as it is said in the abstract) to be crossed with soil and climate properties. However, weathering status in this paper is defined by soil type, which makes this classification at certain point redundant with soil properties and climate. The authors should clarify this decision.

- To assume organic C as total C is only acceptable in organic soils. This assumption can lead to large errors in calcareous soils.

- Why the path analysis is used to explain exclusively diester/monoester ratio and not other P-form? Is this ratio providing specific information on nutrient state of the ecosystem? Is significant for understanding P-limitation or inorganic control over the P cycle? This should be argued.

- I miss a clear explanation on the role of the basement/parent material.

Response to R2 General comments.

- The writing of the paper will be improved making it less dense and easier to read. A native language specialist will revise it.

- We believe that the main messages were responses related to the increasing complexity of phosphorus compounds as pedogenesis progress; therefore, we will emphasize those aspects across the manuscript to make more clear the take home messages.

- We agree that we have too many bivariate graphs, so we reduced one variable in Figure 3 (phosphonate, adding it as supplementary material), and added both climatic figures as supplementary material (organic and inorganic P vs. climatic drivers, Figures 4 and 5).

- Ecological implications of different biomes and plant communities over soil P composition will be added to the manuscript discussion. This will be done through determining the main biomes comprised in our data set using the annual precipitation – temperature diagram (whittaker diagram).

- The potential associated bias that the data could have because most of the samples were from New Zealand will be added to the manuscript. We will relate this discussion with the soil orders and biomes comprised.

- Regarding the redundancy between soil properties and climate with weathering , this is one reason why we used the path analysis. We can control for the redundancy using such statistical analysis. The figure of soil weathering stage relationship with soil age also showed that our classification followed the patterns that could be expected between soil weathering and soil age and between soil weathering and soil C. However, we do not have enough data to base our weathering classification only based on soil age.

-We checked all data from soils with pH higher than 7 and they have measured organic C, instead of total C. We added this description in the Material and Methods section to clarify this issue for readers.

- The diester to monoester ratio is an important soil metric that may be affected by soil and climatic factors (Nash et al. 2014). Ratio factors are important in our analyses

for several reasons (we will add this information to make it clearer to readers.): a) In general, they are more robust variable to compare P dynamics between soils as they minimize methodological between-study differences. b) Their variation is constrained compared with other continuous variables. As such, statistical assumptions are more easily considered. c) Last but not least, assumptions on their dynamics with the factor time are well-defined in literature. d) It is a key variable that give directions on how soil organic P composition is responding to environmental conditions. e) We agree that it is not explaining the inorganic P composition, but to fill this gap we have determined the complex inorganic P (pyrophosphate + polyphosphate) to orthophosphate ratio.

- We added more explanation on the role of the parent material over soil P composition and implications related to the soil weathering stages classification. The soil total P content depends on both weathering stages and parent material, but generally decreases with increasingly weathered soil orders (Yang and Post, 2011). The soil weathering stages classification also takes into account changes in soil P composition, and generally follows the Walker and Syers (1976) conceptual model: there is a gradual decrease and eventual depletion of primary mineral P (mainly apatite P), a decrease of total P, an increase and then decrease of total organic P, and a increase and eventual dominance of occluded P during soil development (Yang and Post, 2011). In highly weathered soils, occluded P increases at the expense of organic P through by encapsulation of mineralized P inside of Fe and Al minerals (Crews et al., 1995).

R2 Specific comments.

- The last sentence of the abstract is not telling anything new "organic and inorganic P pools as well as their functional groups composition are determined by distinctive drivers that regulate key ecological governing their presence. . ."

- Pag 2, line 22, which 5 factors?

- Pag 4, line 27, starts a list with "a) " but no more items are listed.

- Pag 7, line 25, the no effect of many climatic variables can be related to the geographic bias of the dataset. Should be argued.

- Pag 8, line 10. Is obvious that poorly crystalline Fe and Al, do not correspond to weathering status if we consider the classification status than the authors have used. However, the presence of these oxides can deeply influence the P pools and cycles in Oxisols and Ultisols but also Andosols.

- Pag 9, line 10. This is a too ambitious sentence. There is no information presented in this study about the variability among communities or different biomes. It is not explained neither how some edaphic variables depend on climate.

- Precipitation and moisture index give similar bivariate relations, maybe with one of both variables would be enough.

Response to R2 Specific comments.

- The last sentence of the abstract was modified to be more meaningful. It was changed to "We conclude that soil P composition is determined by edaphic and climatic drivers that regulate key ecological processes on terrestrial natural ecosystems. These processes are related to the source of P inputs, primarily determined by parent material and soil forming factors, and after altogether with plants and microbes coexistence, the bio-physico-chemical properties governing soil phosphatase activity, soil solid surface specific reactivity and P losses through leaching, and the P persistence induced by increasing complexity of P organic and inorganic compounds as pedogenesis evolve.

- Pag 2, line 22. We added the five state factors determining soil weathering.

- Pag 4, line 27. The "a)" was remaining from a previous version. It was excluded in the current version.

- Pag 7, line 25. The potential bias promoted by the geographic concentration of the studied sites was added to the manuscript.

[Figure]

- Pag 8, line 10. We added this explanation to the manuscript.

- Pag 9, line 10. We attenuated the sentence following the reviewer suggestion. "Unprecedented" was changed to "wide". Moreover, as described in the Response to R2 General comments we will improve discussion about what are the biomes represented in our dataset (whittaker diagram), and will add this information on that specific part of discussion. How edaphic variables depend on climate is being explored on the path analysis, but we will improve discussion on theoretical aspects of those relationships.

- Given the similarity between precipitation and the moisture index, the latter has been excluded from the manuscript.

References.

Crews, T., Kitayama, K., Fownes, J., Riley, R., Herbert, D., Mueller-Dombois, D., and Vitousek, P.: Changes in soil phosphorus fractions and ecosystem dynamics across a long chronosequence in Hawaii, Ecology, 76, 1407–1424, doi:10.2307/1938144, 1995.

Nash, D.M., Haygarth, P.M., Turner, B.L., Condron, L.M., McDowell, R.W., Richardson, A.E., Watkins, M., and Heaven, M.W.: Using organic phosphorus to sustain pasture productivity: A perspective, Geoderma, 221-222, 11–19, doi: 10.2136/sssaj2013.2105.0187dgs, 2014.

Yang, X. and Post, W.M.: Phosphorus transformations as a function of pedogenesis: A synthesis of soil phosphorus data using Hedley fractionation method, Biogeosciences, 8, 2907-2916, doi: 10.5194/bg-8-2907-2011, 2011.

Walker, T. and Syers, J.: The fate of phosphorus during pedogenesis, Geoderma, 15, 1–19, doi: 10.1016/0016-7061(76)90066-5, 1976.

---

## Referee Report (RR1)

**Biogeosciences**

**Manuscript bg-2017-307 Revised**

**Soil phosphorus dynamics on terrestrial natural ecosystems**

Leonardo Deiss, B. Anibal de Moraes and Vincent Maire

This is a revision of a previously submitted manuscript.

For the most part, the manuscript has been significantly improved compared to the original submission. In particular, the authors have constrained the studies used to avoid methodological differences, and have focused on specific P compounds to minimize differences among research groups (e.g. with respect to correcting for degradation), which could hamper the type of global comparisons they are trying to make. I still think the studies they have included are too heavily skewed to studies by a single author (Turner) and a single location (New Zealand). However, that is partially due to issues with analyses by some studies by other groups or from other locations. Hopefully, future studies will avoid these problems, allowing this type of analysis to be expanded in time. But for now, this is an interesting first attempt at this type of analysis.

There are still some problems that will need to be addressed before this is publishable. The majority of these fall more into minor or moderate revisions, as indicated below. However, the discussion will need some substantial revision, because many parts of it, as indicated below, are based merely on speculation, and are not grounded in fact or even in the authors' own results.

1.  Writing quality: for the most part, the quality of English has improved. However, there are still some problems with some sections. These are specifically addressed below, but I still recommend that the authors have the revised version read by a native English speaker who is familiar with the research topic. There are still problems with the writing in that the authors make sweeping statements of fact, without citing any references to support these statements. This is particularly true in the introduction, and specific instances are noted below.

2.  Title: The current title isn't very informative or very well written (e.g. "soil" by definition is "terrestrial", so including both is redundant). I suggest: "A meta-analysis of ecosystem properties influencing soil phosphorus dynamics in natural ecosystems".

3.  Abstract: Although the main body of the paper appears to have been revised with the assistance of a native English speaker, the abstract appears to have been overlooked.
    Line 7: "phosphorus (P) compounds can be modified by distinctive ecosystem properties" Using the term "modifies" implies that the ecosystem properties directly change P forms. However, most of the changes are indirect. For example, pH can change the soil environment to influence sorption of P forms, or it can affect the organisms that produce P forms, or that produce enzymes to mineralize P forms. But pH itself isn't necessarily changing P forms, except potentially alter the charge of the P form. As such, I think it would be best to change this to ""phosphorus (P) compounds can be influenced by distinctive ecosystem properties".
    Line 8: "soil P dynamics on terrestrial natural ecosystems, relating its organic " should be "soil P dynamics in natural ecosystems, relating organic "
    Line 11: "determined the soil P composition" should be "determined soil P composition"
    Lines 11-12: "nuclear magnetic resonance of soils extracted with NaOH EDTA" should be "nuclear magnetic resonance of NaOH-EDTA extracts of soils"
    Line 12: "models were used to better understand the soil P" should be "models were used to better understand the factors influencing soil P"
    Line 13: "relationships, soil P compounds had similar overall behaviors on mineral and organic layers but with different slopes". This analysis does not give any information about the behavior of soil P compounds; instead, it shows the relationship of soil P compounds with various factors. As such, this should be "relationships, trends for soil P compounds were similar for mineral and organic layers but with different slopes".
    Line 16: "Soil, particularly" should be "Soil factors, particularly"; "ratio" should be "ratios"
    Line 18: "soil P composition on terrestrial natural ecosystems" should be "soil P composition in natural ecosystems"
    Lines 19-20: "and after altogether with plant and microbe coexistence" I do not understand what the authors are trying to say here; something seems to be missing.

4.  Introduction:
    p. 2, line 4: "derives essentially" should be "derives predominantly"; "as an eolian deposit" should be "as eolian deposits". However, I disagree with this statement, because it is only relevant for younger soils (e.g. Chadwick et al. 1999 Nature 397:491-497). And because this is a statement of fact, a reference needs to be cited here.

p. 2, lines 5-6: "Phosphorus goes back to soil as organic materials (Noack et al. 2012" but not necessarily as organic P; Noack et al. showed that much of the P in plant residues was orthophosphate. I think the authors need to explicitly state that here, that "organic materials" are not primarily composed of organic P forms.

p. 2, lines 7-9: "Each new cycle…primary minerals" Please cite a reference to support this statement of fact, because I am not sure that it is in fact true.

p. 2, line 10: Jenny (1941) described these as the five factors of soil formation, not ecosystem functioning. Please cite another reference to support using this term in this way.

p. 2, line 14:"main Po compounds are" should be "main Po compound categories are", because monoesters, diesters, etc. are compound categories, not forms.

p. 2, lines 15-16: "orthophosphate monoester" should be "orthophosphate monoesters", "orthophosphate diester" should be "orthophosphate diesters" and "phosphonate" should be "phosphonates", because these broad compound classes contain a number of different P compounds.

p. 2, line 18: "used by" should be "directly taken up by", because all P compounds can be used by plants and microbes after transformation, just not directly taken up by them.

p. 2, lines 17-19: "Specific phosphatase…and microbes" and "Obviously…optimum" Please cite references to support these statements of fact.

p. 2, lines 19-20: "Phosphomonoesterase is more active in acidic soils while phosphodiesterase is optimized in basic soils (Turner and Haygarth 2005)" NO! It simply is not possible to make such a broad, sweeping statement based on a single study of pasture soils from England. Please rewrite.

p. 2, lines 28-29: "As soil ages…colimitation intermediate stage" Please cite a reference to support this statement of fact.

p. 3, line 4: "precipitations" should be "precipitation".

p. 3, lines 9-14: I think "parent rock" should be changed to "parent material", because soils can form on materials other than a specific type of rock (e.g. sand, glacial till).

p. 3, line 16: "Soil P composition has been studied in soils from ecosystems worldwide" should be "Soil P composition has been studied in ecosystems worldwide"

p. 3, line 17: "(NMR) was a widely used method" should be (NMR) is a widely-used method", because to the best of my knowledge it is still being used.

p. 3, line 19: "NaOH and chelating agent EDTA" should be "NaOH combined with the chelating agent EDTA"

p. 3, line 20 (and elsewhere throughout the manuscript): "NaOH EDTA" should be "NaOH-EDTA".

p. 3, lines 22-24: "The NaOH EDTA…(Turner and Blackwell 2013)" This sentence does not make any sense as written. How does this demonstrate that NaOH-EDTA is quantitative? Please rewrite and include more references.

p. 3, lines 26-27: "state factors of ecosystem functioning" see previous comment and rewrite.

p. 3, line 27: "because known responses were obtained from case-specific conditions". I do not understand what the authors are trying to say here. Please rewrite.

5. Methods:
p. 4, line 5: "that namely estimated an adequate delay time" should be "that used an adequate time"

p. 4, line 15: "only control (unchanged) was used" should be "only control (unchanged) samples were used"

p. 4, lines 23-23: "organic P and its compound" should be "organic P and the compounds"

p. 4, line 24: "NaOH EDTA inorganic P, and its compounds" should be "and NaOH-EDTA inorganic P and the compounds"

p. 4, line 27: "climate characteristic" should be "climate characteristics", because it refers to both precipitation and temperature.

p. 5, lines 29-30: "how the inorganic and organic P compounds variation" should be "how variations in inorganic and organic P compounds"

p. 6, line 4: "Differently" should be "In contrast"

6. Results:
p. 6, line 13: "McDowell and Steward" should be "McDowell and Stewart"; the name is spelled correctly in the References section.

p. 6, line 20: "And temperate rain forest" Don't some of the New Zealand sites fall into temperate rain forest? The authors should check to confirm.

p. 6, line 25: "were absented" should be "were absent"

p. 6, lines 26-27: "All compiled results…polyphosphate" This sentence repeats information already given (p. 4, lines 22-24), and should be deleted in one of these locations in the text.

p. 7, line 3: "The compounds proportions" should be "The proportioning of compounds"

p. 7, line 6: "Into the Pi pool" should be "In the Pi pool"; "layers, the orthophosphate" should be "layers, orthophosphate"

p. 7, lines 12-13: "There was no clay effect on both Pi and Po and their compounds proportions" should be "Clay had no effect on either Pi or Po, or on the proportions of compounds in these categories"

p. 7, line 18: "effect on both Pi and Po" should be "effect on either Pi or Po"

p. 7, line 24: "in both mineral and organic " should be "in either mineral or organic"

p. 7, lines 30-31: "In is important to note…usually comprises the residual P not recovered by the NaOH EDTA extractant". This sentence doesn't make sense to me. Firstly, the x axes in figures 2 and 3 vary depending on the factor (e.g. %, % organic P, log scale mg/kg). Secondly, total P doesn't just include residual P (which is what "comprises the residual P" means), it includes both the extracted P and the residual P. This sentence needs to be rewritten.

p. 8, line 2: "As the percentage" should be "As a percentage"

p. 8, line 12: "on the soil P composition on terrestrial natural ecosystems" should be " in the soil P composition in natural ecosystems"

p. 8, line 13: "over total " should be "on total"

p. 8, line 18 and line 19: "affected the soil age and CP ratio" should be "affected soil age and CP ratios"

p. 8, line 29: "of the ecosystem's properties" should be "of ecosystem properties"

p. 8, line 31: "and its compound" should be "and its compounds"

p. 9, line 2: "poorly defined" should be "poorly-defined"

p. 9, line 3: "of the ecosystem's properties" should be "of ecosystem properties"

p. 9, lines 7-8: "phosphonate had most of its variation explained by" should be "most of the variation in phosphonates was explained by"

p. 9, line 19: "by precipitation into the " should be "by precipitation in the"

p. 9, line 21: "Into the" should be "in the"

p. 9, line 26: "Pyrophosphate had a positive influence of" should be "Pyrophosphate was positively influenced by"

p. 9, line 27: "inositol was negatively" should be "inositol phosphates were negatively"

p. 9, line 29: "Phosphonate was" should be "Phosphonates were"

7. Discussion: Much of this is very poorly done, and relies on speculation rather than data. Major revisions are needed in this part of the manuscript.

p. 10, line 2: "compounds respond to" should be "compounds responded to"

p. 10, line 3: "at a wide" should be "on a wide"

p. 10, line 4: "terrestrial nature ecosystems" should be "natural ecosystems"

p. 10, line 7: "persistence on ecosystems" should be "persistence in ecosystems"

p. 10, lines 7-11. This is a very long, confusing sentence; as written, the point that the authors are trying to make with this sentence is not clear. It should be rewritten.

p. 10, line 14: "the decaying degree of C element is lower than the P" I do not understand what the authors are trying to say here.

p. 10, line 15: "resulting in the slowed decomposition of the older soil system" Again, I do not understand what the authors are trying to say here. As currently written, it indicates that the soil system itself is decomposing more slowly. However, soils do not decompose. Do they mean that decomposition is slower in older soil systems? If so, then decomposition of what? Plant material? Or do they mean mineralization of specific P forms? This needs to be rewritten for clarity.

p. 10, lines 16-19: "more weathered soils are remote from the parent material". Again, this is confusing as written. Do the authors mean physically remote? If so, wouldn't that depend on the nature of the soil weathering processes, and even on the soils themselves? Or do they mean "substantially changed"? Please rewrite.

p. 10, line 25: "occluded P increases at the expense of organic P". This sentence implies that organic P compounds cannot be occluded, which isn't the case – one of the reasons that inositol phosphates increase with weathering is that they become occluded in the same way that inorganic P compounds become occluded. The inaccuracy of this statement is due to citing a dated reference (Crews et al. 1995) that relied on Hedley fractionation. Please update this statement and cite something that uses P-NMR or other modern techniques.

p. 10, line 31 and p. 11, line 2: "phosphorus" should be abbreviated as P.

p. 11, line 2: "with species maximum" should be "with species' maximum"

p. 11, line 5: "shapes" should be "shaped", because it is discussing a student that has been published.

p. 11, line 9: "As soil aged" should be "As soils aged"

p. 11, line 11: "weathering stages had a major role" should be "weathering stage had a major influence"

p. 11, line 11: "dynamic" should be "dynamics"

p. 11, lines 16-23: It is well-established that ectomycorrhizal fungi convert the orthophosphate that they take up from the soil into polyphosphates, and translocate the polyphosphate along fungal hyphae, sometimes a great distance from where the orthophosphate is taken up (e.g. Bücking and Heyser 1999 Mycol. Res. 103:31-39; Plassard and Dell 2010 Tree Physiol. 30:1129-1139). This needs to be mentioned, as well as the need to define plant and microbial communities in studies of P forms.

p. 11, line 28 to p.12, line 2: Polyphosphates can potentially degrade to pyrophosphates during extraction and analysis by P-NMR (e.g. Cade-Menun et al. 2006, Environ. Sci. Technol. 40:7874-7880), so pyrophosphate and polyphosphate shouldn't be considered as fully distinct P forms, and the authors should use care when discussing hydrolysis of pyrophosphate separately from polyphosphate.

p. 12, lines 5-6: "Plant and microorganism breakdown diesters need" should be "Plant and microorganism breakdown of diesters needs"

p. 12, lines 8-9: "P limitation increased phosphoesterases synthesis as a way to increase the organic P breakdown to the bioavailable P". As written, it doesn't make sense. Do the authors want to say: "P limitation may stimulate increased phosphoesterase synthesis as a way to increase bioavailable P by the mineralization of organic P"?

p. 12, lines 12-16: "Therefore we hypothesize that as P got scarcer, plant and soil microorganisms may have been stimulated to produce phosphomonoesterases in greater amounts compared to phosphodiesterase because of the lower iinvestment required for the organic P acquisition. Even though acid phosphatases require greater activation energy than alkaline phosphatases (Hui et al. 2013), breaking down diesters would require both enzymes; therefore a greater investment in energy". These two sentences are baseless speculation, and should be deleted from the paper. The authors did not include any measures of phosphatase activities of any kind from any of the published studies, and thus are unqualified to say anything about how enzyme activity changes with soil factors. Their results in Fig. 3 show that the proportion of total organic P, as well as the proportions of IHP and DNA decrease with increasing pH. This is related to increased charge of these compounds with decreasing pH, and thus increased sorption. This is discussed in detail in Condron et al. 2005, among other papers.

p. 12, lines 18-20: As noted for the introduction, it isn't possible to take the results of one study into pasture soils from one country and extrapolate to a global model about enzyme activity, especially when the current study did not include any measures of enzyme activity. This needs to be deleted.

p. 12, lines 23-26: As noted above, the influence of pH on DNA sorption is well-established (Condron et al. 2005). The authors need to revise these lines of the discussion.

p. 12, lines 28 to p. 13, line 12: The authors need to be very precise in their terminology. In this section, they discuss "inositol phosphates". However, the data they include from various studies is for inositol hexaphosphates, which very specifically are inositols with 6 phosphate groups. These will behave very differently in soils from inositol phosphates with fewer phosphate groups. Please rewrite this section to be more precise. In addition, all of the processes discussed here to govern the behavior of "inositol phosphates" in soil will also apply to other P forms, especially DNA.

p. 13, lines 9-10: This sentence discusses changes in the concentration of "inositol phosphate" with soil weathering. However, the authors only present changes in "inositol phosphates" as proportions of organic P, not as concentrations. As such, this is all speculation and should be revised or deleted.

p. 13, line 14: "there were no inositol phosphates on tropical, more weathered soils". Could this not also relate to either inputs from plants or production of phytases by plants and microbes? The authors did not include any factors that might influence the cycling of IHP in these types of soils. As such, this is merely speculation and should be deleted.

p. 13, lines 20-23: "Recent investigations have contradicted the often-cited literature" please cite references here to support this statement of fact, both for the "often cited literature", and the "recent investigations".

p. 14, lines 10-30: One thing missing from this section is the role of vegetation with respect to inputs of different P forms. It should also be noted that a greater soil wetness is not necessarily associated with leaching, if there is impeded drainage of the soils in some ways (e.g. from the formation of placic horizons). This section of the manuscript is really vague, and doesn't add anything to our knowledge of P cycling in soils. The authors must do a far more thorough literature review to discuss their results properly.

p. 14, line 32: "As the orthophosphate percentage decreased following precipitation, the pyrophosphate percentage increased". This does not necessarily mean that the concentration of orthophosphate decreased, or that of pyrophosphate increased, or imply a cause-and-effect relationship. This line suggests to me that the authors do not understand compositional analysis: as the percent of one thing decreases the percentage of another thing will increase because the total must add to 100%. In my opinion, reporting and discussing these as percentages is misleading, for this reason. It would be far better to use concentrations.

p. 15, lines 6, 7: "phosphorus" should be "P"

p. 15, lines 7-8: "greater organic P concentrations were associated with increasing biomass production". Studies of biomass (e.g. Noack et al. 2012) show that the majority of P in plant biomass is as orthophosphate, not organic P compounds. As such, this sentence doesn't make sense to me, and isn't supported by the literature. Please revise.

8. Conclusions:
p. 15, lines 22-23: "after altogether with plant and microbe coexistence" as noted for the abstract, I do not understand what the authors are trying to say here. Something seems to be missing, or mistranslated.

9. Figures: In all figures, "phosphonate" should be "phosphonates", because this is a general compound category containing a number of specific P compounds. Also, "inositol" should be IHP or inositol hexakisphosphate.

10. References: there are many problems with this section of the manuscript. The authors need to carefully proof-read and correct this section.

a) Many of the references are out of order alphabetically, including the entire "Y" section (which should come after W), Deiss et al. 2017 should come after Damon et al. 2014; Laliberté et al. 2017 should come before Legendre and Legendre 2102 and Li et al. 2015; Whittaker 1975 should come before Wilson et al. 2013.
b) There are formatting differences among references. Cade-Menun et al. 2000 has the volume and page number listed as 30:1714-1725, while other references use a comma (e.g. Celi et al., 2013, 367, 121-134). Chen et al. 2004 does not include periods after abbreviations in the journal title (Aust J Soil Res); Cade-Menun and Preston 1996, Kizewski et al. 2011, Turner et al. 2013 have all the words capitalized in the manuscript title.
c) The journal volume number and/or page numbers are missing from Deiss et al. 2017, Doolette et al. 2016, Li et al. 2015, while George et al. 2017 is missing the name of the journal as well as the volume number.
d) Rumpel et al. is 2015 in the References, but 2016 in the text; Bünemann is "Bunemann" in the text (p. 11, line 16)

Condron, L.M., B.L. Turner and B.J. Cade-Menun. 2005. Chemistry and dynamics of soil organic phosphorus.  pp. 87-121.  In: J.T. Sims and A.N. Sharpley, eds. Phosphorus, Agriculture and the Environment.  Monograph no 46. Soil Science Society of America.  Madison, WI.

---

## Referee Report (RR2)

**Reviewer Comments**

The manuscript presents some interesting results and the results are now quite clearly presented. While some aspects have been improved in the revision, other aspects remain problematic, including some issues already pointed out by earlier reviewers. The four main issues I see are:

- Inadequate understanding / discussion of the limitations of the NMR method
- Precision and clarity of terminology
- A discussion section that does not go beyond what is already known
- English language

**Limitations of the NMR method**

Liquid $^{31}$P NMR is a powerful method to study the presence of certain P forms in soils, and it is of interest to combine results produced by different studies in a meta-analysis. However, it is absolutely necessary that the authors are aware and clearly and transparently communicate the limitations of liquid $^{31}$P NMR on NaOH-EDTA extracts. There are many other methods out there to study P cycling (Kruse et al. 2015), and only if results are correctly interpreted in light of the limitations of the methods can readers integrate knowledge gained from one method with knowledge gained from another method. Firstly, as already pointed out by reviewer 1 (first round), NaOH-EDTA extraction never extracts all soil P and often only a small portion of total soil P. Please state the fraction of total P extracted by NaOH-EDTA for all soils used in your analysis. Secondly, $^{31}$P NMR is a tool for P speciation, providing important information on P stocks, it does not, however, provide information on dynamics. Turnover, exchange kinetics, mineralisation rates, etc, which is what I understand under "P dynamics", can only be assessed using isotopic techniques (Frossard et al. 2011). Thirdly, liquid $^{31}$P NMR is not the preferred method for studying inorganic P species, since it cannot give information on predominant inorganic P species such as Fe-, Al- and Ca-phosphate. XANES is a more preferred method for looking at inorganic P species. This does not mean that the results on pyrophosphate, orthophosphate and polyphosphates concentrations are not useful, however, the paper reads as if these are the only inorganic P species of importance in soils. Please be more honest on the potentials and limitations of this approach.

These limitations need to be stated clearly and transparently in the introduction, since they provide the scope for the meta-analysis.

**Precision and clarity of terminology**

As mentioned above, the word "dynamics" is misleading for a study looking at P forms. I suggest changing the title to something more accurate, e.g. "The impact of soil, climatic, and temporal drivers on inorganic and organic P compounds".
An often-recurring term is "complex P compounds". From my understanding, "complex" is used to refer to high-molecular weight organic compounds of variable composition (McLaren et al. 2015). Please use a more precise term than "complex P compounds".

**Discussion section**

The discussion section remains the main weak spot of the article. As it stands, the discussion could be summarized by Fig 1 from the introduction. This is a problem because Fig 1 is a very generic and commonly used figure in the field. I suggest the authors follow Mensh and Kording (2017), who provide useful tips for structuring a discussion (Mensh and Kording 2017).

There follows a few ideas for rewriting the discussion that authors may use if they find them helpful. The first paragraph can more or less stay as is, it summarizes the results. The next two paragraphs could be dedicated to outlining the limitations of the analysis. Here it would

be worth pointing out that climate and weathering drive soil properties, so that it is not entirely appropriate to compare their influence on P forms, because much of the variation in soil properties can be explained by climate and weathering. Also, it might be worth discussing why variation in polyphosphates could not be explained by the models. The final two paragraphs could point out how this study adds to the literature. The focus of these paragraphs should not be comparisons to individual NMR studies, since they are anyways incorporated into the data, rather it would be interesting to weave the findings together with insights from studies using XANES, enzyme activities, or isotopes, to start painting a full picture.

**English language**
Authors stated that a native English speaker revised the manuscript. However, I don't think that the quality of the writing has improved from the previous version. There are still many grammatical issues and awkward writing, which make the manuscript difficult to read at times.

**References**
Frossard, E., D. L. Achat, S. M. Bernasconi, J.-c. Fardeau, J. Jansa, C. Morel, L. Randriamanantsoa, S. Sinaj, and A. Oberson. 2011. The use of tracers to investigate phosphate cycling in soil–plant systems. Pages 59-91 *in* E. K. Bünemann, editor. Springer, Heidelberg.

Kruse, J., M. Abraham, W. Amelung, C. Baum, R. Bol, O. Kühn, H. Lewandowski, J. Niederberger, Y. Oelmann, C. Rüger, J. Santner, M. Siebers, N. Siebers, M. Spohn, J. Vestergren, A. Vogts, and P. Leinweber. 2015. Innovative methods in soil phosphorus research: A review. Journal of Plant Nutrition and Soil Science **178**:43-88.

McLaren, T. I., R. J. Smernik, M. J. McLaughlin, T. M. McBeath, J. K. Kirby, R. J. Simpson, C. N. Guppy, A. L. Doolette, and A. E. Richardson. 2015. Complex Forms of Soil Organic Phosphorus-A Major Component of Soil Phosphorus. Environmental Science and Technology **49**:13238−13245.

Mensh, B., and K. Kording. 2017. Ten simple rules for structuring papers. PLoS Comput Biol **13**:e1005619.

---

## Referee Report (RR3)

**Biogeosciences**

**Manuscript bg-2017-307 Revision2**

**Soil phosphorus dynamics on terrestrial natural ecosystems**

Leonardo Deiss, B. Anibal de Moraes and Vincent Maire

This is a second revision of a previously submitted manuscript. I also reviewed the previous submissions.

For the most part, the manuscript has been significantly improved with each revision. However, there are still a few problems to be addressed, as noted below.

1.  Writing quality: for the most part, the quality of English has improved. However, there are still some problems in some places, which are noted below.

2.  I am puzzled about some of the literature cited in the introduction and discussion, and still think there is a bias to citations of specific authors rather than a good overview of the literature.
a)  Please cite original papers where possible, rather than recent review papers. For example, why cite Nash et al. (2014), rather than Newman and Tate 1980, Commun. Soil Sci. Plant Anal 11:835-842) or Tate and Newman (1982, Soil Biol. Biochem 14:191-196) for the description of P forms on p. 2, line 17? Why cite Turner 2008a for phosphatase and orthophosphate uptake (p. 2, line 20) – this was well-known prior to that paper. Why cite Huang et al. 2017 for vegetation and organisms (p. 10, l. 24) or mycorrhizae (p. 13, l. 16), when it presents no new information on either of these topics? The same is also true for Yu et al. 2013. There has been a lot of good research into many aspects of P cycling by a lot of good scientists; citing the review papers instead of the original research is unfair to the original researchers, and runs the risk of introducing errors into the literature if the original work was cited incorrectly in the review. Would the authors of this manuscript not prefer to have it cited directly?
b)  Please be careful citing studies using fractionation methods (e.g. Hedley) in study discussing specific chemical forms. Fractions from fractionation methods are operationally defined, not chemically defined, especially the Hedley fractionation method.

3.  Abstract:
    Lines 11-12: "nuclear magnetic resonance of NaOH-EDTA extracts" should be "nuclear magnetic resonance spectroscopy of NaOH-EDTA extracts" (something I missed in my previous review.
    Lines 16-17: "organic and inorganic P compounds variations" should be "variations in organic and inorganic P compounds"
    Lines 19-20: "and after altogether with plant and microbe" I still do not understand what the authors are trying to say here; something seems to be missing. Do they mean "and together with plant and microbe"?

4.  Introduction:
    p. 2, line 6 and elsewhere: "phosphorus" should be "P" after the first use of the abbreviation, except at the start of a sentence.
    p. 2, line 14: "composed by specific" should be "composed of specific"
    p 2, lines 17 and 18: please replace "Nash et al. 2014" and "Cade-Menun and Preston 1996" with more appropriate references, because these compound classes were recognized prior to the publication of these papers
    p. 2, line 20: "As most enzymes, the activity of soil P cycling enzymes" should be "As with most enzymes, the activities of soil phosphatases"
    p. 2, line 21: "specific enzyme optimum" should be "specific enzyme optima"
    p. 2, lines 24-25: DNA adsorption is only below 5 (the isoelectric point of DNA); also, why is a review paper (Yu et al. 2013) cited here, rather than one of the original references about pH? Please see the comments for page 12, below.
    p. 2, lines 27-31: studies using P fractionation (e.g. Walker and Syers 1976, Yang and Post 2011) do not give any information about $P_o$ and $P_i$ forms and compounds, because fractionation by definition can only give information about operationally-defined pools. Please rewrite this paragraph, and cite better references that actually describe changes in $P_i$ and $P_o$ compounds
    p. 3, line 6: "While this study" should be "While the Feng et al. (2016) study"
    p. 3, line 11: "Variation in" should be "Variations in"
    p. 3, lines 27-28: "the NaOH-EDTA extraction does not separate Fe-, Al-and Ca-phosphate compounds (Kizewski et al. 2011)" although this might be what Kizewski et al. said, it isn't very accurate. A better way to say it is: "the high pH of the NaOH-EDTA extraction separates P species from the cations (e.g. Al, Fe, Ca) with which they were associated in soil".

p. 3, lines 28-30: Yes, there are other methods to study P dynamics and soil P composition, but none of these methods is perfect individually. For example, while XANES is a solid-state technique that does not require extraction, P concentrations are often below the detection limit, so it can only detect broad P species groups (e.g. Fe-P, Ca-P), but can't for example say if DNA is sorbed to Fe or Al. The most thorough studies of soil P use a combination of techniques together, and not any single technique (e.g. Liu et al. 2013 J. Environ. Qual 42:1763-1770; Liu et al. 2015 Environ Sci Technol 49:168-176).

p. 3, lines 30-32: "This does not mean that the results on pyrophosphate, polyphosphate and total orthophosphate concentrations are not useful, however, there are other inorganic P compounds of importance in soils". This sentence makes no sense to me. No, there are no other inorganic P compound in soil other than orthophosphate, pyrophosphate and polyphosphate. However, as noted above, extraction with NaOH-EDTA removes inorganic (and organic) P compound from the cations with which they are associated in soils. Thus, as noted below, a combination of techniques will give the most complete picture of soill P speciation and dynamics. Please rewrite this sentence.

5. Methods: These are now clear and well-written.

6. Results:
p. 7, line 1: "there was no pH effect on both pools" should be "there was no pH effect on either pool"
p. 7, line 6: "additional Appendix S3" delete "additional"
p. 7, line 8: "additional Appendix S6" delete "additional"
p. 9, lines 5-6: "organic and inorganic P" should be "$P_o$ and $P_i$" to be consistent with the rest of the manuscript

7. Discussion: In general, this part of the manuscript is improved compared to previous versions.
p. 10, line 5: "P organic and inorganic compounds" should be "$P_o$ and $P_i$ compounds"
p. 10, line 8: "the decaying degree of C element is lower than the P" I do not understand what you are trying to say here. Is it "organic matter degrades more slowly than $P_o$ compounds"? Please rewrite.
p. 10, line 9: "Turner and Condron 2013" is an opening paper introducing a special issue, and does not contain data to support this statement of fact. Please cite another reference that actually contains data.
p. 10, lines 16-17: It is not possible to determine specific P compounds such as apatite with Hedley fractionation, and the long extraction times likely also degrade organic P. Please cite a better reference, with actual soil chemical data to support this point, and not a review article of Hedley fractionation (Yang and Post 2011).
p. 10, lines 23-25: please cite an original study that contains data, and not a review paper (Huang et al. 2017) to support this statement of fact
p. 11, line 6: "proportion" should be "proportions".
p. 11, lines 26-28: These are also conditions under which ectomycorrhizal fungi are found. These fungi produce hyphal mats in the forest floor, so an increase in polyphosphates could reflect an increase in ectomycorrhizal hyphal mats.
p. 12, line 5: "pH affect" should be "pH affects"
p. 12, line 5: "organic and inorganic P compounds" should be "$P_o$ and $P_i$ compounds"
p. 12, line 14: "on the clay minerals", delete "the"
p. 12, lines 15-18: I am pleased to see the authors citing original studies about DNA sorption at the start of this section. However, I do not understand why they cited Yu et al. 2013 at the end of the section (line 18), because this is a review paper. Please replace this citation with a paper containing original research to support this statement of fact.
p. 12, line 22: "for the P acquisition" delete "the"
p. 12, line 25: "organic P" should be "$P_o$"
p. 12, line 28: "George et al. 2017" is a broad review paper; please cite a more specific reference to support this statement of fact.
p. 12, line 28: "Plant and microorganism" should be "Plants and microorganisms"; "diesters" should be "orthophosphate diesters" or "phosphodiesters".
p. 12, line 29: "monoesters" should be "orthophosphate monoesters", or "phosphomonoesters"; monoesters and diesters are general bond descriptions; the "orthophosphate" or "phospho" is more specific
p. 12, line 30: Why is Turner 2008a cited to support this statement of fact about phosphatases? This was known years before the Turner paper was published (e.g. Halsted 1964, Skujins 1967, Tabatabai and Bremner 1969).
p. 12, line 33: "both solubilization and hydrolysis by the phytase" As written, this implies that phytase both solubilizes and hydrolyzes inositol hexaphosphates, which is incorrect. Organic acids are required to desorb inositol phosphates, so that phytases can hydrolyze them. Please rewrite these lines, with a more appropriate reference than Turner 2008a.
p. 13, line 7: "effect in" should be "effect on"
p. 13, line 8: "did not had" should be "had no"
p. 13, line 16: Huang et al. 2017 provides no direct evidence of phytate mineralization by ectomycorrhizal fungi. Please cite a better reference to support this statement of fact.

p. 14, lines 12-13: Turner et al. 2002 is a review paper about inositol phosphates, so why is it being cited to support a general statement about temperature and phosphatase activity? Please replace this with a more appropriate reference.

p. 14, lines 24-29: Why are Walker and Syers 1976 and Feng et al. 2016 being cited to support discussion about decreased orthophosphate measured in NaOH-EDTA extracts by NMR? Neither of the cited references used NaOH-EDTA extraction or NMR, so they are irrelevant to the discussion here. Please replace these with better references. The authors should also mention here that caution needs to be used when discussing changes in orthophosphate extracted by NaOH-EDTA for NMR, because it will preferentially extract organic P rather than orthophosphate. As such, studies of the residuals after NaOH-EDTA show that it is mainly orthophosphate, especially for soils with low total P recovery by NaOH-EDTA.

p. 15, line 9, "Even though" should be "However"

p. 15, lines 13 and 17: "Vitousek et al. 1995" is cited in the text but is not listed in the References

p. 15, lines 23-25: "phosphorus" should be abbreviated to be consistent with the rest of the manuscript.

p. 15, line 29: "why inositol hexakisphosphates have not been found" should be "why are inositol hexakisphosphates not found"

p. 16, line 15: "functional groups only (i.e. diesters and monoesters)" should be "broad compound classes only (i.e. orthophosphate diesters and monoesters"

p. 16, line 15: "when compounds concentrations" should be "when compound concentrations"

8. Conclusions:
   p. 15, line 27: "as pedogenesis evolve" should be "as pedogenesis evolves"

9. Acknowledgements: "NSERC-Discover" should be "NSERC Discovery"; "from which studies" should be "from whose studies"

10. References: in general, this section is greatly improved compared to the previous version of this manuscript. However, there are still differences in formatting (e.g. Cloy et al. 2014) and some spelling mistakes (e.g. "Biophsyical" in Wang et al. 2014". Please proofread this section carefully.

---

## Author Response (AR2)

*General comment to the Editor*

Dear Associate Editor Sönke Zaehle,

We sincerely thank you for re-evaluating our manuscript and thank the reviewers for the significant improvements they promoted on the manuscript through their comments. We responded to each comment and revised the manuscript. We made appropriate changes in the manuscript or have provided an explanation below where we did not make a change suggested by a reviewer. To the best of our understanding, no remaining issues are left in this corrected version.

Responses to the reviewers comments are described below under "**Responses by the authors to the specific comments**". Changes made on the manuscript were all based on these responses. These responses were adapted from the ones provided on the response to the reviewers ('Response to the Anonymous Referees #1 and #2, on 18 Nov 2017). To facilitate the evaluation, we provided the comments made by the reviewers before each response by the authors. Responses made by the authors are in blue and indicated with the terms "***Response to".*** After the responses to the reviewers comments one can find the manuscript with all changes marked (after page 15 on this document). Changes made on the supplementary file are also being provided.

The major changes made on the manuscript were that we focused on specific organic P compounds (i.e. DNA and IHP) instead of its respective functional groups (diester and monoester). All statistical analyses regarding these variables were re-made, and we added a new path analysis regarding inorganic P compounds. New results and discussions were added, and we provided new references to support our discussions. We also updated our data set to November 17, 2017 (previously January 2017). We added more supplementary material as appendices to better explain the random factor effects over the studied variables, and also added the Whittaker's diagram to determine the main biomes comprised in our dataset. We reduced the number of figures on the main document, and have provided them as supplementary material. Finally a native language specialist revised the English. All the other specific changes can be found on the responses to the reviewers or the marked document, both provided below.

Just for clarification, changes were made in two steps, the first was before the language revision, and the second was after the language revision, but the documents regarding the two steps are now merged to facilitate the evaluation process (instead of providing separate documents for each step as we have done in the first attempt to submit the revised manuscript). The alterations made by a non-author refer to the language correction personnel (marked changes made by Cindy Joyac). We apologize for the confusion on the first attempt to submit the revised manuscript. We now followed all requested steps and they are specifically described along this document.

**Responses by the authors to the specific comments made by the reviewer 1.**

R1 General comment
Organic phosphorus (P) cycling in soils is a topic that has received attention in recent years. As more papers are published, meta-analyses that link the data from these papers together to identify trends in organic P cycling become possible, at least in theory, and a paper presenting novel findings could be of interest to readers. However, deriving meaningful interpretations from a meta-analysis of soil P-NMR studies requires a clear understanding of the P-NMR method and its limitations, in order to correct for known artifacts of analysis. This was not done for this manuscript. As such, it cannot be published in its present form, and will require a major revision, including reanalysis of data, to make it publishable.

*Response to R1 General comment*
We understand the point the reviewer made about using the correction for potentially degraded peaks (of diesters converted to monoesters). Just to clarify, we did not use the correction previously because 39% of inositol phosphate (comprehending all tropical results and other locates) and 12% of DNA results were absent from the compiled data. We knew that correction was possible through adding to the total diesters concentration, the $\alpha$ - and $\beta$ -glycerophosphate concentrations (potentialy degraded peaks), but the reviewer also provided additional details that could improve our analysis. To address the issue, we followed reviewer suggestion. Using the available data, we focused on specific organic P compounds (i.e. DNA and IHP) instead of its respective functional groups (diester and monoester). Given the huge proportions of potentially degraded peaks (non-inositol monoesters), and the uncertain about which compounds were present in this potentially degraded fraction, we choose to not to work with the corrected di-to-mono ratio, focusing on DNA and IHP compounds instead.

R1 Comment 1.

Writing quality: a) The quality of English in the manuscript is poor in many places. If the authors revise this manuscript, I suggest they have it read by someone more familiar with English, who also understands the research field. b) Please check that you are using the correct spelling of the names of authors whose papers are cited. For example, "Vincent" is repeatedly cited as "Vicent", including in the supplemental files. c) Be specific with terminology. The term "P" is an abbreviation for the element for phosphorus. However, the authors use it interchangeably for phosphate, which is incorrect.

*Response to R1 Comment 1*
a) In the new manuscript version, a native language specialist revised the English.

b) "Vicent" will be replaced by "Vincent", and all the names of the other authors whose papers are cited were checked.
c) In the new manuscript version, the terminology was revised regarding the proper use of abbreviations. "P" was used as an abbreviation for the element phosphorus, Po and Pi were used for the respective organic and inorganic pools, and the other P compounds were described by their proper names.

R1 Comment 2.

As P-NMR has become more widely used to characterize soil P forms, enough data has become available to indicate the possibility of using these data in meta-analyses to look at soil factors controlling P forms, especially organic P. However, those of us who use this technique the most also recognize its limitations. Although the use of P-NMR has advanced our understanding of soil organic P cycling more than almost any other method to date, the technique is not perfect. It is important to understand the artifacts of the method. It is also important to separate P-NMR results on a soil extract from the P forms that would have been present in the original soil sample prior to extraction. After all, isn't that the objective of a soil science study? Unfortunately, it isn't clear to me that the authors of this manuscript are familiar enough with the soil P-NMR technique to understand its limitations and address them. This has produced a study that clearly involved a lot of work by the authors, but which ultimately has not produced any new insights with respect to soil P.

*Response to R1 Comment 2*
   • We recognize that P-NMR can have limitations, and we have addressed them in specific parts of the manuscript. We emphasized those limitations according to the suggested comments.
   • Regarding the separations of P-NMR results from other P forms present in the original soil sample, we worked with P-NMR results obtained from NaOH-EDTA extracts only (Y axis on figures 2, 3, 4, 5, and 7, which do not include the residual P, i.e. difference between soil total P and NaOH-EDTA P). The total P of NaOH-EDTA extracts could be obtained by adding Organic P (e.g. figure 3A) to Inorganic P (e.g. figure 2A), but it does not correspond to the soil total P. The total P (obtained with other method – not P-NMR, e.g. digestion) was also presented in the manuscript, but acknowledging that it was obtained by a different method. In the new manuscript version, added more information in the figure captions to avoid misunderstandings, i.e., results in the Y axis are from NaOH-EDTA P-NMR results.

Some specific areas of concern are:
a) Concentration: It is not possible to determine absolute concentrations of P forms or compound classes using NMR; only relative percentages can be determined, because it is a compositional analysis in which the total must be 100%.

Concentrations of P forms are then determined by multiplying by the total extracted P concentration by the percentage of each P form, which is still based on the compositional analysis. This is why the proportions and concentrations of total organic P and total inorganic P (Figs. 2 and 3) show inverse relationships to one another – together they have to add to 100%. This is exactly what would be expected, so it is strange to me that the authors would comment on this (p. 6, lines 13-16). The authors also do not seem to understand the relationship between total P in the soils and P extraction in NaOH- EDTA. In natural (non-tilled) samples, P is stratified, such that concentrations are higher at the soil surface and lower with depth. There will also be an increase in organic P at the soil surface from inputs of plant material, which will decrease with depth – especially in forests with limited mixing and with greater fungal activity in mats in the forest floor (as is typical for temperate forests, where the majority of these studies were conducted). This needs to be accounted for somehow.

**Response to R1 Comment 2a**

Our total organic and inorganic P results, on mg kg-1 basis, are from NaOH–EDTA extracts only (do not include residual P, i.e. difference between soil total P and total P of NaOH-EDTA extracts). Based on our understanding, the results on mg kg-1 basis were determined from the proportion (%) of each P compound or functional group on spectra (determined by integration of peaks area or deconvolution) multiplied by the total P extracted with NaOH–EDTA. Most authors have presented their P-NMR results (forms and compounds) on both % and mg kg-1 basis (from P-NMR results of NaOH–EDTA extracts), including most of the ones we compiled data from. In the new manuscript version, we added more information on figure captions to state that results on Y axis are from NaOH–EDTA extracts only. Usually, P-NMR results from NaOH-EDTA soil extracts are presented in both ways: (a) on mg kg−1 basis (non-including residual P), and (b) relative distribution of P (%). We followed the same criteria used by those papers to present our results.

We do understand that results are based on a compositional analysis (i.e. P forms are determined by multiplying the total P extracted with NaOH-EDTA by the percentage of each P form), but the description of the inverse relation (obviously a inverse relation) between organic and inorganic concentration (% of total NaOH EDTA P) meant to explore the phenomena of pH or other variable impacting these forms. It was the way we found to describe our results. In the new manuscript version, we reformulated the text avoiding the obviousness on describing results from percentages. In the specific case, the sentence containing "they showed a contrasting behavior" was excluded.

We do understand that soil total P is different than soil P extracted with NaOH- EDTA. We have mentioned that on Page 7 lines 10-13 "It's important to note that the reported total P is the one obtained by digestion and usually comprise the residual P non-recovered by the NaOH EDTA extractant. The recovery of total P by NaOH EDTA extraction is variable depending on soil characteristics and laboratory procedures (Cade-Menun and Liu, 2014)." Moreover, knowing that there is a potential effect of soil conditions and laboratory procedures, we used the P recovery

(percentage of P extracted with NaOH EDTA from soil total P) as a random factor in all bivariate regression models.

We agree that natural (non-tilled) samples have stratified nutrient distributions. Our supplemental Figure S2 presented the results obtained regarding this effect. But contrary to what was expected, we found no effect of sampling depth over organic P concentration in mg kg$^{-1}$, neither for both organic and inorganic on % basis (even though functional groups of organic and inorganic P responded dynamically to soil depth, even having contrasting responses for organic and mineral soil layers). We did find a sampling depth effect for inorganic P concentration in mg kg$^{-1}$. Therefore, knowing that there is a potential effect of sampling depth, we used it as a random factor in all bivariate regression models.

b) Extraction efficiency and soil pH: It has been very well established that the recovery of total P from soil samples with NaOH-EDTA extraction is never 100%, and is higher from samples with lower pH. The extraction seems to favor samples high in iron and aluminum, with generally poor P recovery from samples high in calcium; the reasons for this are unclear. As such, any meta-analysis comparing across a range of sample must take into account differences in P recovery among studies, and even among depths within the same soil profile or at different points along a soil chronosequence. For example, the recovery of total P in the samples for the Turner et al. (2003) paper ranged from 14-45%, in the Turner et al. (2007) paper 63-91%, and in the McDowell et al. (2007) paper 11-75%. If the purpose of this meta-analysis is to look at factors controlling soil P, then these differences in recovery must be factored in. Is it even possible to compare the results for a soil where only 11% of the total P was extracted to one with 91% extraction? What about the 89% of total P that wasn't extracted? The authors of this manuscript don't even mention this as a factor, let alone correct for it. And that, unfortunately, undermines their results.

*Response to R1 Comment 2b*
We do understand that soil chemical characteristics can impact the recovery of P with a NaOH-EDTA extraction. We also agree that an "analysis comparing across a range of sample must take into account differences in P recovery among studies, and even among depths within the same soil profile". We have already addressed that using the 1) P recovery, and 2) sampling depths as random factors (and also latitude for other purpose not directly associated with the comment) in the analysis (which were described in the methods section Page 5 lines 7-15). An example of the impact of a random factor is described in the Page 6 lines 18-20: "There was no pH effect over this inorganic compounds in the organic layer (even though there is an apparent trend, these relationships became non-significant after including sampling depth as random effect on models; Supplementary Appendix S2 shows the sampling depth effect over soil P composition)."

c) Degradation: As noted, it is important for any soil study to ensure that the forms discussed, or the ratios of compound classes such as orthophosphate monoesters and diesters, are based on what was in the original soil sample, and not what was produced during extraction and analysis. It is well established that some orthophosphate diesters such as RNA and phospholipids can degrade to the orthophosphate monoesters $\alpha$- and $\beta$-glycerophosphates (phospholipids) and various monophosphates (RNA) when analyzed at the high pH required for good peak separation in P-NMR spectra [e.g. Turner et al. 2003; Doolette et al. 2009; He et al. 2011, Vincent et al. 2013; Schneider et al. 2016. The degree of degradation will vary depending on the length of NMR experiment and other factors [see Cade-Menun and Liu (2014) and Cade-Menun (2015) for more details]. It is essential that these degradation peaks are identified and quantified in order to determine the correct concentrations of orthophosphate monoesters and diesters that were in the original soil sample; doing so improves any comparison of these P forms to other soil properties (e.g. Young et al., 2013; Liu et al. 2013 J. Environ. Qual. 42:1763-1770). Unfortunately, most studies before 2010 did not identify these compounds and correct for degradation. The authors of this manuscript acknowledge that degradation can occur (p. 4), but for some reason have chosen to ignore it, which is a major problem. The issue of degradation MUST be addressed for any study of edaphic and climatic characteristics to have any meaning. If the concentrations of orthophosphate monoesters and diesters were not corrected in the original study, then the authors of this manuscript could have applied some correction factor to compensate. For example, Vincent et al. (2013) note that most non-inositol phosphate monoesters were diester breakdown products (p. 160). The studies used by the authors here all included some measurement of inositol phosphates (at least myo-IHP and scyllo-IHP). As such, the authors could have assumed that those were the only true monoesters, and corrected the remaining proportion of monoesters to diesters. It would have at least been more meaningful that what they did, which was to ignore degradation but then reach the conclusion that the ratio of diesters to monoesters was a significant factor in the study.

*Response to R1 Comment 2c*
We understand and agree with the reviewer's comment. But, as described in the methods section Page 4 lines 14-16: "We know that it is possible to correct degraded peaks of diesters converted to monoesters (e.g., Young et al., 2013 and Cade-Menun et al., 2010), but since some papers only showed functional groups like monoesters and diesters, and not species (specific P compounds) inside these functional groups, this correction was not done." Not all studies used in this manuscript included some measurement of inositol phosphates (at least myo-IHP and scyllo-IHP).

Specifically, the following papers did not present P species (including myo-IHP and scyllo-IHP) inside these functional groups (monoesters and diesters) are: Celi et al., 2013, n=4; Vincent et al. 2010, n=1; Turner, 2008b, n=1; Turner et al 2003 (native soil sample), n=1; Turner and Engelbrecht, 2011, n=19; Turner et al 2014, n=10; and therefore correction was not possible to be addressed properly based on

our previous knowledge. Some of these authors acknowledge that there is a small contribution of inositol phosphates (most tropical soils) while others have provided no explanation about with they did not present specific P compounds results. Therefore, we thought it will still be biased to assume something that we were not certain of (i.e. amount of inositol phosphates).

As described earlier, to address the issue, we followed the reviewer suggestion. Using the available data, we focused on specific organic P compounds (i.e. DNA and IHP) instead of its respective functional groups (diester and monoester).

According to the gathered data, non-inositol monoesters (potentially degraded peaks, as suggested by the reviewer) corresponded to 66.76 % in average of the total amount of non-corrected monoesters (ranging from 7.8 to 100%), previously reported as total monoesters content, from papers that presented IHP results (n=61). The same non-inositol monoesters (potentially degraded peaks) corresponded to 53.94 % in average of the total NaOH EDTA organic P amount (ranging from 6.47 to 100%) from papers that presented IHP results (n=61).

Based of the results presented by the authors we could not calculate how much of the potentially degraded peaks were: $\alpha$ - and $\beta$ -glycerophosphate (Doolette et al., 2009), nor RNA and phospholipid (which includes glycerophosphates) (Vincent et al., 2013); which were determined as degraded peaks by those authors.

Therefore, given the proportions, correcting for potentially degraded peaks has a huge impact on the results, and it is a not completely unbiased calculation, since we don't know if all potentially degraded peaks were $\alpha$ - and $\beta$ - glycerophosphate (Doolette et al., 2009), or RNA and phospholipid (Vincent et al., 2013), so we choose to not work with the di-to-mono ratio.

Inositol plus DNA represented 59.20% in average of total NaOH EDTA organic P (n=51) from papers that presented both DNA and IHP results. Therefore, it is also a huge proportion and could be an unbiased approach for those results.

The reported proportions are not closing exactly due to the different datasets (n = 51 and n=61).

To re-analyze data, IHP was not considered for tropical soil results because they have non-detected concentrations of this compound (but tropical results will be maintained for the other variables).

The following two paragraphs were written just to clarify why we have done the analysis in the previous way.

We tried to be as clear as possible about this issue, as it is written in the Page 4 lines 18-20: "We expect for future researches to provide results of as much soil P species they can find rather than functional groups only, even when species concentrations are low (and describe when species are not detected), what may enable future analysis to avoid possible confounding effects of organic P species inside functional groups (e.g., inositol and monoesters)." So, we believe that some questions will still remain to be addressed regarding soil P composition in terrestrial natural ecosystems, but our manuscript will provide significant and robust information using currently available results from literature.

We understand the importance of what the reviewer is asking for, and recognize that in the manuscript, but as described we could not reach that level of detail due to absence of data (all specific P compounds). We have used an approach used by other authors. The same approach of not correcting for potentially degraded peaks was used in another recent paper, for example, that combined results from pasture soils using P-NMR results of NaOH–EDTA extracts (Nash et al., 2014). Essentially, they did not corrected for any degraded peak to determine the diester-to-monoester ratio, and described that this was out of their scope, but we agree that their approach is also not optimal.

R1 Comment 3.

Selection of studies: The authors indicate in the methods that they were careful in their selection of papers to include in their meta-analysis, such as native vegetation. As such, I am puzzled as to why the Turner et al. 2003 paper was included as the only study from the USA, because it used agricultural soils. And while the abstract and elsewhere in the text indicate a "dataset including 88 sites", these are overwhelmingly biased to sites in New Zealand (59) and Panama (21), which does not cover a range of "temporal, edaphic and climatic characteristics". The sites selected are also mainly from chronosequence studies, which may also have affected the P forms and their relationship to soil properties.

*Response to R1 Comment 3*
The Turner et al. 2003 paper included most soils under arable cropping, although there was a native site, and this was the one we included in our analysis. We understand that we were not able to cover a vast representative sample, at global level, but we included as much as we could, given the data availability on the literature. This compilation made this study to have the wider geographical coverage on the topic (terrestrial environments with native vegetation - P-NMR results of NaOH–EDTA extracts).

R1 Comment 4.

Introduction:
a) Please include references for all statements of fact, and make sure those facts are correct. For example, p. 1, lines 24-25: "Once P has been dissolved as free orthophosphate" It isn't possible for free orthophosphate to exist in the soil solution; it will still be associated with cations, although as more soluble forms.
b) Be careful with terminology. Page 2, line 1: "inorganic and organic P pools are each composed by fractions or functional groups". No, they are composed of specific P compounds. The term "functional group" is used elsewhere in the introduction. Please indicate what is meant by this term, which isn't one used for soil P chemistry. And note that fractionation measures operationally-defined P pools, rather than specific P forms.

c) Page 2, line 10: Turner 2007 is not cited in the references.

*Response to R1 Comment 4*
a) In the new manuscript version, all statements of fact were referenced, and it was make sure that those facts were correct. Specifically, "as free orthophosphate" will be excluded from the sentence. In other occurrences we used "available" instead of "free" when referring to P that could be potentially taken up by plants.

b) In the new manuscript version, the statement was reviewed clarifying that inorganic and organic P pools are composed of specific P compounds. "Functional groups" were changed to compounds in the whole manuscript when describing P compounds.

c) It was corrected in the new manuscript version. The correction is "Turner et al., 2007", which was previously cited in other parts of the manuscript.

R1 Comment 5.

Methods:
a) See comments above about site selection.
b) Page 4, lines 14-23: This discussion about degradation belongs in the Discussion section, not the methods section.
c) The authors have made a lot of assumptions here, particularly for soil classification. Please justify these assumptions in the Discussion section of the manuscript.

*Response to R1 Comment 5*
a) It was answered on Authors' response to comment 3.

b) In the new manuscript version, we moved the part about degradation to the Discussion section.

c) In the new manuscript version, the assumptions about soil classification were justified in the discussion section. The assumptions include:
- The soil total P content depends on both weathering stages and parent material, but generally decreases with increasingly weathered soil orders (Yang and Post, 2011).
- The soil weathering stages classification also takes into account changes in soil P composition, and generally follows the Walker and Syers (1976) conceptual model: there is a gradual decrease and eventual depletion of primary mineral P (mainly apatite P), a decrease of total P, an increase and then decrease of total organic P, and a increase and eventual dominance of occluded P during soil development (Yang and Post, 2011).

- In highly weathered soils, occluded P increases at the expense of organic P through by encapsulation of mineralized P inside of Fe and Al minerals (Crews et al., 1995).

R1 Comment 6.

Results: a) I am puzzled by the phrase "concentration (% of total NaOH EDTA P)", page 6 line 30. Do you mean % or concentration in mg/kg? They are not the same thing, although they are derived from the same data (% of P forms multiplied by extract concentration).
b) As noted above, any results related to total concentrations or percentages of orthophosphate monoesters, orthophosphate diesters and the diester:monoester ratio are meaningless if not corrected for degradation. The authors must remove all reference to uncorrected concentrations and ratios. They could correct them as suggested above, or they could focus on specific P forms (e.g. DNA or IHP).

*Response to R1 Comment 6*
a) We meant % of NaOH EDTA P in %.

b) We consider to be the same response of "Authors' response to comment 2c)". We did not correct them for degradation in the previous manuscript version. As described earlier, to address the issue, we will follow the reviewer suggestion. Using the available data, we now focused on specific organic P compounds (i.e. DNA and IHP) instead of its respective functional groups (diester and monoester).

R1 Comment 7. Discussion: Given the issues noted above, I am not sure there is anything meaningful in the discussion section, which as written is a review of the temporal, edaphic and climatic characteristics affecting P forms in NaOH-EDTA extracts, rather than in the original soils themselves. This is really unfortunate given the amount of work the authors put into this study. I hope the authors will address these issues. When they do, I expect much of the discussion section to change.

*Response to R1 Comment 7*
As described above, we focused on specific organic P compounds (i.e. DNA and IHP). Specifically, we deleted discussion about the mechanisms that prompted the inverse response of monoesters and diesters as P limitation increased (since those functional groups results were excluded from the manuscript). Discussion was added about why DNA concentration increased as both P limitation and soil acidity increased in older, more weathered soil systems. Discussion was also added about the increase in inositol phosphates concentrations at more acidic soil environments.

R1 Comment 8.

Figures:
a) The two figures used for Figure 1 were both published elsewhere, and thus are covered by copyright. However, the authors do not indicate anywhere that they have permission to use these figures in their manuscript, which must be obtained from the publishers of the original papers.
b) All figures containing references to total orthophosphate monoesters and diesters, and the diester:monoester ratio (e.g. 3, 5, 7, 8, 9, S4.1, S4.2, S4.3, S4.4) must be corrected for degradation. And all figures will likely change when the authors have normalized the data used in this study for P recovery.

*Response to R1 Comment 8*
a) In the new manuscript version, we provided the coverage by copyright. License Numbers: 4210920836823 (Elsevier) and 4210930550479 (John Wiley and Sons).

b) The response about the correction for degradation is on "Authors' response to comment 2c)", and regarding the normalization for P recovery is addressed on the "Authors' response to comment 2a)".

**Responses by the authors to the specific comments made by the reviewer 2.**

- The writing of the paper needs to be improved. The paper is dense and hard to read like it is.
- The main messages to take home are not clear, these must be highlighted.
- There are to many bivariate graphs that distract to understand the main messages. I would suggest to add most of them for the supplementary material and keep in the main text the ones that are significant and are used to describe main processes in the text.
- The authors present the patterns shown as global, but there is no reference on the role of different biomes and plant communities, which are in turn related to soil properties. Ecological implications for the relations seen are missing.
- Because of the distribution of the dataset, where most of the samples are from New Zealand, the authors should address the associated bias that the data could have.
- The authors consider the weathering status as a temporal proxy (as it is said in the abstract) to be crossed with soil and climate properties. However, weathering status in this paper is defined by soil type, which makes this classification at certain point redundant with soil properties and climate. The authors should clarify this decision.
- To assume organic C as total C is only acceptable in organic soils. This assumption can lead to large errors in calcareous soils.
- Why the path analysis is used to explain exclusively diester/monoester ratio and not other P-form? Is this ratio providing specific information on nutrient state of the ecosystem? Is significant for understanding P-limitation or inorganic control over the P cycle? This should be argued.
- I miss a clear explanation on the role of the basement/parent material.

*Response to R2 General comments*
- The writing of the paper was improved making it less dense and easier to read, and a native language specialist revised it.

- We believe that the main messages were responses related to the increasing complexity of phosphorus compounds as pedogenesis progress; therefore, we emphasized those aspects across the manuscript to make more clear the take home messages.
- We agree that we have too many bivariate graphs, so we reduced one variable in Figure 3 (phosphonate, adding it as supplementary material), and added both climatic figures as supplementary material (organic and inorganic P vs. climatic drivers, Figures 4 and 5).
- Ecological implications of different biomes and plant communities over soil P composition were added to the manuscript discussion. This was done through determining the main biomes comprised in our data set using the annual precipitation – temperature diagram (whittaker diagram).
- The potential associated bias that the data could have because most of the samples were from New Zealand will be added to the manuscript. We related this discussion with the soil orders and biomes comprised.

- Regarding the redundancy between soil properties and climate with weathering, this is one reason why we used the path analysis. We can control for the redundancy using such statistical analysis. The figure of soil weathering stage relationship with soil age also showed that our classification followed the patterns that could be expected between soil weathering and soil age and between soil weathering and soil C. However, we do not have enough data to base our weathering classification only based on soil age.

-We checked all data from soils with pH higher than 7 and they have measured organic C, instead of total C. We added this description in the Material and Methods section to clarify this issue for readers.

- We agree that it is not explaining the inorganic P composition, but to fill this gap we have determined the path analysis for the inorganic P compounds.

- We added more explanation on the role of the parent material over soil P composition and implications related to the soil weathering stages classification. The soil total P content depends on both weathering stages and parent material, but generally decreases with increasingly weathered soil orders (Yang and Post, 2011). The soil weathering stages classification also takes into account changes in soil P composition, and generally follows the Walker and Syers (1976) conceptual model: there is a gradual decrease and eventual depletion of primary mineral P (mainly apatite P), a decrease of total P, an increase and then decrease of total organic P, and a increase and eventual dominance of occluded P during soil development (Yang and Post, 2011). In highly weathered soils, occluded P increases at the expense of organic P through by encapsulation of mineralized P inside of Fe and Al minerals (Crews et al., 1995).

R2 Specific comments

- The last sentence of the abstract is not telling anything new "organic and inorganic P pools as well as their functional groups composition are determined by distinctive drivers that regulate key ecological governing their presence..."

- Pag 2, line 22, which 5 factors?

- Pag 4, line 27, starts a list with "a) " but no more items are listed

- Pag 7, line 25, the no effect of many climatic variables can be related to the geographic bias of the dataset. Should be argued.

- Pag 8, line 10. Is obvious that poorly crystalline Fe and Al, do not correspond to weathering status if we consider the classification status than the authors have used. However, the presence of these oxides can deeply influence the P pools and cycles in Oxisols and Ultisols but also Andosols.

- Pag 9, line 10. This is a too ambitious sentence. There is no information presented in this study about the variability among communities or different biomes. It is not explained neither how some edaphic variables depend on climate.

- Precipitation and moisture index give similar bivariate relations, maybe with one of both variables would be enough.

***Response to R2 Specific comments***
- The last sentence of the abstract was modified to be more meaningful. It was changed to "We conclude that soil P composition is determined by edaphic and climatic drivers that regulate key ecological processes on terrestrial natural ecosystems. These processes are related to the source of P inputs, primarily determined by parent material and soil forming factors, and after altogether with plants and microbes coexistence, the bio-physico-chemical properties governing soil phosphatase activity, soil solid surface specific reactivity and P losses through leaching, and the P persistence induced by increasing complexity of P organic and inorganic compounds as pedogenesis evolve.
- Pag 2, line 22. We added the five state factors determining soil weathering.
- Pag 4, line 27. The "a)" was remaining from a previous version. It was excluded in the current version.
- Pag 7, line 25. The potential bias promoted by the geographic concentration of the studied sites was added to the manuscript.
- Pag 8, line 10. We added this explanation to the manuscript.
- Pag 9, line 10. We attenuated the sentence following the reviewer suggestion. "Unprecedented" was changed to "wide". Moreover, as described in the Response to R2 General comments we will improve discussion about what are the biomes represented in our dataset (whittaker diagram), and will add this information on that specific part of discussion. How edaphic variables depend on climate is being explored on the path analysis, but we will improve discussion on theoretical aspects of those relationships.
- Given the similarity between precipitation and the moisture index, the latter has been excluded from the manuscript.

**References**

[revised manuscript text omitted]

L D 12/13/2017 11:22 AM

Unknown

Maire, Vincent 1/9/2018 10:01 PM
Comment [7]: Please specify if we are in mg/kg or in % unit

L D 10/27/2017 8:05 PM

L D 9/19/2017 3:42 PM

L D 1/16/2018 5:04 PM

[Figure]

[Figure]

[Figure]

Unknown

Maire, Vincent 1/9/2018 10:02 PM

**Comment [8]:** I've checked path analyses and I've got different results with better P-value when parent rock is included. For that, I transform the va... [332]

L D 1/16/2018 4:49 PM

**Comment [9]:** OK

L D 10/27/2017 8:05 PM

L D 12/13/2017 8:42 PM

L D 1/16/2018 4:44 PM

L D 1/16/2018 4:45 PM

L D 12/13/2017 9:00 PM

L D 12/13/2017 9:00 PM

L D 12/13/2017 9:00 PM

[revised manuscript text omitted]

We expect for future researches to provide results of as much soil P species they can find rather than functional groups only, even when species concentrations are low (and describe when species are not detected), what may enable future analysis to avoid possible confounding effects of organic P species inside functional groups (e.g., inositol and monoesters). And researchers must determine variances or standard errors for soils with distinctive properties. Then, as stated by Stewart (2010), future analysis can use the different amounts of information that studies of different sizes and different quality present in a meta-analytic approach.

| Page 4: [64] Formatted | L D | 9/20/17 10:12 PM |
|---|---|---|

Highlight

| Page 4: [65] Deleted | L D | 1/17/18 12:08 PM |
|---|---|---|

weathering st

| Page 4: [65] Deleted | L D | 1/17/18 12:08 PM |
|---|---|---|

weathering st

| Page 4: [65] Deleted | L D | 1/17/18 12:08 PM |
|---|---|---|

weathering st

| Page 5: [66] Deleted | Cindy Joyac | 1/5/18 11:20 AM |
|---|---|---|

It was not found any duplicity

| Page 5: [66] Deleted | Cindy Joyac | 1/5/18 11:20 AM |
|---|---|---|

It was not found any duplicity

| Page 5: [66] Deleted | Cindy Joyac | 1/5/18 11:20 AM |
|---|---|---|

It was not found any duplicity

| Page 5: [67] Deleted | Cindy Joyac | 1/5/18 11:20 AM |
|---|---|---|

Not all results had all variables available

| Page 5: [67] Deleted | Cindy Joyac | 1/5/18 11:20 AM |
|---|---|---|

Not all results had all variables available

| Page 5: [68] Moved from page 5 (Move #3) | L D | 12/8/17 1:49 PM |
|---|---|---|

The resultant resulting dataset is available in the SupplementarySupplementaryadditional Appendix S1.

| Page 5: [69] Deleted | Cindy Joyac | 1/5/18 11:21 AM |
|---|---|---|

the

| Page 5: [69] Deleted | Cindy Joyac | 1/5/18 11:21 AM |
| --- | --- | --- |

the

| Page 5: [70] Deleted | Cindy Joyac | 1/5/18 11:21 AM |
| --- | --- | --- |

the

| Page 5: [70] Deleted | Cindy Joyac | 1/5/18 11:21 AM |
| --- | --- | --- |

the

| Page 5: [71] Moved to page 5 (Move #3) | L D | 12/8/17 1:49 PM |
| --- | --- | --- |

The resultant dataset is available in the Supplementary Appendix S1.

| Page 5: [72] Deleted | L D | 12/6/17 3:02 PM |
| --- | --- | --- |

The moisture index which is the ratio between precipitation and evapotranspiration was considered as another climatic variable that likely impact soil weathering and properties. It was calculated based on Wang et al. (2014a).

| Page 5: [73] Deleted | Cindy Joyac | 1/5/18 10:30 AM |
| --- | --- | --- |

,

| Page 5: [73] Deleted | Cindy Joyac | 1/5/18 10:30 AM |
| --- | --- | --- |

,

| Page 5: [74] Deleted | Cindy Joyac | 1/5/18 10:30 AM |
| --- | --- | --- |

,

| Page 5: [74] Deleted | Cindy Joyac | 1/5/18 10:30 AM |
| --- | --- | --- |

,

| Page 5: [75] Deleted | Cindy Joyac | 1/5/18 11:25 AM |
| --- | --- | --- |

variable

| Page 5: [75] Deleted | Cindy Joyac | 1/5/18 11:25 AM |
| --- | --- | --- |

variable

| Page 5: [75] Deleted | Cindy Joyac | 1/5/18 11:25 AM |
| --- | --- | --- |

variable

| Page 5: [76] Deleted | Cindy Joyac | 1/5/18 10:30 AM |
| --- | --- | --- |

,

| Page 5: [76] Deleted | Cindy Joyac | 1/5/18 10:30 AM |
| --- | --- | --- |

,

| Page 5: [77] Deleted | Cindy Joyac | 1/5/18 11:25 AM |
| --- | --- | --- |

s

**Page 5: [77] Deleted**      Cindy Joyac      1/5/18 11:25 AM

s

**Page 5: [78] Deleted**      L D      9/21/17 9:53 AM

effect

**Page 5: [78] Deleted**      L D      9/21/17 9:53 AM

effect

**Page 5: [79] Deleted**      L D      9/21/17 9:55 AM

Appendix

**Page 5: [79] Deleted**      L D      9/21/17 9:55 AM

Appendix

**Page 5: [79] Deleted**      L D      9/21/17 9:55 AM

Appendix

**Page 5: [80] Moved from page 5 (Move #4)**      L D      1/17/18 12:11 PM

Statistical models of soil P compounds were adjusted considering variables as outcome measures in decimal units, where 1 = 100%.

**Page 5: [81] Deleted**      L D      1/17/18 12:11 PM

Based on the Wald-type chi-square test, we verified the significance of factor effects significance. Where any factors or interactions effect were was detected, we excluded factors and interactioninteractionsthem of the model, and used only and only used the covariate as model moderator, to test covariate effectthe overall results. of covariate effects overall results. A likelihood ratio test (LRT) was used to test the significance of excluding moderators. Statistical models of soil P

**Page 5: [82] Moved to page 5 (Move #4)**      L D      1/17/18 12:11 PM

Statistical models of soil P compoundfunctional groupscompounds were adjusted considering variables as outcome measures in decimal units, where 1 = 100 %.

**Page 5: [83] Deleted**      L D      9/21/17 4:58 PM

functional groupscompounds were adjusted considering variables as outcome measures in decimal units, where 1 = 100 %.

**Page 5: [84] Deleted**      Cindy Joyac      1/5/18 12:31 PM

o

**Page 5: [84] Deleted**      Cindy Joyac      1/5/18 12:31 PM

o

**Page 5: [85] Deleted**      Cindy Joyac      1/5/18 12:31 PM

the

| Page 5: [85] Deleted | Cindy Joyac | 1/5/18 12:31 PM |

the

| Page 6: [86] Deleted | L D | 1/18/18 9:35 AM |

| Page 6: [86] Deleted | L D | 1/18/18 9:35 AM |

| Page 6: [87] Formatted | L D | 12/6/17 3:17 PM |

Font color: Red, Highlight

| Page 6: [88] Formatted | L D | 1/16/18 8:18 PM |

Superscript

| Page 6: [89] Formatted | Maire, Vincent | 1/27/18 11:39 AM |

Small caps

| Page 6: [90] Comment [1] | Maire, Vincent | 1/9/18 2:54 PM |

I'm confused with the mg/kg -- % thing when analyzing the variation of P compounds. First, regarding %, do we consider the perc to total P or to total Pi or Po. Second, do we consider P compounds only with % across analyses ? For instance, for the variation partitioning and the path analysis, I have the feeling that we use the mg/kg unit, do we?
It would be important to be consistent with the unit to use across analyses.
Finally, it would be important to have an argument to focus either on % or mg/kg unit.

| Page 6: [91] Comment [2] | L D | 1/16/18 5:50 PM |

    i)      Pi or Po mg kg-1: total P amount in NaOH EDTA extract
    ii)     Pi or Po %: percentage to total NaOH EDTA P
    iii)    organic P compounds - percentage to total Pi or Po
    iv)    total P mg kg-1: soil total P in mg kg-1 after acid extraction and digestion

| Page 6: [92] Formatted | Maire, Vincent | 1/27/18 11:39 AM |

French (Canada)

| Page 6: [93] Deleted | Cindy Joyac | 1/5/18 10:30 AM |

,

| Page 6: [93] Deleted | Cindy Joyac | 1/5/18 10:30 AM |

,

| Page 6: [93] Deleted | Cindy Joyac | 1/5/18 10:30 AM |

,

| Page 6: [93] Deleted | Cindy Joyac | 1/5/18 10:30 AM |

,

| Page 6: [93] Deleted | Cindy Joyac | 1/5/18 10:30 AM |

,

| Page 6: [94] Deleted | Cindy Joyac | 1/5/18 10:30 AM |
|---|---|---|

,

| Page 6: [94] Deleted | Cindy Joyac | 1/5/18 10:30 AM |
|---|---|---|

,

| Page 6: [95] Formatted | Maire, Vincent | 1/27/18 11:39 AM |
|---|---|---|

French (Canada)

| Page 6: [96] Deleted | Cindy Joyac | 1/5/18 10:30 AM |
|---|---|---|

,

| Page 6: [96] Deleted | Cindy Joyac | 1/5/18 10:30 AM |
|---|---|---|

,

| Page 6: [96] Deleted | Cindy Joyac | 1/5/18 10:30 AM |
|---|---|---|

,

| Page 6: [97] Deleted | L D | 9/8/17 12:06 PM |
|---|---|---|

n=1;

| Page 6: [97] Deleted | L D | 9/8/17 12:06 PM |
|---|---|---|

n=1;

| Page 6: [98] Deleted | Cindy Joyac | 1/5/18 10:30 AM |
|---|---|---|

,

| Page 6: [98] Deleted | Cindy Joyac | 1/5/18 10:30 AM |
|---|---|---|

,

| Page 6: [99] Deleted | Cindy Joyac | 1/5/18 12:33 PM |
|---|---|---|

on

| Page 6: [99] Deleted | Cindy Joyac | 1/5/18 12:33 PM |
|---|---|---|

on

| Page 6: [100] Deleted | Cindy Joyac | 1/5/18 12:33 PM |
|---|---|---|

didn't

| Page 6: [100] Deleted | Cindy Joyac | 1/5/18 12:33 PM |
|---|---|---|

didn't

| Page 6: [101] Deleted | Cindy Joyac | 1/5/18 12:33 PM |
|---|---|---|

,

| Page 6: [101] Deleted | Cindy Joyac | 1/5/18 12:33 PM |
|---|---|---|

,

| Page 6: [102] Deleted | Cindy Joyac | 1/5/18 12:33 PM |

selected

| Page 6: [103] Deleted | Cindy Joyac | 1/5/18 12:33 PM |

,

| Page 7: [104] Deleted | L D | 9/21/17 7:00 PM |

When these values were normalized as a percentage of NaOH EDTA total P, they showed a contrasting behavior.

| Page 7: [105] Deleted | L D | 12/22/17 10:31 AM |

concentration

| Page 7: [106] Deleted | L D | 12/22/17 10:31 AM |

concentration

| Page 7: [107] Deleted | Cindy Joyac | 1/5/18 12:37 PM |

as

| Page 7: [108] Deleted | L D | 1/27/18 11:39 AM |
|---|---|---|

pH

| Page 7: [108] Deleted | L D | 1/27/18 11:39 AM |
|---|---|---|

pH

| Page 7: [108] Deleted | L D | 1/27/18 11:39 AM |
|---|---|---|

pH

| Page 7: [109] Deleted | Cindy Joyac | 1/5/18 12:37 PM |
|---|---|---|

ver

| Page 7: [109] Deleted | Cindy Joyac | 1/5/18 12:37 PM |
|---|---|---|

ver

| Page 7: [110] Deleted | L D | 1/27/18 11:39 AM |
|---|---|---|

S

| Page 7: [110] Deleted | L D | 1/27/18 11:39 AM |
|---|---|---|

S

| Page 7: [111] Deleted | L D | 12/22/17 10:32 AM |
|---|---|---|

concentration

| Page 7: [111] Deleted | L D | 12/22/17 10:32 AM |
|---|---|---|

concentration

| Page 7: [111] Deleted | L D | 12/22/17 10:32 AM |
|---|---|---|

concentration

| Page 7: [111] Deleted | L D | 12/22/17 10:32 AM |
|---|---|---|

concentration

| Page 7: [111] Deleted | L D | 12/22/17 10:32 AM |
|---|---|---|

concentration

| Page 7: [112] Deleted | Cindy Joyac | 1/5/18 12:39 PM |
|---|---|---|

no

| Page 7: [112] Deleted | Cindy Joyac | 1/5/18 12:39 PM |
|---|---|---|

no

| Page 7: [112] Deleted | Cindy Joyac | 1/5/18 12:39 PM |
|---|---|---|

no

| Page 7: [112] Deleted | Cindy Joyac | 1/5/18 12:39 PM |
|---|---|---|

no

| Page 7: [113] Deleted | L D | 9/22/17 7:39 PM |
|---|---|---|

This response resulted in the increase of diester-to-monoester ratio as pH decreased, and changes occurred in greater intensity in organic layers. There was no pH effect on phosphonates concentration.

| Page 7: [114] Formatted | L D | 9/22/17 7:35 PM |
|---|---|---|

Highlight

| Page 7: [115] Deleted | L D | 1/27/18 11:39 AM |
|---|---|---|

| Page 7: [115] Deleted | L D | 1/27/18 11:39 AM |
|---|---|---|

| Page 7: [116] Deleted | L D | 12/22/17 10:34 AM |
|---|---|---|

(% of total NaOH EDTA P) concentrations

| Page 7: [116] Deleted | L D | 12/22/17 10:34 AM |
|---|---|---|

(% of total NaOH EDTA P) concentrations

| Page 7: [117] Deleted | L D | 12/22/17 10:34 AM |
|---|---|---|

| Page 7: [117] Deleted | L D | 12/22/17 10:34 AM |
|---|---|---|

| Page 7: [118] Deleted | L D | 1/27/18 11:39 AM |
|---|---|---|

| Page 7: [118] Deleted | L D | 1/27/18 11:39 AM |
|---|---|---|

| Page 7: [119] Deleted | L D | 12/22/17 10:42 AM |
|---|---|---|

Inorganic P concentration (% of total NaOH EDTA P)

| Page 7: [119] Deleted | L D | 12/22/17 10:42 AM |
|---|---|---|

Inorganic P concentration (% of total NaOH EDTA P)

| Page 7: [120] Deleted | L D | 12/22/17 10:35 AM |
|---|---|---|

concentrations (% of total NaOH EDTA P)

| Page 7: [120] Deleted | L D | 12/22/17 10:35 AM |
|---|---|---|

concentrations (% of total NaOH EDTA P)

| Page 7: [120] Deleted | L D | 12/22/17 10:35 AM |
|---|---|---|

concentrations (% of total NaOH EDTA P)

| Page 7: [121] Deleted | L D | 12/22/17 10:43 AM |
|---|---|---|

| Page 7: [121] Deleted | L D | 12/22/17 10:43 AM |

| Page 7: [121] Deleted | L D | 12/22/17 10:43 AM |

| Page 7: [121] Deleted | L D | 12/22/17 10:43 AM |

| Page 7: [121] Deleted | L D | 12/22/17 10:43 AM |

| Page 7: [121] Deleted | L D | 12/22/17 10:43 AM |

| Page 7: [122] Deleted | L D | 1/27/18 11:39 AM |

)

| Page 7: [122] Deleted | L D | 1/27/18 11:39 AM |

)

| Page 7: [122] Deleted | L D | 1/27/18 11:39 AM |

)

[revised manuscript text omitted]

available

available

| | | |
|---|---|---|
| **Page 9: [152] Deleted** | **L D** | **1/27/18 11:39 AM** |

S

| | | |
|---|---|---|
| **Page 9: [152] Deleted** | **L D** | **1/27/18 11:39 AM** |

S

| | | |
|---|---|---|
| **Page 9: [153] Deleted** | **Cindy Joyac** | **1/5/18 1:11 PM** |

general

| | | |
|---|---|---|
| **Page 9: [153] Deleted** | **Cindy Joyac** | **1/5/18 1:11 PM** |

general

| | | |
|---|---|---|
| **Page 9: [154] Deleted** | **L D** | **12/22/17 11:12 AM** |

| | | |
|---|---|---|
| **Page 9: [154] Deleted** | **L D** | **12/22/17 11:12 AM** |

| | | |
|---|---|---|
| **Page 9: [154] Deleted** | **L D** | **12/22/17 11:12 AM** |

| | | |
|---|---|---|
| **Page 9: [155] Formatted** | **L D** | **12/20/17 12:24 PM** |

Not Highlight

| | | |
|---|---|---|
| **Page 9: [156] Deleted** | **L D** | **12/20/17 12:16 PM** |

59%

| | | |
|---|---|---|
| **Page 9: [156] Deleted** | **L D** | **12/20/17 12:16 PM** |

59%

| | | |
|---|---|---|
| **Page 9: [156] Deleted** | **L D** | **12/20/17 12:16 PM** |

59%

| | | |
|---|---|---|
| **Page 9: [157] Deleted** | **L D** | **9/23/17 11:47 AM** |

was

| | | |
|---|---|---|
| **Page 9: [157] Deleted** | **L D** | **9/23/17 11:47 AM** |

was

| | | |
|---|---|---|
| **Page 9: [157] Deleted** | **L D** | **9/23/17 11:47 AM** |

was

| | | |
|---|---|---|
| **Page 9: [158] Deleted** | **L D** | **9/23/17 11:47 AM** |

and monoesters concentration

| | | |
|---|---|---|
| **Page 9: [158] Deleted** | **L D** | **9/23/17 11:47 AM** |

and monoesters concentration

| Page 9: [159] Formatted | L D | 12/20/17 12:24 PM |
|---|---|---|

Not Highlight

| Page 9: [160] Deleted | L D | 12/20/17 12:21 PM |
|---|---|---|

the remaining being explained at lower extent by weathering, climatic variables, and their combined effects.

| Page 9: [161] Deleted | L D | 9/23/17 11:52 AM |
|---|---|---|

Diesters concentration had the greatest amount of variation explained by models achieving 78% of the total variation, being mostly explained by soil variables and combined effects of soil and weathering. The diester-to-monoester ratio had 50% of total variance mostly explained by soil and weathering.

| Page 9: [162] Formatted | L D | 12/13/17 9:09 PM |
|---|---|---|

Highlight

| Page 9: [162] Formatted | L D | 12/13/17 9:09 PM |
|---|---|---|

Highlight

| Page 9: [162] Formatted | L D | 12/13/17 9:09 PM |
|---|---|---|

Highlight

| Page 9: [163] Formatted | L D | 12/20/17 12:24 PM |
|---|---|---|

Highlight

| Page 9: [163] Formatted | L D | 12/20/17 12:24 PM |
|---|---|---|

Highlight

| Page 9: [164] Deleted | L D | 1/16/18 3:42 PM |
|---|---|---|

ces

| Page 9: [164] Deleted | L D | 1/16/18 3:42 PM |
|---|---|---|

ces

| Page 9: [165] Deleted | L D | 10/27/17 8:09 PM |
|---|---|---|

| Page 9: [165] Deleted | L D | 10/27/17 8:09 PM |
|---|---|---|

| Page 9: [166] Formatted | L D | 1/27/18 11:39 AM |
|---|---|---|

English (US)

| Page 9: [166] Formatted | L D | 1/27/18 11:39 AM |
|---|---|---|

English (US)

| Page 9: [166] Formatted | L D | 1/27/18 11:39 AM |
|---|---|---|

English (US)

**Page 9: [167] Deleted**      **L D**      **1/16/18 3:43 PM**

, but when it was present, the models were consistently far weaker (p<0.0001; Supplementary additional Appendices S10.1 and S10.3; S11.1 to S11.3)

**Page 9: [168] Formatted**      **L D**      **1/27/18 11:39 AM**

English (US)

**Page 9: [168] Formatted**      **L D**      **1/27/18 11:39 AM**

English (US)

**Page 9: [169] Formatted**      **L D**      **1/27/18 11:39 AM**

English (US)

**Page 9: [170] Deleted**      **L D**      **12/20/17 12:48 PM**

The most parsimonious path analysis model explained 44% of the diester-to-monoester ratio variation, and the causal effects are described in the following. There was no significance on models tested for the soil pyrophosphate + polyphosphate to orthophosphate ratio (chi-square statistics, p<0.05); hence they are not presented here (see Supplementary additional Appendix S11).

**Page 9: [171] Deleted**      **L D**      **1/16/18 3:45 PM**

**Page 9: [171] Deleted**      **L D**      **1/16/18 3:45 PM**

**Page 9: [171] Deleted**      **L D**      **1/16/18 3:45 PM**

**Page 9: [171] Deleted**      **L D**      **1/16/18 3:45 PM**

**Page 9: [172] Deleted**      **L D**      **1/16/18 3:46 PM**

;, and the causal effects are described in the following (Figure 7). Climate and soil weathering drivers were independently related to soil variables (total P, pH, clay, and total soil C), and soil varaibles were considered as direct effets in the model.

**Page 9: [173] Deleted**      **L D**      **12/20/17 12:34 PM**

The parent rock was used as latent variable, which was fixed by pH in this model.

**Page 9: [173] Deleted**      **L D**      **12/20/17 12:34 PM**

The parent rock was used as latent variable, which was fixed by pH in this model.

**Page 9: [174] Formatted**      **L D**      **1/16/18 9:29 PM**

Not Highlight

**Page 9: [174] Formatted**      **L D**      **1/16/18 9:29 PM**

Not Highlight

| Page 9: [174] Formatted | L D | 1/16/18 9:29 PM |
|---|---|---|

Not Highlight

| Page 9: [174] Formatted | L D | 1/16/18 9:29 PM |
|---|---|---|

Not Highlight

| Page 9: [174] Formatted | L D | 1/16/18 9:29 PM |
|---|---|---|

Not Highlight

| Page 9: [174] Formatted | L D | 1/16/18 9:29 PM |
|---|---|---|

Not Highlight

| Page 9: [174] Formatted | L D | 1/16/18 9:29 PM |
|---|---|---|

Not Highlight

| Page 9: [174] Formatted | L D | 1/16/18 9:29 PM |
|---|---|---|

Not Highlight

| Page 9: [174] Formatted | L D | 1/16/18 9:29 PM |
|---|---|---|

Not Highlight

| Page 9: [174] Formatted | L D | 1/16/18 9:29 PM |
|---|---|---|

Not Highlight

| Page 9: [174] Formatted | L D | 1/16/18 9:29 PM |
|---|---|---|

Not Highlight

| Page 9: [174] Formatted | L D | 1/16/18 9:29 PM |
|---|---|---|

Not Highlight

| Page 9: [174] Formatted | L D | 1/16/18 9:29 PM |
|---|---|---|

Not Highlight

| Page 9: [174] Formatted | L D | 1/16/18 9:29 PM |
|---|---|---|

Not Highlight

| Page 9: [174] Formatted | L D | 1/16/18 9:29 PM |
|---|---|---|

Not Highlight

| Page 10: [175] Deleted | L D | 1/16/18 9:37 PM |
|---|---|---|

that greater mean annual precipitation promoted soil weathering, and negatively influenced reducedthe soil total P, pH and clay

| Page 10: [176] Formatted | L D | 1/16/18 8:38 PM |
|---|---|---|

Highlight

| Page 10: [176] Formatted | L D | 1/16/18 8:38 PM |
|---|---|---|

Highlight

| Page 10: [176] Formatted | L D | 1/16/18 8:38 PM |
|---|---|---|

Highlight

| Page 10: [177] Deleted | L D | 12/20/17 5:29 PM |
|---|---|---|

,. The Mmean annual temperature was positively related to

| Page 10: [178] Formatted | L D | 1/16/18 8:38 PM |
|---|---|---|

Highlight

| Page 10: [178] Formatted | L D | 1/16/18 8:38 PM |
|---|---|---|

Highlight

| Page 10: [179] Deleted | L D | 1/27/18 11:39 AM |
|---|---|---|

 soil pH andaffected the clay concentration

| Page 10: [180] Formatted | L D | 1/16/18 8:38 PM |
|---|---|---|

Highlight

| Page 10: [180] Formatted | L D | 1/16/18 8:38 PM |
|---|---|---|

Highlight

| Page 10: [181] Formatted | L D | 12/20/17 5:40 PM |
|---|---|---|

Not Highlight

| Page 10: [182] Deleted | L D | 1/27/18 11:39 AM |
|---|---|---|

oil. Soil weathering was negatively related to reducedthe soil total P and pH, and positively

| Page 10: [183] Formatted | L D | 1/16/18 8:38 PM |
|---|---|---|

Highlight

| Page 10: [183] Formatted | L D | 1/16/18 8:38 PM |
|---|---|---|

Highlight

| Page 10: [183] Formatted | L D | 1/16/18 8:38 PM |
|---|---|---|

Highlight

| Page 10: [183] Formatted | L D | 1/16/18 8:38 PM |
|---|---|---|

Highlight

| Page 10: [184] Deleted | L D | 12/20/17 5:40 PM |
|---|---|---|

increasedrelated to the soil clay and total C. The parent rock was strongly and positively related to soil total P and pH, and to a lesser extent clay concentration. It's important to note that when parent rock was absent the models were consistently far weaker (p<0.0001; Supplementary Appendix S4); hence they are not presented. There were also significant direct and positive effects among all soil

| Page 10: [185] Formatted | L D | 1/16/18 8:38 PM |
|---|---|---|

Highlight

Highlight

Highlight

Highlight

Highlight

 soil total P that was positively related to ofexcept for the soil total C, and soil total C

Highlight

Highlight

Highlight

that was negatively related to soil pH, which were negatively related. Inositol was negatively affected by precipitation, and soil weathering, but positively affected by the soil C. In contrast, DNA was positively affected by precipitation, soil weathering, soil C and clay, and it was negatively affected by

Highlight

Highlight

[revised manuscript text omitted]

Figure

| Page 15: [277] Deleted | L D | 10/27/17 8:15 PM |
|---|---|---|

7C),

| Page 15: [277] Deleted | L D | 10/27/17 8:15 PM |
|---|---|---|

7C),

| Page 15: [277] Deleted | L D | 10/27/17 8:15 PM |

7C),

| Page 15: [278] Deleted | Cindy Joyac | 1/5/18 10:34 AM |

,

| Page 15: [278] Deleted | Cindy Joyac | 1/5/18 10:34 AM |

,

| Page 15: [278] Deleted | Cindy Joyac | 1/5/18 10:34 AM |

,

| Page 15: [278] Deleted | Cindy Joyac | 1/5/18 10:34 AM |

,

| Page 15: [278] Deleted | Cindy Joyac | 1/5/18 10:34 AM |

,

| Page 15: [278] Deleted | Cindy Joyac | 1/5/18 10:34 AM |

,

| Page 15: [278] Deleted | Cindy Joyac | 1/5/18 10:34 AM |

,

| Page 15: [279] Deleted | L D | 12/22/17 11:50 AM |

concentration

| Page 15: [279] Deleted | L D | 12/22/17 11:50 AM |

concentration

| Page 15: [279] Deleted | L D | 12/22/17 11:50 AM |

concentration

| Page 15: [279] Deleted | L D | 12/22/17 11:50 AM |

concentration

| Page 15: [279] Deleted | L D | 12/22/17 11:50 AM |

concentration

| Page 15: [280] Deleted | Cindy Joyac | 1/5/18 10:27 AM |

predominate

| Page 15: [280] Deleted | Cindy Joyac | 1/5/18 10:27 AM |

predominate

| Page 15: [280] Deleted | Cindy Joyac | 1/5/18 10:27 AM |

predominate

| Page 15: [281] Deleted | Cindy Joyac | 1/5/18 10:34 AM |
| --- | --- | --- |

,

| Page 15: [282] Deleted | L D | 10/27/17 8:16 PM |
| --- | --- | --- |

Figure

| Page 15: [283] Deleted | L D | 10/27/17 8:16 PM |
| --- | --- | --- |

Figure

| Page 15: [284] Deleted | Cindy Joyac | 1/5/18 2:12 PM |
| --- | --- | --- |

could

| Page 15: [285] Deleted | Cindy Joyac | 1/5/18 2:13 PM |
| --- | --- | --- |

o

| Page 15: [286] Moved from page 4 (Move #1) | L D | 9/21/17 9:02 AM |
| --- | --- | --- |

Finally, we expect for future researches to provide results of as much many soil P species as they can find rather than functional groups only (i.e., diesters and monoesters), even when species concentrations are low (and describe when main soil species are not detected), what which may enable future analysisanalyseis to avoid possible confounding effects of P compounds inside functional groups (e.g., inositol phosphates and monoesters), and to make a more precise correction for potential degraded peaks occurred occurring during the alkaline extraction and reading process. We also urge researchers to determine variances or standard errors for soils with distinctive properties. Then, as stated by Stewart (2010), future analysis cananalyseis couldan use the different amounts of information that provided by studies of different sizes scopes and different quality present in a meta-analyticanalytical approach.

| Page 15: [287] Deleted | Cindy Joyac | 1/5/18 2:15 PM |
| --- | --- | --- |

for

| Page 15: [287] Deleted | Cindy Joyac | 1/5/18 2:15 PM |
|---|---|---|

for

| Page 15: [288] Deleted | Cindy Joyac | 1/5/18 2:15 PM |
|---|---|---|

what

| Page 15: [288] Deleted | Cindy Joyac | 1/5/18 2:15 PM |
|---|---|---|

what

| Page 15: [289] Deleted | Cindy Joyac | 1/5/18 2:16 PM |
|---|---|---|

'

| Page 15: [289] Deleted | Cindy Joyac | 1/5/18 2:16 PM |
|---|---|---|

'

| Page 15: [290] Deleted | Cindy Joyac | 1/5/18 2:16 PM |
|---|---|---|

i

| Page 15: [290] Deleted | Cindy Joyac | 1/5/18 2:16 PM |
|---|---|---|

i

| Page 15: [290] Deleted | Cindy Joyac | 1/5/18 2:16 PM |
|---|---|---|

i

| Page 15: [290] Deleted | Cindy Joyac | 1/5/18 2:16 PM |
|---|---|---|

i

| Page 15: [291] Deleted | Cindy Joyac | 1/5/18 2:17 PM |
|---|---|---|

sizes

| Page 15: [291] Deleted | Cindy Joyac | 1/5/18 2:17 PM |
|---|---|---|

sizes

| Page 15: [291] Deleted | Cindy Joyac | 1/5/18 2:17 PM |
|---|---|---|

sizes

| Page 17: [292] Formatted | Maire, Vincent | 1/27/18 11:39 AM |
|---|---|---|

Small caps

| Page 17: [293] Formatted | L D | 12/26/17 4:12 PM |
|---|---|---|

Font:Bold

| Page 17: [294] Formatted | L D | 12/26/17 4:12 PM |
|---|---|---|

Font:Bold

| Page 17: [295] Formatted | L D | 1/27/18 11:39 AM |
|---|---|---|

English (US)

| Page 17: [296] Formatted | L D | 12/26/17 4:12 PM |
|---|---|---|

Font:Bold

| Page 17: [297] Formatted | L D | 12/26/17 4:12 PM |
|---|---|---|

Font:Bold

[revised manuscript text omitted]

mineral layer, DNA = 27.8 − 3.63 pH, $r^2$ = 0.19 (n=64); organic layer, monoesters = 22.0+ 10.3 pH, $r^2$ =0.56; mineral layer, diesters

| Page 27: [320] Deleted | L D | 1/27/18 11:39 AM |

mineral layer, DNA = 27.8 − 3.63 pH, $r^2$ = 0.19 (n=64); organic layer, monoesters = 22.0+ 10.3 pH, $r^2$ =0.56; mineral layer, diesters

| Page 27: [320] Deleted | L D | 1/27/18 11:39 AM |

mineral layer, DNA = 27.8 − 3.63 pH, $r^2$ = 0.19 (n=64); organic layer, monoesters = 22.0+ 10.3 pH, $r^2$ =0.56; mineral layer, diesters

| Page 27: [320] Deleted | L D | 1/27/18 11:39 AM |

mineral layer, DNA = 27.8 − 3.63 pH, $r^2$ = 0.19 (n=64); organic layer, monoesters = 22.0+ 10.3 pH, $r^2$ =0.56; mineral layer, diesters

| Page 27: [320] Deleted | L D | 1/27/18 11:39 AM |

mineral layer, DNA = 27.8 − 3.63 pH, $r^2$ = 0.19 (n=64); organic layer, monoesters = 22.0+ 10.3 pH, $r^2$ =0.56; mineral layer, diesters

| Page 27: [320] Deleted | L D | 1/27/18 11:39 AM |

mineral layer, DNA = 27.8 − 3.63 pH, $r^2$ = 0.19 (n=64); organic layer, monoesters = 22.0+ 10.3 pH, $r^2$ =0.56; mineral layer, diesters

| Page 27: [320] Deleted | L D | 1/27/18 11:39 AM |
|---|---|---|

mineral layer, DNA = 27.8 – 3.63 pH, $r^2$ = 0.19 (n=64); organic layer, monoesters = 22.0+ 10.3 pH, $r^2$ =0.56; mineral layer, diesters

| Page 27: [321] Deleted | L D | 9/19/17 11:29 AM |
|---|---|---|

80.0

| Page 27: [322] Deleted | L D | 9/19/17 11:31 AM |
|---|---|---|

55; mineral layer, di-to-mono= 44.5 – 6.00 pH, $r^2$ = 0.26;

| Page 27: [323] Deleted | L D | 9/19/17 12:21 PM |
|---|---|---|

mineral layer, monoesters = 97.6 – 7.30 log(total C), $r^2$ = 0.08; organic layer, monoesters = 265.0 – 77.0 log(total C), $r^2$ = 0.59; mineral layer, diesters = 2.86 + 5.82 log(total C), $r^2$ = 0.06; organic layer, diesters = -16.0 + 74.6 log(total C), $r^2$ = 0.60; mineral layer, di-to-mono =0.03 + 0.073 log(total C), $r^2$ = 0.04; organic layer, di-to-mono = -3.71 + 1.64 log(total C), $r^2$ =0.53; mineral layer,

| Page 27: [324] Deleted | L D | 9/19/17 12:26 PM |
|---|---|---|

mineral layer, monoesters = 69.0 + 6.64 log(total P), $r^2$ = 0.10; mineral layer, diesters

| Page 27: [325] Deleted | L D | 9/19/17 12:27 PM |
|---|---|---|

3; mineral layer, di-to-mono = 0.47 – 0.12 log(total P), $r^2$ = 0.14;

| Page 27: [325] Deleted | L D | 9/19/17 12:27 PM |
|---|---|---|

3; mineral layer, di-to-mono = 0.47 – 0.12 log(total P), $r^2$ = 0.14;

| | | |
|---|---|---|
| **Page 27: [326] Deleted** | **L D** | **9/19/17 12:32 PM** |

mineral layer, monoesters = 90.1 –

| | | |
|---|---|---|
| **Page 27: [327] Deleted** | **L D** | **1/27/18 11:39 AM** |

=

| | | |
|---|---|---|
| **Page 27: [327] Deleted** | **L D** | **1/27/18 11:39 AM** |

=

| | | |
|---|---|---|
| **Page 27: [327] Deleted** | **L D** | **1/27/18 11:39 AM** |

=

| | | |
|---|---|---|
| **Page 27: [327] Deleted** | **L D** | **1/27/18 11:39 AM** |

=

| | | |
|---|---|---|
| **Page 27: [327] Deleted** | **L D** | **1/27/18 11:39 AM** |

=

| | | |
|---|---|---|
| **Page 27: [327] Deleted** | **L D** | **1/27/18 11:39 AM** |

=

| | | |
|---|---|---|
| **Page 28: [328] Deleted** | **L D** | **10/27/17 7:07 PM** |

[Figure]

**Figure 4: Relationship between climatic properties and soil inorganic phosphorus (P) composition from soil mineral and organic layers on terrestrial natural ecosystems.**

Regression models (n = 80 mineral layer and n = 20 mineral layer): mineral layer, total $P_i$ (%) = 89.4 − 14.7 log(precipitation), $r^2$ = 0.08; mineral layer, orthophosphate = 127 − 11.7 log(precipitation), $r^2$ = 0.24; mineral layer, pyrophosphate = -28.1 + 11.9 log(precipitation), $r^2$ = 0.24; mineral layer, total $P_i$ (%) = 45.9 - 13.0 log(moisture), $r^2$ = 0.13; mineral layer, orthophosphate = 92.1 − 13.4 log(moisture), $r^2$ = 0.32; mineral layer, pyrophosphate = 7.49 + 13.5 log(moisture), $r^2$ = 0.32.

[Figure]

**Figure 5: Relationship between climatic properties and soil organic phosphorus (P) composition from soil mineral and organic layers on terrestrial natural ecosystems. Regression models (n = 80 mineral layer and n = 20 mineral layer): mineral layer, total $P_o$ (%) = 10.6 + 14.7 log(precipitation), $r^2$ = 0.08; mineral layer, monoesters = 119 – 10.6 log(precipitation), $r^2$ = 0.11; mineral layer, total $P_o$ (%) = 54.2 + 13.1 log(moisture), $r^2$ = 013; mineral layer, monoesters = 87.1 – 10.1 log(moisture), $r^2$ = 0.13; mineral layer, di-to-mono = 0.14 + 0.092 log(moisture), $r^2$ = 0.06.**

[Figure]

Page 28: [329] Formatted

Font:(Default) Arial

Page 28: [329] Formatted

Font:(Default) Arial

Page 28: [330] Deleted          L D          10/27/17 8:05 PM

Page 28: [330] Deleted          L D          10/27/17 8:05 PM

Page 28: [330] Deleted          L D          10/27/17 8:05 PM

Page 28: [330] Deleted          L D          10/27/17 8:05 PM

Page 28: [330] Deleted          L D          10/27/17 8:05 PM

[Figure]

[Figure]

I've checked path analyses and I've got different results with better P-value when parent rock is included. For that, I transform the variable so that the variance of each variable is standardized and get a pvalue <

0.05 (required to meet statistical requirement of the path analysis). In addition, I let the MAT and MAP covaried. You can check the r file and the doc file with final result.

the model.

**Appendices - Soil phosphorus dynamics on terrestrial natural ecosystems**

**Appendix S1 - Dataset**

The dataset of "Soil phosphorus dynamics on terrestrial natural ecosystems" is attached as a supplementary Excel file.

**Appendix S2 – Global biomes according to the Whittaker' diagram**

L D 1/30/2018 12:37 PM

[Figure]

**Figure S2**. The Whittaker' diagram used to determine the main biomes comprised in our dataset.

[Figure]

[Figure]

**Figure S3**. Soil inorganic and organic phosphorus (P) composition in NaOH EDTA extract as influenced by soil sampling depth on terrestrial natural ecosystems. Significant relationships are indicated with regression lines.

L D 1/30/2018 12:37 PM

L D 1/30/2018 12:37 PM

[Figure]

**Figure S4**. Soil inorganic and organic phosphorus (P) composition in NaOH EDTA extract as influenced by latitude on terrestrial natural ecosystems. Significant relationships are indicated with regression lines.

[Figure]

**Figure S5**. Soil inorganic and organic phosphorus (P) composition in NaOH EDTA extract as influenced by soil sampling depth on terrestrial natural ecosystems. Significant relationships are indicated with regression lines.

[Figure]

**Figure S6**. Relationship between edaphic properties and soil organic phosphonates in NaOH EDTA extract from soil mineral and organic layers on terrestrial natural ecosystems. No significant relationships were found.

[Figure]

**Figure S7**. Relationship between climatic properties and soil inorganic phosphorus (P) composition in NaOH EDTA extract from soil mineral and organic layers on terrestrial natural ecosystems. Regression models (n = 80 mineral layer and n = 20 mineral layer): mineral layer, total $P_i$ (%) = 89.4 – 14.7 log(precipitation), $r^2$ =

L D 1/30/2018 12:37 PM

**Appendix S8 – Climatic properties and soil organic phosphorus**

[Figure]

**Figure S8**. Relationship between climatic properties and soil organic phosphorus (P) composition in NaOH EDTA extract from soil mineral and organic layers on

terrestrial natural ecosystems. Regression models: mineral layer, total $P_o$ (%) =
10.6 + 14.7 log(precipitation), $r^2$ = 0.08 (n=80).

**Appendix S9 – Soil weathering stages and poorly crystalline Al and Fe
concentration**

[Figure]

**Soil Weathering Stage**

**Figure S9.** Soil weathering stage relationship with soil poorly crystalline Al and
Fe (n = 49) on terrestrial natural ecosystems. For all three panels p>0.1.

L D 1/30/2018 12:37 PM

L D 1/30/2018 12:37 PM

L D 1/30/2018 12:37 PM

L D 1/30/2018 12:37 PM

**Appendix S10 – Models tested to explore the interdependences between edaphic and climatic variables (path analysis) as the main environmental predictors of soil inorganic and organic P compounds.**

We expected some directionalities in the relationships, based on the literature (theoretical models on Figures S.10.1 and S10.2, top panels).

[Figure]

[Figure]

**Figure S10.1.** Theoretical models set to explore the interdependences between edaphic and climatic variables (path analysis) as the main environmental

L D 1/30/2018 12:37 PM

L D 1/30/2018 12:37 PM

L D 1/30/2018 12:37 PM

L D 1/30/2018 12:37 PM

L D 1/30/2018 12:37 PM

L D 1/30/2018 12:37 PM

L D 1/30/2018 12:37 PM

L D 1/30/2018 12:37 PM

L D 1/30/2018 12:37 PM

predictors of soil inorganic (upper panel) and organic (bottom panel) P compounds.

L D 1/30/2018 12:37 PM

L D 1/30/2018 12:37 PM

L D 1/30/2018 12:37 PM

**Moved up [1]:** .

L D 1/30/2018 12:37 PM

L D 1/30/2018 12:37 PM

).

Final model 1

[Figure]

Figure S4.2 Final model 1 set to explore

climatic variables (path analysis).

[Figure]

Figure S4.3 Theoretical model 2 set to explore the interdependences between edaphic and climatic variables (path analysis).

[Figure]

Figure S4.4 Final model 2 set to explore the interdependences between edaphic and climatic variables (path analysis).

---

## Author Response (AR3)

**Biogeosciences**
**Manuscript bg-2017-307**

**Dear Associate Editor Sönke Zaehle,**

**Thank you for your evaluation of the manuscript. We also thank the reviewers for the great effort they made in reviewing the manuscript, and for the significant improvements they promoted on it. We are sincerely grateful for the comprehensive review of our manuscript, and we valued each comment made. We added acknowledgments to reviewers recognizing their effort.**

**We responded to each comment and revised the manuscript. We made appropriate changes in the manuscript or have provided an explanation where we could not follow exactly what reviewers suggested. To the best of our understanding, no remaining issues are left in this corrected version.**

**The major changes made on the manuscript are briefly described in the following. A new title was introduced including a more specific description of the study. In the introduction, we added the rationale about the limitations of the 31P NMR method. For the Material and Methods, a complementation was made on how the structural equation modeling was developed. For the results, we added the average and range of P extracted by NaOH-EDTA (%) for both mineral and organic layers. Moreover, most changes were made on the discussion. These included adding discussion and several references on the effect of acidifying soils, which may increase the charge of some organic compounds, and thus increase sorption. Discussion on why inositol hexakisphosphates proportions increase as soils get more weathered in non-tropical environments, but no inositol hexakisphosphates have been found in very weathered soils of tropical environments was added. We also added discussion on how vegetation change during pedogenesis and how it affects soil P composition. Ultimately, we discussed future research priorities such as disentangling confounding effects among soil biotic and abiotic components, climate and vegetation; and coupling $^{31}$P NMR with other important techniques to better understand P cycling and composition in terrestrial ecosystems. We excluded the terms "dynamics" and "terrestrial" throughout the manuscript. Finally, a native person with an affinity to the topic reviewed the whole manuscript again for precision and clarity of terminology.**

**Responses to the reviewers' comments are described below under "Response". Changes made on the manuscript were all based on these responses. To facilitate the evaluation, we left the reviewers comments before each response.**

**Responses made by the authors are in bold and indicated with the term "Response". After the responses to the reviewers comments one can find the manuscript with all changes marked.**

**REPORT 1**

For the most part, the manuscript has been significantly improved compared to the original submission. In particular, the authors have constrained the studies used to avoid methodological differences, and have focused on specific P compounds to minimize differences among research groups (e.g. with respect to correcting for degradation), which could hamper the type of global comparisons they are trying to make. I still think the studies they have included are too heavily skewed to studies by a single author (Turner) and a single location (New Zealand). However, that is partially due to issues with analyses by some studies by other groups or from other locations. Hopefully, future studies will avoid these problems, allowing this type of analysis to be expanded in time. But for now, this is an interesting first attempt at this type of analysis.

There are still some problems that will need to be addressed before this is publishable. The majority of these fall more into minor or moderate revisions, as indicated below. However, the discussion will need some substantial revision, because many parts of it, as indicated below, are based merely on speculation, and are not grounded in fact or even in the authors' own results.

1. Writing quality: for the most part, the quality of English has improved. However, there are still some problems with some sections. These are specifically addressed below, but I still recommend that the authors have the revised version read by a native English speaker who is familiar with the research topic. There are still problems with the writing in that the authors make sweeping statements of fact, without citing any references to support these statements. This is particularly true in the introduction, and specific instances are noted below.

2. Title: The current title isn't very informative or very well written (e.g. "soil" by definition is "terrestrial", so including both is redundant). I suggest: "A meta-analysis of ecosystem properties influencing soil phosphorus dynamics in natural ecosystems".

**Response: The "meta-analysis" term is being avoided because no meta-analytic methods were used to statistically analyze the data. These statistical methods were not used due to the lack of error measurements on the compiled data (e.g., variance or standard error). The title was changed to a more informative version, as suggested by the reviewer: "Environmental drivers of soil phosphorus composition in natural ecosystems". We used "compounds" instead of "dynamics" as suggested by the Reviewer #2.**

3. Abstract: Although the main body of the paper appears to have been revised with the assistance of a native English speaker, the abstract appears to have been overlooked.

Line 7: “phosphorus (P) compounds can be modified by distinctive ecosystem properties” Using the term “modifies” implies that the ecosystem properties directly change P forms. However, most of the changes are indirect. For example, pH can change the soil environment to influence sorption of P forms, or it can affect the organisms that produce P forms, or that produce enzymes to mineralize P forms. But pH itself isn’t necessarily changing P forms, except potentially alter the charge of the P form. As such, I think it would be best to change this to “phosphorus (P) compounds can be influenced by distinctive ecosystem properties”.

**Response: OK. “influenced by” was used instead of “modifies”.**

Line 8: “soil P dynamics on terrestrial natural ecosystems, relating its organic “should be “soil P dynamics in natural ecosystems, relating organic “

**Response: Changes were made.**

Line 11: “determined the soil P composition” should be “determined soil P composition”

**Response: Changes were made.**

Lines 11-12: “nuclear magnetic resonance of soils extracted with NaOH EDTA” should be “nuclear magnetic resonance of NaOH-EDTA extracts of soils”

**Response: Changes were made.**

Line 12: “models were used to better understand the soil P” should be “models were used to better understand the factors influencing soil P”

**Response: Changes were made. We used “environmental properties” instead of “factors”.**

Line 13: “relationships, soil P compounds had similar overall behaviors on mineral and organic layers but with different slopes”. This analysis does not give any information about the behavior of soil P compounds; instead, it shows the relationship of soil P compounds with various factors. As such, this should be “relationships, trends for soil P compounds were similar for mineral and organic layers but with different slopes”.

**Response: OK.**

Line 16: “Soil, particularly” should be “Soil factors, particularly”; “ratio” should be “ratios”

**Response: Changes were made. We used “properties” instead of “factors”.**

Line 18: “soil P composition on terrestrial natural ecosystems” should be “soil P composition in natural ecosystems”

**Response: Changes were made.**

Lines 19-20: “and after altogether with plant and microbe coexistence” I do not understand what the authors are trying to say here; something seems to be missing.

**Response: It was changed to “and after altogether through plant and microbe P cycling”.**

4. Introduction

p. 2, line 4: “derives essentially” should be “derives predominantly”; “as an eolian deposit” should be “as eolian deposits”. However, I disagree with this statement, because it is only relevant for younger soils (e.g. Chadwick et al. 1999 Nature 397:491-497). And because this is a statement of fact, a reference needs to be cited here.

**Response: We added the term “young” before “ecosystem” to fulfill the statement according to the Chadwick et al. (1999) findings and the reference was added: “Phosphorus input into a young ecosystem derives predominantly from the weathering of rocks, with little input as eolian deposits”.**

p. 2, lines 5-6: “Phosphorus goes back to soil as organic materials (Noack et al. 2012” but not necessarily as organic P; Noack et al. showed that much of the P in plant residues was orthophosphate. I think the authors need to explicitly state that here, that “organic materials” are not primarily composed of organic P forms.

**Response: The sentence was changed to “Phosphorus goes back to soil as organic materials, which are composed by organic and inorganic P compounds”.**

p. 2, lines 7-9: “Each new cycle...primary minerals” Please cite a reference to support this statement of fact, because I am not sure that it is in fact true.

**Response: McDowell et al. (2007) was added to support the following sentence “Each new cycle loop leads to more complex and less bioavailable P compounds”, and “Walker and Syers (1976) was used to “ultimately seriously limiting ecosystem productivity in the absence of ‘fresh’ P input as primary minerals.”**

p. 2, line 10: Jenny (1941) described these as the five factors of soil formation, not ecosystem functioning. Please cite another reference to support using this term in this way.

**Response: “soil formation” was used instead of “ecosystem functioning” in this sentence and across the manuscript. Therefore, Jenny (1941) was maintained for this statement.**

p. 2, line 14:”main Po compounds are” should be “main Po compound categories are”, because monoesters, diesters, etc. are compound categories, not forms.

**Response: Changes were made.**

p. 2, lines 15-16: “orthophosphate monoester” should be “orthophosphate monoesters”, “orthophosphate diester” should be “orthophosphate diesters” and “phosphonate” should be “phosphonates”, because these broad compound classes contain a number of different P compounds.

**Response: Changes were made across the manuscript.**

p. 2, line 18: “used by” should be “directly taken up by”, because all P compounds can be used by plants and microbes after transformation, just not directly taken up by them.

**Response: Changes were made.**

p. 2, lines 17-19: “Specific phosphatase...and microbes” and “Obviously...optimum” Please cite references to support these statements of fact.

**Response: Changes were made. Turner (2008a) was used for “Specific phosphatase...and microbes” and Frankenberger and Johanson (1982) was used for “Obviously...optimum”.**

p. 2, lines 19-20: “Phosphomonoesterase is more active in acidic soils while phosphodiesterase is optimized in basic soils (Turner and Haygarth 2005)” NO! It simply is not possible to make such a broad, sweeping statement based on a single study of pasture soils from England. Please rewrite.

**Response: The sentence was excluded to avoid the broadening of the statement, and also because the rationale was supported by the previous sentence.**

p. 2, lines 28-29: “As soil ages...colimitation intermediate stage” Please cite a reference to support this statement of fact.

**Response: Walker and Syers 1976 and Turner and Condron (2013) were used to supporting the statement.**

p. 3, line 4: “precipitations” should be “precipitation”.

**Response: Change was made.**

p. 3, lines 9-14: I think “parent rock” should be changed to “parent material”, because soils can form on materials other than a specific type of rock (e.g. sand, glacial till).

**Response: “parent material” was used instead of “parent rock” in all instances across the manuscript.**

p. 3, line 16: “Soil P composition has been studied in soils from ecosystems worldwide” should be “Soil P composition has been studied in ecosystems worldwide”

**Response: Changes were made.**

p. 3, line 17: “(NMR) was a widely used method” should be (NMR) is a widely-used method”, because to the best of my knowledge it is still being used.

**Response: Changes were made.**

p. 3, line 19: “NaOH and chelating agent EDTA” should be “NaOH combined with the chelating agent EDTA”

**Response: Changes were made.**

p. 3, line 20 (and elsewhere throughout the manuscript): “NaOH EDTA” should be “NaOH-EDTA”.

**Response: It was changed throughout the manuscript.**

p. 3, lines 22-24: “The NaOH EDTA...(Turner and Blackwell 2013)” This sentence does not make any sense as written. How does this demonstrate that NaOH-EDTA is quantitative? Please rewrite and include more references.

**Response: OK. The sentence was changed to “The NaOH-EDTA extraction method is recognized to quantitatively extract P compounds from the soil.”**

p. 3, lines 26-27: “state factors of ecosystem functioning” see previous comment and rewrite.

**Response: Changes were made. “soil formation” was used instead of “ecosystem functioning”.**

p. 3, line 27: “because known responses were obtained from case-specific conditions”. I do not understand what the authors are trying to say here. Please rewrite.

**Response: the sentence was rewritten to “how soil P composition is affected by different state factors of soil formation because most know responses were obtained from specific chronosequences.”**

5. Methods

p. 4, line 5: “that namely estimated an adequate delay time” should be “that used an adequate time”

**Response: Changes were made.**

p. 4, line 15: “only control (unchanged) was used” should be “only control (unchanged) samples were used”

**Response: Changes were made.**

p. 4, lines 23-23: “organic P and its compound” should be “organic P and the compounds”

**Response: Changes were made.**

p. 4, line 24: “NaOH EDTA inorganic P, and its compounds” should be “and NaOH-EDTA inorganic P and the compounds”

**Response: Changes were made.**

p. 4, line 27: “climate characteristic” should be “climate characteristics”, because it refers to both precipitation and temperature.

**Response: Changes were made.**

p. 5, lines 29-30: “how the inorganic and organic P compounds variation” should be “how variations in inorganicand organic P compounds”

**Response: Changes were made.**

p. 6, line 4: “Differently” should be “In contrast”

**Response: Changes were made.**

6. Results:

p. 6, line 13: “McDowell and Steward” should be “McDowell and Stewart”; the name is spelled correctly in the References section.

**Response: Changes were made.**

p. 6, line 20: “And temperate rain forest” Don’t some of the New Zealand sites fall into temperate rain forest? The authors should check to confirm.

**Response: According to Whittaker's diagram we found that some of those plots were classified as intermediates between the temperate rainforest and**

**boreal forest, and tropical rainforest and temperate rainforest. It is being described at the beginning of the third paragraph of the discussion section.**

p. 6, line 25: “were absented” should be “were absent”

**Response: Changes were made.**

p. 6, lines 26-27: “All compiled results...polyphosphate” This sentence repeats information already given (p. 4, lines 22-24), and should be deleted in one of these locations in the text.

**Response: OK. The given sentence was deleted.**

p. 7, line 3: “The compounds proportions” should be “The proportioning of compounds”

**Response: Changes were made.**

p. 7, line 6: “Into the Pi pool” should be “In the Pi pool”; “layers, the orthophosphate” should be “layers, orthophosphate”

**Response: OK. Both were corrected.**

p. 7, lines 12-13: “There was no clay effect on both Pi and Po and their compounds proportions” should be “Clay had no effect on either Pi or Po, or on the proportions of compounds in these categories”

**Response: Changes were made.**

p. 7, line 18: “effect on both Pi and Po” should be “effect on either Pi or Po”

**Response: Changes were made.**

p. 7, line 24: “in both mineral and organic “ should be “in either mineral or organic”

**Response: Changes were made.**

p. 7, lines 30-31: “In is important to note...usually comprises the residual P not recovered by the NaOH EDTA extractant”. This sentence doesn’t make sense to me. Firstly, the x axes in figures 2 and 3 vary depending on the factor (e.g. %, % organic P, log scale mg/kg). Secondly, total P doesn’t just include residual P (which is what “comprises the residual P” means”), it includes both the extracted P and the residual P. This sentence needs to be rewritten.

**Response: The sentence was rewritten as “It is important to note that the reported total P (x-axis on Figures 2 and 3) is the one obtained by digestion and it includes both the extracted P and the residual P.”**

p. 8, line 2: “As the percentage” should be “As a percentage”

**Response: Changes were made.**

p. 8, line 12: “on the soil P composition on terrestrial natural ecosystems” should be “ in the soil P composition in natural ecosystems”

**Response: Changes were made.**

p. 8, line 13: “over total “ should be “on total”

**Response: Changes were made.**

p. 8, line 18 and line 19: “affected the soil age and CP ratio” should be “affected soil age and CP ratios”

**Response: Changes were made.**

p. 8, line 29: “of the ecosystem’s properties” should be “of ecosystem properties”

**Response: Changes were made.**

p. 8, line 31: “and its compound” should be “and its compounds”

**Response: Changes were made.**

p. 9, line 2: “poorly defined” should be “poorly-defined”

**Response: Changes were made.**

p. 9, line 3: “of the ecosystem’s properties” should be “of ecosystem properties”

**Response: Changes were made.**

p. 9, lines 7-8: “phosphonate had most of its variation explained by” should be “most of the variation in phosphonates was explained by”

**Response: Changes were made.**

p. 9, line 19: “by precipitation into the “ should be “by precipitation in the”

**Response: Changes were made.**

p. 9, line 21: “Into the” should be “in the”

**Response: Changes were made.**

p. 9, line 26: “Pyrophosphate had a positive influence of” should be “Pyrophosphate was positively influenced by”

**Response: Changes were made.**

p. 9, line 27: “inositol was negatively” should be “inositol phosphates were negatively”

**Response: Changes were made.**

p. 9, line 29: “Phosphonate was” should be “Phosphonates were”

**Response: Changes were made.**

7. Discussion: Much of this is very poorly done, and relies on speculation rather than data. Major revisions are needed in this part of the manuscript. p. 10, line 2: “compounds respond to” should be “compounds responded to”

**Response: Changes were made.**

p. 10, line 3: “at a wide” should be “on a wide”

**Response: Changes were made.**

p. 10, line 4: “terrestrial nature ecosystems” should be “natural ecosystems”

**Response: Changes were made.**

p. 10, line 7: “persistence on ecosystems” should be “persistence in ecosystems”

**Response: Changes were made.**

p. 10, lines 7-11. This is a very long, confusing sentence; as written, the point that the authors are trying to make with this sentence is not clear. It should be rewritten.

**Response: The phrase has been rewritten in order to clarify it.**

p. 10, line 14: “the decaying degree of C element is lower than the P” I do not understand what the authors are trying to say here.

**Response: OK. The sentences were changed to “As time passes after the onset of pedogenesis, the ecosystem accumulates organic matter up to a maximum, and then starts to decline. Along with this decline, there are also changes in the chemical composition of organic matter, in which the decaying degree of C element is lower than the P, and concomitantly there is an increasingly acidic environment (Walker 1965; Turner and Condron 2013).”**

p. 10, line 15: “resulting in the slowed decomposition of the older soil system” Again, I do not understand what the authors are trying to say here. As currently written, it indicates that the soil system itself is decomposing more slowly. However, soils do not decompose. Do they mean that decomposition is slower in older soil systems? If so, then decomposition of what? Plant material? Or do they mean mineralization of specific P forms? This needs to be rewritten for clarity.

**Response: The sentence “resulting in the slowed decomposition of the older soil system” was excluded because it was not being relevant to the understanding of the manuscript. What we meant was that the organic matter decomposition is slower in older soil systems due to a higher CP ratio and a more acidic soil environment.**

p. 10, lines 16-19: “more weathered soils are remote from the parent material”. Again, this is confusing as written. Do the authors mean physically remote? If so, wouldn’t that depend on the nature of the soil weathering processes, and even on the soils themselves? Or do they mean “substantially changed”? Please rewrite.

**Response: We meant physically remote, but we understood and agreed with the point the reviewer is making. To fulfill that we changed “remote” to “substantially changed”.**

p. 10, line 25: “occluded P increases at the expense of organic P”. This sentence implies that organic P compounds cannot be occluded, which isn’t the case – one of the reasons that inositol phosphates increase with weathering is that they become occluded in the same way that inorganic P compounds become occluded. The inaccuracy of this statement is due to citing a dated reference (Crews et al. 1995) that relied on Hedley fractionation. Please update this statement and cite something that uses P-NMR or other modern techniques.

**Response: The sentence was modified to “In highly weathered soils, occluded P increases through the encapsulation of the organic and inorganic P compounds inside of Fe and Al minerals (McDowell et al., 2007; Turner et al. 2007).”**

p. 10, line 31 and p. 11, line 2: “phosphorus” should be abbreviated as P.

**Response: Changes were made.**

p. 11, line 2: “with species maximum” should be “with species’ maximum”

**Response: Changes were made.**

p. 11, line 5: “shapes” should be “shaped”, because it is discussing a student that has been published.

**Response: Changes were made.**

p. 11, line 9: “As soil aged” should be “As soils aged”

**Response: Changes were made.**

p. 11, line 11: "weathering stages had a major role" should be "weathering stage had a major influence"

**Response: Changes were made.**

p. 11, line 11: "dynamic" should be "dynamics"

**Response: Changes were made.**

p. 11, lines 16-23: It is well-established that ectomycorrhizal fungi convert the orthophosphate that they take up from the soil into polyphosphates, and translocate the polyphosphate along fungal hyphae, sometimes a great distance from where the orthophosphate is taken up (e.g. Bücking and Heyser 1999 Mycol. Res. 103:31-39; Plassard and Dell 2010 Tree Physiol. 30:1129-1139). This needs to be mentioned, as well as the need to define plant and microbial communities in studies of P forms.

**Response: The information about the ectomycorrhizal fungi was added and the need to define plant and microbial communities in studies of P forms was emphasized.**

p. 11, line 28 to p.12, line 2: Polyphosphates can potentially degrade to pyrophosphates during extraction and analysis by P-NMR (e.g. Cade-Menun et al. 2006, Environ. Sci. Technol. 40:7874-7880), so pyrophosphate and polyphosphate shouldn't be considered as fully distinct P forms, and the authors should use care when discussing hydrolysis of pyrophosphate separately from polyphosphate.

**Response: Changes were made.**

p. 12, lines 5-6: "Plant and microorganism breakdown diesters need" should be "Plant and microorganism breakdown of diesters needs"

**Response: Changes were made.**

p. 12, lines 8-9: "P limitation increased phosphoesterases synthesis as a way to increase the organic P breakdown to the bioavailable P". As written, it doesn't make sense. Do the authors want to say: "P limitation may stimulate increased phosphoesterase synthesis as a way to increase bioavailable P by the mineralization of organic P"?

**Response: The sentence was changed to the proposed one.**

p. 12, lines 12-16: “Therefore we hypothesize that as P got scarcer, plant and soil microorganisms may have been stimulated to produce phosphomonoesterases in greater amounts compared to phosphodiesterase because of the lower investment required for the organic P acquisition. Even though acid phosphatases require greater activation energy than alkaline phosphatases (Hui et al. 2013), breaking down diesters would require both enzymes; therefore a greater investment in energy”. These two sentences are baseless speculation, and should be deleted from the paper. The authors did not include any measures of phosphatase activities of any kind from any of the published studies, and thus are unqualified to say anything about how enzyme activity changes with soil factors. Their results in Fig. 3 show that the proportion of total organic P, as well as the proportions of IHP and DNA decrease with increasing pH. This is related to increased charge of these compounds with decreasing pH, and thus increased sorption. This is discussed in detail in Condron et al. 2005, among other papers.

**Response: That part of the discussion was rewritten and our hypothesis deleted.**

p. 12, lines 18-20: As noted for the introduction, it isn’t possible to take the results of one study into pasture soils from one country and extrapolate to a global model about enzyme activity, especially when the current study did not include any measures of enzyme activity. This needs to be deleted.

**Response: Sentence and assumption about those results were excluded.**

p. 12, lines 23-26: As noted above, the influence of pH on DNA sorption is well-established (Condron et al. 2005). The authors need to revise these lines of the discussion.

**Response: The discussion was strongly improved following the note above.**

p. 12, lines 28 to p. 13, line 12: The authors need to be very precise in their terminology. In this section, they discuss “inositol phosphates”. However, the data they include from various studies is for inositol hexaphosphates, which very specifically are inositols with 6 phosphate groups. These will behave very differently in soils from inositol phosphates with fewer phosphate groups. Please rewrite this section to be more precise. In addition, all of the processes discussed here to govern the behavior of “inositol phosphates” in soil will also apply to other P forms, especially DNA.

**Response: the terms "inositol phosphates" were changed to "inositol hexakisphosphates" throughout the manuscript. Moreover, the rationale was changed to include both inositol hexakisphosphates and DNA.**

p. 13, lines 9-10: This sentence discusses changes in the concentration of "inositol phosphate" with soil weathering. However, the authors only present changes in "inositol phosphates" as proportions of organic P, not as concentrations. As such, this is all speculation and should be revised or deleted.

**Response: The text was revised. We think that the rationale is important in the manuscript to justify why inositol hexakisphosphates concentrations could decrease in more weathered soils. In fact, all of our tropical results had negligible inositol hexakisphosphates concentrations. To complement the rationale we added a text that describes our results in concentrations from figure 7.**

p. 13, line 14: "there were no inositol phosphates on tropical, more weathered soils". Could this not also relate to either inputs from plants or production of phytases by plants and microbes? The authors did not include any factors that might influence the cycling of IHP in these types of soils. As such, this is merely speculation and should be deleted.

**Response: The text was rewritten including the plant role on the inositol hexakisphosphates acquisition and more references were added. The motive for not excluding the rationale is described in the comment above.**

p. 13, lines 20-23: "Recent investigations have contradicted the often-cited literature" please cite references here to support this statement of fact, both for the "often cited literature", and the "recent investigations".

**Response: the phrase was modified excluding the "often cited literature" part and references for "recent investigations" were added across the present and next paragraphs. We thought it would be redundant citing the recent investigations references at the present sentence and again along the following sentences.**

p. 14, lines 10-30: One thing missing from this section is the role of vegetation with respect to inputs of different P forms. It should also be noted that a greater soil wetness is not necessarily associated with leaching, if there is impeded drainage of the soils in some ways (e.g. from the formation of placic horizons). This section of the manuscript is really vague, and doesn't add anything to our knowledge of P

cycling in soils. The authors must do a far more thorough literature review to discuss their results properly.

**Response: We agree that vegetation may have played a role on the results, but we were unsuccessful in finding studies that clearly explained the role of vegetation, without confounding effects of soil or climate, on specific soil P compounds. These issues were also recently pointed out by another paper that dealt with the same dataset (Huang et al. 2017). Therefore, we added a section to the end of the discussion, along with other suggestions made by the other reviewer, to point out future research priorities. These included, among others, a suggestion for studies aiming to disentangle confounding effects among vegetation, soil biotic and abiotic components, and climate. The terms "Soil wetness" were excluded and that part was rewritten using "precipitation (in soils with no impeded drainage)".**

p. 14, line 32: "As the orthophosphate percentage decreased following precipitation, the pyrophosphate percentage increased". This does not necessarily mean that the concentration of orthophosphate decreased, or that of pyrophosphate increased, or imply a cause-and-effect relationship. This line suggests to me that the authors do not understand compositional analysis: as the percent of one thing decreases the percentage of another thing will increase because the total must add to 100%. In my opinion, reporting and discussing these as percentages is misleading, for this reason. It would be far better to use concentrations.

**Response: We completely agreed with the reviewer's point so we added the concentration results from Figure 7, which followed the same behavior, to imply in the rationale a cause-and-effect relationship.**

p. 15, lines 6, 7: "phosphorus" should be "P"

**Response: Changes were made.**

p. 15, lines 7-8: "greater organic P concentrations were associated with increasing biomass production".

**Response: Changes were made.**

Studies of biomass (e.g. Noack et al. 2012) show that the majority of P in plant biomass is as orthophosphate, not organic P compounds. As such, this sentence doesn't make sense to me, and isn't supported by the literature. Please revise.

**Response: We acknowledge that most of P in crop biomass (e.g. Noack et al. 2012) and possibly native vegetation biomass is as orthophosphate. To clarify the rationale we added a note "However, it is important to note that the majority of P in plant biomass (e.g. Noack et al. 2012) is as orthophosphate and not as organic compounds. We believe that even with higher orthophosphate inputs through plant biomass, organic P concentrations in soils would increase altogether with inorganic P concentrations, and also at expense of soil orthophosphate due to the greater bioavailability to plants and organisms of that inorganic form."**

8. Conclusions:
p. 15, lines 22-23: "after altogether with plant and microbe coexistence" as noted for the abstract, I do not understand what the authors are trying to say here. Something seems to be missing, or mistranslated.

**Response: The sentence was rewritten to "and then altogether through plant and microbe P cycling".**

9. Figures: In all figures, "phosphonate" should be "phosphonates", because this is a general compound category containing a number of specific P compounds. Also, "inositol" should be IHP or inositol hexakisphosphate.

**Response: All figures were changed to correct "phosphonate" to "phosphonates", and "inositol" to "IHP". We also corrected the line 21 of Figure 7 to a dotted line.**

10. References: there are many problems with this section of the manuscript. The authors need to carefully proof- read and correct this section.

. a) Many of the references are out of order alphabetically, including the entire "Y" section (which should come after W), Deiss et al. 2017 should come after Damon et al. 2014; Laliberté et al. 2017 should come before Legendre and Legendre 2102 and Li et al. 2015; Whittaker 1975 should come before Wilson et al. 2013.

**Response: All references were ordered. Y is coming after X, Y after W. Deiss after Damon. Laliberté was placed before Legendre and Legendre. Whittaker was placed before Wilson.**

. b) There are formatting differences among references. Cade-Menun et al. 2000 has the volume and page number listed as 30:1714-1725, while other references use a comma (e.g. Celi et al., 2013, 367, 121-134). Chen et al. 2004 does not include periods after abbreviations in the journal title (Aust J Soil Res); Cade- Menun and

Preston 1996, Kizewski et al. 2011, Turner et al. 2013 have all the words capitalized in the manuscript title.

**Response: volume and page number were standardized across all references. Periods after abbreviations in the journal title were added for Chen. Capitalized letters of title name of Cade-Menun and Preston 1996, Kizewski et al. 2011, Turner et al. 2013 were turned to lowercase, and other occurrences were followed by the same rule.**

. c) The journal volume number and/or page numbers are missing from Deiss et al. 2017, Doolette et al. 2016, Li et al. 2015, while George et al. 2017 is missing the name of the journal as well as the volume number.

**Response: Deiss, Doolette, and Li were completed with volume number and/or page numbers, and George was completed with Journal Name. We did not find volume number for the latter.**

. d) Rumpel et al. is 2015 in the References, but 2016 in the text; Bünemann is "Bunemann" in the text (p. 11, line 16)

**Response: Rumpel was corrected to 2015. Bünemann was corrected in the text.**

**REPORT 2**

Reviewer Comments

The manuscript presents some interesting results and the results are now quite clearly presented. While some aspects have been improved in the revision, other aspects remain problematic, including some issues already pointed out by earlier reviewers. The four main issues I see are:

1. Inadequate understanding / discussion of the limitations of the NMR method

2. Precision and clarity of terminology

3. A discussion section that does not go beyond what is already known

4. English language

Limitations of the NMR method.

Liquid 31P NMR is a powerful method to study the presence of certain P forms in soils, and it is of interest to combine results produced by different studies in a meta-analysis. However, it is absolutely necessary that the authors are aware and clearly and transparently communicate the limitations of liquid 31P NMR on NaOH-EDTA extracts. There are many other methods out there to study P cycling (Kruse et al. 2015), and only if results are correctly interpreted in light of the limitations of the methods can readers integrate knowledge gained from one method with knowledge gained from another method.

**Response: We added the rationale for the introduction to clarify the limitations of the 31P NMR following the reviewer suggestions.**

Firstly, as already pointed out by reviewer 1 (first round), NaOH-EDTA extraction never extracts all soil P and often only a small portion of total soil P. Please state the fraction of total P extracted by NaOH-EDTA for all soils used in your analysis.

**Response: The fraction of total P extracted by NaOH-EDTA is being reported on the Figure S5, and was used in all models as a random factor, but to complement those, we added the average and range of % extracted by NaOH-EDTA for both mineral and organic layers to the results section (end of second paragraph of results section).**

Secondly, 31P NMR is a tool for P speciation, providing important information on P stocks, it does not, however, provide information on dynamics. Turnover, exchange kinetics, mineralisation rates, etc, which is what I understand under “P dynamics”, can only be assessed using isotopic techniques (Frossard et al. 2011).

**Response: The "dynamics" term was excluded from the manuscript when referring to soil P composition results.**

Thirdly, liquid 31P NMR is not the preferred method for studying inorganic P species, since it cannot give information on predominant inorganic P species such as Fe-, Al- and Ca-phosphate. XANES is a more preferred method for looking at inorganic P species. This does not mean that the results on pyrophosphate, orthophosphate and polyphosphates concentrations are not useful, however, the paper reads as if these are the only inorganic P species of importance in soils. Please be more honest on the potentials and limitations of this approach.
These limitations need to be stated clearly and transparently in the introduction, since they provide the scope for the meta-analysis.

**Response: As described earlier, we added this rationale on the introduction to clarify the limitations of the 31P NMR and other methods for studying soil P. Moreover, we included this topic about the orthophosphate speciation as a main "Research Priority" in the last, new section of discussion.**

Precision and clarity of terminology. As mentioned above, the word "dynamics" is misleading for a study looking at P forms. I suggest changing the title to something more accurate, e.g. "The impact of soil, climatic, and temporal drivers on inorganic and organic P compounds".

**Response: The "dynamics" term was excluded from the manuscript, and a new title was introduced.**

An often-recurring term is "complex P compounds". From my understanding, "complex" is used to refer to high-molecular weight organic compounds of variable composition (McLaren et al. 2015). Please use a more precise term than "complex P compounds".

**Response: The terms "complex P compounds" throughout the manuscript. Complexity, otherwise, was maintained in some instances to explain modifications in soil P composition in later stages of pedogenesis.**

Discussion section
The discussion section remains the main weak spot of the article. As it stands, the discussion could be summarized by Fig 1 from the introduction. This is a problem because Fig 1 is a very generic and commonly used figure in the field. I suggest the authors follow Mensh and Kording (2017), who provide useful tips for structuring a discussion (Mensh and Kording 2017).

There follows a few ideas for rewriting the discussion that authors may use if they find them helpful. The first paragraph can more or less stay as is, it summarizes the results.

**Response: We thank the reviewer for the suggestions about the paper to better structure the discussion. We followed it to improve the discussion presentation and organization. We maintained those initial paragraphs of the discussion.**

The next two paragraphs could be dedicated to outlining the limitations of the analysis. Here it would
be worth pointing out that climate and weathering drive soil properties, so that it is not entirely appropriate to compare their influence on P forms, because much of the variation in soil properties can be explained by climate and weathering. Also, it might be worth discussing why variation in polyphosphates could not be explained by the models.

**Response: We added the point about interacting effects between climate and weathering on soils along with the first paragraph of discussion ("While the soil P composition was mainly directly influenced by soil properties, the impact of climate and weathering stage occurred mainly through indirect paths and their influence on soil properties"), and we also included this topic in the discussion "Research Priorities" section. We added a possible explanation to the poor response of the polyphosphates at the end of the third paragraph of the item "4.1 Soil properties and the soil P composition". The other reviewer presented an interesting point and we believe that it had a role in our responses. So we added this specific discussion at that part.**

The final two paragraphs could point out how this study adds to the literature. The focus of these paragraphs should not be comparisons to individual NMR studies, since they are anyways incorporated into the data, rather it would be interesting to weave the findings together with insights from studies using XANES, enzyme activities, or isotopes, to start painting a full picture.

**Response: We added a new section in the discussion to point out future opportunities for studies we struggled to find responses for while exploring our results.**

English language
Authors stated that a native English speaker revised the manuscript. However, I don't think that the quality of the writing has improved from the previous version. There are still many grammatical issues and awkward writing, which make the manuscript difficult to read at times.

**Response: A different native person with an affinity for the topic revised the English again throughout the manuscript.**

Mensh, B., and K. Kording. 2017. Ten simple rules for structuring papers. PLoS Comput Biol 13:e1005619.

**Environmental drivers of soil phosphorus composition in natural ecosystems**

Leonardo Deiss[1], Anibal de Moraes[1], Vincent Maire[2]

[revised manuscript text omitted]

L D 4/7/2018 4:05 PM
L D 4/9/2018 3:17 PM
L D 4/7/2018 4:05 PM
L D 5/21/2018 12:52 PM
L D 4/9/2018 3:18 PM
L D 5/21/2018 12:52 PM
L D 4/9/2018 3:19 PM
L D 4/7/2018 4:05 PM
L D 4/7/2018 4:05 PM
L D 4/7/2018 4:05 PM
L D 4/9/2018 3:20 PM
L D 5/21/2018 12:52 PM
L D 4/11/2018 3:01 PM
L D 4/9/2018 3:21 PM

pyrophosphate, the total variation explained by models ranged from 46% to 89%, and they were mostly explained by soil variables and combined effects of soil and weathering. Polyphosphates had a poorly-defined response to the variation partitioning ofecosystem properties (<0.01% of the total variation explained).

[revised manuscript text omitted]

L D 4/15/2018 10:11 AM

L D 4/13/2018 10:06 AM

L D 4/11/2018 5:22 PM
**Formatted** ... [111]

L D 4/11/2018 3:00 PM

L D 4/15/2018 10:32 AM
**Moved down [5]:** ; however, some auth ... [113]

Maire, Vincent 5/7/2018 11:45 PM
**Moved up [7]:** In our study, we found th ... [114]

Maire, Vincent 5/7/2018 11:42 PM

L D 4/15/2018 10:32 AM

L D 4/15/2018 10:32 AM
**Moved (insertion) [5]**

L D 4/15/2018 10:32 AM

Maire, Vincent 5/7/2018 11:47 PM
**Comment [8]:** The sentence is too long a ... [118]

L D 4/15/2018 10:31 AM
**Moved (insertion) [4]**

L D 4/22/2018 9:15 PM

L D 4/13/2018 10:12 AM
**Moved (insertion) [2]**

L D 4/22/2018 9:45 PM

L D 4/13/2018 10:06 AM
**Moved (insertion) [1]**

L D 4/13/2018 10:12 AM
**Moved up [2]:** Other P compounds such ... [121]

L D 4/13/2018 10:26 AM

of the P was organically bound (Werner et al. 2017). Yang et al. (2016) also showed that only limited portions of fine mineral surfaces contributed to soil organic matter stabilization. So, these facts could justify the minor association between bulk soil clay concentration and soil P compounds as observed in our study.

Soil organic matter stabilization could be facilitated in more weathered soils by the potential increase in amorphous Al and Fe oxides (Albrecht 1957; Walker 1965), and consequently more reactive surface area availability to absorb and stabilize soil organic matter; however, we did not observe a significant overall soil weathering effect on Al and Fe oxide concentrations (additional Appendix S9), suggesting that it may depend on other factors such as the parent material or specific soil orders. Moreover, contrasting soil organic matter responses to short-range-ordered (amorphous) Fe or Al oxides have been found in literature. Some investigations found a pronounced role promoted by Al oxides (Heister 2016; Kaiser et al. 2016), whereas others found Fe oxides as the main soil organic matter stabilizing mechanism (Wilson et al. 2013; Catoni et al. 2016; Deiss et al. 2017). Investigations also found no apparent relationship between soil organic matter and both Al and Fe oxides (Cloy et al. 2014; Vogel et al. 2015; Rumpel et al. 20155). Therefore, we could not confirm the role of Al and Fe oxides as influenced by soil weathering stages over the soil P composition.

**4.2 Climate and the soil P composition**

Climatic variables exerted an important role on the soil P composition but to a lesser extent when compared to soil variables (Figure 6). Contradicting what we expected, our results showed that temperatures ranging from -0.4 to 27 Cº had no effect on both $P_i$ and $P_o$ pools and their compounds (additional Appendices S7 and S8). It was expected that the soil $P_o$ concentration would decrease with increasing temperatures because higher temperatures are optimal for the breakdown of the soil $P_o$ compounds by the microbial biomass through phosphatase enzymes release (Turner et al. 2002). Another investigation confirmed that greater maximum phosphatase activity occurred at incubation temperatures >25°C when compared to 20°C, but no differences were observed among temperatures greater than 25°C (Hui et al. 2013). Therefore, phosphatase activity may depend on the range and magnitude of temperatures; and our results covered a greater range of markedly lower temperatures, which may reduce microbial activity variability even more due to a slowdown in the microorganisms' metabolism.

In contrast, precipitation affected several variables in the soil $P_i$ pool. This result was also expected based on the classic paper of Walker and Syers (1976), which suggested that pedogenesis depends predominantly on the volume of water leached through soil. In our results, precipitation was negatively related to the soil total P concentration and pH (Figure 7), and decreased orthophosphate proportion (additional Appendix S7C). As precipitation increased (additional Appendix 7C) and the soil was in a higher weathering stage (Figure 5C), the orthophosphate proportion (% of NaOH-EDTA P) possibly decreased because of increased leaching (Walker and Syers 1976; Feng et al. 2016). Feng et al. (2016) evaluated P transformations along a climosequence and observed that greater precipitation (in soils with no impeded drainage) reduced the inorganic concentration of P linked to Ca, corresponding to a marked decline in soil exchangeable Ca and suggesting an

L D 5/8/2018 11:48 AM

L D 4/9/2018 9:00 AM

L D 5/21/2018 12:52 PM

L D 4/26/2018 9:22 PM

L D 4/26/2018 1:44 PM

L D 4/26/2018 1:44 PM

L D 5/8/2018 11:48 AM

L D 4/7/2018 4:05 PM

L D 4/23/2018 8:53 AM

enhanced leaching of P along with weatherable cations. Moreover, greater soil water availability, and consequent greater primary productivity, may have increased the demand for P in its bioavailable form and contributed to the orthophosphate depletion.

As the orthophosphate percentage and concentration (additional Appendix 7C and Figure 7, respectively) decreased following greater precipitation, the pyrophosphate percentage and concentration (additional Appendix 7D and Figure 7, respectively) increased suggesting that this compound predominates underunder these environmental conditions. As previously described, this may be due to the incorporation of these compounds into recalcitrant soil organic matter (Turner et al. 2007). Moreover, given the microbial origin of pyrophosphate and its association with the microbial P biomass (Koukol et al. 2008; Turner and Engelbrecht 2011; Reitzel and Turner 2014), pyrophosphate (Figure 7, additional Appendix 7D) possibly mirrored the response of the total $P_o$ (additional Appendix 8B) to climatic variables, which may have resulted from greater soil organic matter accumulation following greater productivity (i.e., plants and organisms) ini these ecosystems with greater water availability. Evaluating the P budget of the whole ecosystem, Turner et al. (2013) demonstrated the dominance of microbial P in mature soils. Wang et al. (2014) found that greater organic P concentrations were associated with increasing biomass production (i.e., primary production and microbial biomass) because plants and microbes incorporate P into biomass and return it to the soil. However, it is important to note that the majority of P in plant biomass is as orthophosphate (e.g. Noack et al. 2012) and not as organic compounds. Even though, we believe that with higher orthophosphate inputs through plant biomass, soil organic P concentrations would increase altogether with orthophosphate P concentrations, and also at expense of soil orthophosphate due to the greater bioavailability to plants and organisms of that latter P compound.

Changes in vegetation are expected to occur during pedogenesis, and climatic variables may govern magnitudes of these alterations along with soil changes. Vitousek et al. (1995) showed that as ecosystems develop, the pattern of P concentration in plants leaves follows a non-linear response to time, in which lower concentrations occur at either early or late stages of pedogenesis, and a maximum is reached at an intermediate stage of pedogenesis. In addition, precipitation can affect the magnitude of that maximum response (intermediate stage), where the P concentration in plant leaves is higher in mesic gradients when compared to more wet gradients (Vitousek et al. 1995). Moreover, as described earlier, the soil available P, along with other climatic variables, governs maximum photosynthetic rates, but a trend that is expected to gradually decline in more weathered soils, due to a lower P availability (Maire et al. 2015). Phosphorus limitation can become sufficiently intense in the late stages of ecosystem development (also known as the retrogressive phase) to cause a decline in forest biomass, and productivity (Wardle et al. 2004). The exception seems to be tropical forests (Turner et al. 2007), which exhibit very diverse tree communities on old, infertile soils (Losos and Leigh 2004). Moreover, Turner et al. (2018) showed that in lowland tropical ecosystems, P limitation affects individual species, but species-specific P limitation does not translate into a community-wide response, because some species grow rapidly on infertile soils despite extremely low P availability.

L D 5/8/2018 11:48 AM

Maire, Vincent 5/7/2018 11:49 PM

Maire, Vincent 5/7/2018 11:49 PM

L D 4/22/2018 12:35 PM

L D 4/22/2018 12:26 PM

L D 4/26/2018 1:47 PM

Maire, Vincent 5/7/2018 11:50 PM

L D 5/21/2018 12:52 PM

L D 4/13/2018 11:13 AM

L D 4/13/2018 11:13 AM

Maire, Vincent 5/8/2018 11:48 AM
**Comment [9]:** Most chronoséquences show a positive relationship between age and plant biodiversity (see Laliberté last papers)

L D 5/21/2018 12:52 PM

L D 5/21/2018 12:52 PM

L D 5/21/2018 12:52 PM

Maire, Vincent 5/7/2018 11:50 PM

**4.3 Future research priorities**

[revised manuscript text omitted]

Maire, Vincent 5/7/2018 11:54 PM

L D 4/13/2018 11:58 AM

jordonwade@gmail.…, 5/21/2018 6:58 AM
**Comment [10]:** This sentence is very long, which makes it difficult to understand what you are trying to say. Consider revising for clarity.

Maire, Vincent 5/7/2018 11:56 PM
**Comment [11]:** We did not include it in path analysis. Should someone be frustrated that CP pops up here…?

jordonwade@gmail.…, 5/21/2018 7:00 AM

jordonwade@gmail.…, 5/21/2018 7:00 AM

jordonwade@gmail.…, 5/21/2018 7:00 AM

jordonwade@gmail.…, 5/21/2018 7:00 AM

L D 5/21/2018 12:52 PM

jordonwade@gmail.…, 5/21/2018 7:01 AM

L D 5/21/2018 12:52 PM

L D 4/25/2018 9:23 PM

L D 5/8/2018 9:18 PM

**6 APPENDICES (SUPPLEMENTARY FILES)**

[revised manuscript text omitted]

**Page 3: [30] Moved to page 3 (Move #6)** **Maire, Vincent** **5/6/18 7:43 AM**

Most importantly, we need to investigate the hierarchical nature of causal effects between state factors, soil weathering, soil properties and Po and Pi composition.

**Page 3: [31] Comment [2]** **jordonwade@gmail.com** **5/15/18 6:59 PM**

I’m not sure exactly what you mean by this. Another word may be more appropriate.

**Page 3: [32] Deleted** **jordonwade@gmail.com** **5/15/18 7:00 PM**

also determines differences

| Page 3: [32] Deleted | jordonwade@gmail.com | 5/15/18 7:00 PM |
|---|---|---|

also determines differences

| Page 3: [33] Deleted | Maire, Vincent | 5/6/18 7:43 AM |
|---|---|---|

between soils for a given soil weathering stage

| Page 3: [34] Deleted | jordonwade@gmail.com | 5/15/18 7:05 PM |
|---|---|---|

also

| Page 3: [34] Deleted | jordonwade@gmail.com | 5/15/18 7:05 PM |
|---|---|---|

also

| Page 3: [34] Deleted | jordonwade@gmail.com | 5/15/18 7:05 PM |
|---|---|---|

also

| Page 3: [34] Deleted | jordonwade@gmail.com | 5/15/18 7:05 PM |
|---|---|---|

also

| Page 3: [34] Deleted | jordonwade@gmail.com | 5/15/18 7:05 PM |
|---|---|---|

also

| Page 3: [35] Deleted | Maire, Vincent | 5/6/18 4:10 PM |
|---|---|---|

| Page 3: [35] Deleted | Maire, Vincent | 5/6/18 4:10 PM |
|---|---|---|

| Page 3: [36] Deleted | jordonwade@gmail.com | 5/15/18 7:07 PM |
|---|---|---|

, which define the soil P retention potential

| Page 3: [37] Deleted | jordonwade@gmail.com | 5/15/18 7:08 PM |
|---|---|---|

important

| Page 3: [37] Deleted | jordonwade@gmail.com | 5/15/18 7:08 PM |
|---|---|---|

important

| Page 3: [38] Deleted | jordonwade@gmail.com | 5/15/18 7:09 PM |
|---|---|---|

Soil P composition has been studied in ecosystems

| Page 3: [39] Deleted | L D | 4/9/18 9:16 AM |
|---|---|---|

Soil P composition has been studied in soils from ecosystems worldwide

| Page 3: [40] Deleted | jordonwade@gmail.com | 5/15/18 7:09 PM |
|---|---|---|

, and

| Page 3: [40] Deleted | jordonwade@gmail.com | 5/15/18 7:09 PM |
|---|---|---|

, and

| Page 3: [41] Deleted | L D | 5/21/18 12:52 PM |
|---|---|---|

nuclear

| Page 3: [41] Deleted | L D | 5/21/18 12:52 PM |
|---|---|---|

nuclear

| Page 3: [42] Deleted | L D | 4/26/18 12:55 PM |
|---|---|---|

| Page 3: [42] Deleted | L D | 4/26/18 12:55 PM |
|---|---|---|

| Page 3: [42] Deleted | L D | 4/26/18 12:55 PM |
|---|---|---|

| Page 3: [43] Deleted | jordonwade@gmail.com | 5/15/18 7:11 PM |
|---|---|---|

,

| Page 3: [43] Deleted | jordonwade@gmail.com | 5/15/18 7:11 PM |
|---|---|---|

,

| Page 3: [44] Deleted | jordonwade@gmail.com | 5/15/18 7:11 PM |
|---|---|---|

t

| Page 3: [44] Deleted | jordonwade@gmail.com | 5/15/18 7:11 PM |
|---|---|---|

t

| Page 3: [45] Deleted | L D | 5/21/18 12:52 PM |
|---|---|---|

more

| Page 3: [45] Deleted | L D | 5/21/18 12:52 PM |
|---|---|---|

more

| Page 3: [45] Deleted | L D | 5/21/18 12:52 PM |
|---|---|---|

more

| Page 3: [46] Comment [3] | jordonwade@gmail.com | 5/15/18 7:12 PM |
|---|---|---|

I’m not sure of the distinction you are trying to make here.

| Page 3: [47] Deleted | L D | 5/21/18 12:52 PM |
|---|---|---|

).

| Page 3: [47] Deleted | L D | 5/21/18 12:52 PM |
|---|---|---|

).

| Page 3: [47] Deleted | L D | 5/21/18 12:52 PM |
|---|---|---|

).

**Page 3: [48] Moved from page 3 (Move #8)jordonwade@gmail.com 5/15/18 7:14 PM**

SomeHowever, there are drawbacks of using NaOH-EDTA extractant for $^{31}$P NMR analysis. exists such as that NaOH-EDTA usuallyNaOH-EDTA does not extract all soil P and the highly alkaline environment, the extractant can potentially degrade some P compounds compounds due to the highly alkaline environment (Cade-Menun et al. 2006; Cade-Menun and Liu 2014). Additionally, the NaOH-EDTA extraction, and it does not separate Fe-, Al- and Ca-phosphate compounds (Kizewski et al. 2011).

**Page 3: [49] Deleted jordonwade@gmail.com 5/15/18 7:14 PM**

Some

**Page 3: [49] Deleted jordonwade@gmail.com 5/15/18 7:14 PM**

Some

**Page 3: [50] Moved to page 3 (Move #8) jordonwade@gmail.com 5/15/18 7:14 PM**

Some drawbacks of $^{31}$P NMR exists such as that NaOH-EDTA usually does not extract all soil P, the extractant can potentially degrade some P compounds due to the highly alkaline environment (Cade-Menun et al. 2006; Cade-Menun and Liu 2014), and it does not separate Fe-, Al- and Ca-phosphate compounds (Kizewski et al. 2011).

**Page 4: [51] Comment [4] Maire, Vincent 5/6/18 7:49 AM**

I have removed this line as most of our database is composed from chronosequence. Then, it is a bit dangerous to say we will look at the effect other than the ones linked to chronosequence

**Page 4: [52] Deleted Maire, Vincent 5/6/18 7:49 AM**

because most know responses were obtained from specific chronosequences

**Page 4: [53] Deleted L D 4/14/18 12:53 PM**

different state factors of ecosystem functioning because known responses were obtained from case-specific conditions.

**Page 4: [54] Formatted L D 4/9/18 9:26 AM**

Highlight

**Page 4: [55] Deleted jordonwade@gmail.com 5/15/18 7:19 PM**

and

**Page 4: [55] Deleted jordonwade@gmail.com 5/15/18 7:19 PM**

and

**Page 4: [56] Deleted Maire, Vincent 5/6/18 9:24 PM**

could help better understand how soil P dynamicscomposition is influenced by environmental properties. Therefore, w

**Page 4: [57] Deleted jordonwade@gmail.com 5/15/18 7:20 PM**

enable

| Page 4: [57] Deleted | jordonwade@gmail.com | 5/15/18 7:20 PM |
|---|---|---|

enable

| Page 4: [58] Deleted | jordonwade@gmail.com | 5/15/18 7:20 PM |
|---|---|---|

determine

| Page 4: [58] Deleted | jordonwade@gmail.com | 5/15/18 7:20 PM |
|---|---|---|

determine

| Page 4: [59] Deleted | jordonwade@gmail.com | 5/15/18 7:21 PM |
|---|---|---|

the

| Page 4: [59] Deleted | jordonwade@gmail.com | 5/15/18 7:21 PM |
|---|---|---|

the

[revised manuscript text omitted]

**Appendix S1 - Dataset**

The dataset of “Soil phosphorus dynamics on terrestrial natural ecosystems” is attached as a supplementary Excel file.

**Appendix S2 – Global biomes according to the Whittaker' diagram**

[Figure]

**Figure S2**. The Whittaker' diagram used to determine the main biomes comprised in our dataset.

**Appendix S3 – Soil depth effect on soil P composition**

[Figure]

**Figure S3**. Soil inorganic and organic phosphorus (P) composition in NaOH EDTA extract as influenced by soil sampling depth on terrestrial natural ecosystems. Significant relationships are indicated with regression lines.

L D 4/26/2018 7:36 PM

[Figure]

Unknown

**Appendix S4 – Latitude effect on soil P composition**

[Figure]

**Figure S4**. Soil inorganic and organic phosphorus (P) composition in NaOH EDTA extract as influenced by latitude on terrestrial natural ecosystems. Significant relationships are indicated with regression lines.

L D 4/26/2018 7:38 PM

[Figure]

Unknown

**Appendix S5 – Percentage of P extracted with NaOH EDTA effect on soil P composition**

[Figure]

**Figure S5**. Soil inorganic and organic phosphorus (P) composition in NaOH EDTA extract as influenced by soil sampling depth on terrestrial natural ecosystems. Significant relationships are indicated with regression lines.

L D 4/26/2018 7:40 PM

[Figure]

Unknown

**Appendix S6 – Soil properties and soil organic phosphonates**

[Figure]

**Figure S6**. Relationship between edaphic properties and soil organic phosphonates in NaOH EDTA extract from soil mineral and organic layers on terrestrial natural ecosystems. No significant relationships were found.

L D 4/26/2018 7:41 PM

[Figure]

Unknown

L D 4/26/2018 7:42 PM

**Appendix S7 – Climatic properties and soil inorganic phosphorus**

[Figure]

**Figure S7**. Relationship between climatic properties and soil inorganic phosphorus (P) composition in NaOH EDTA extract from soil mineral and organic layers on terrestrial natural ecosystems. Regression models (n = 80 mineral layer and n = 20 mineral layer): mineral layer, total $P_i$ (%) = 89.4 – 14.7 log(precipitation), $r^2$ = 0.08; mineral layer, orthophosphate = 127 – 11.7 log(precipitation), $r^2$ = 0.24; mineral layer, pyrophosphate = -28.1 + 11.9 log(precipitation), $r^2$ = 0.24.

**Appendix S8 – Climatic properties and soil organic phosphorus**

[Figure]

**Figure S8**. Relationship between climatic properties and soil organic phosphorus (P) composition in NaOH EDTA extract from soil mineral and organic layers on terrestrial natural ecosystems. Regression models: mineral layer, total $P_o$ (%) = 10.6 + 14.7 log(precipitation), $r^2$ = 0.08 (n=80).

L D 4/26/2018 7:43 PM

[Figure]

Unknown

**Appendix S9 – Soil weathering stages and poorly crystalline Al and Fe concentration**

[Figure]

**Figure S9**. Soil weathering stage relationship with soil poorly crystalline Al and Fe (n = 49) on terrestrial natural ecosystems. For all three panels p>0.1.

**Appendix S10 – Models tested to explore the interdependences between edaphic and climatic variables (path analysis) as the main environmental predictors of soil inorganic and organic P compounds.**

We expected some directionalities in the relationships, based on the literature (theoretical models on Figures S.10.1 and S10.2, top panels).

[Figure]

[Figure]

**Figure S10.1.** Theoretical models set to explore the interdependences between edaphic and climatic variables (path analysis) as the main environmental predictors of soil inorganic (upper panel) and organic (bottom panel) P compounds.

L D 4/26/2018 7:47 PM

[Figure]

Unknown

---

## Author Response (AR4)

**Biogeosciences**
**Manuscript bg-2017-307**

**Dear Editor Sönke Zaehle,**

**Thank you for your work on our manuscript. We are humbled with all the suggestions made by the reviewer. We would like to thank him/her again for this immense effort on reviewing our manuscript. Those insights and precisions strengthened in the most positive way our manuscript.**

**The changes made on the manuscript are briefly described in the following. We corrected all English problems noted by the reviewer. We cited the original papers instead of some more recent reviews. We added additional information, and adjusted the non-accurate statements following his/her directions. The citations regarding fractionation methods (e.g. Hedley) were reviewed, and exclusions were made when they were not appropriate (when referring to soil P composition). We changed nomenclature for "fractionation" or "fractions" when those citations were required on the text. We corrected the missing abbreviations for phosphorus, inorganic phosphorus, and organic phosphorus (P, $P_i$ and $P_o$), except at the beginning of sentences, and when we described the search strategy on methods section. New references were added, and some were excluded following the reviewer. All the references were reviewed for formatting, and several details adjusted.**

**Responses to the reviewers' comments are described below and are indicated by "Response". Changes made on the manuscript were all based on these responses. To facilitate the evaluation, we left the reviewers comments before each response.**

**Responses made by the authors are in bold and indicated with the term "Response". After the responses to the reviewers comments one can find the manuscript with all changes marked.**

**Responses to the comments made by the Anonymous Referee #1**

1. Writing quality: for the most part, the quality of English has improved. However, there are still some problems in some places, which are noted below.

   **Response: We followed and corrected all instances pointed out by the reviewer.**

2. I am puzzled about some of the literature cited in the introduction and discussion, and still think there is a bias to citations of specific authors rather than a good overview of the literature.

   a) Please cite original papers where possible, rather than recent review papers. For example, why cite Nash et al. (2014), rather than Newman and Tate 1980, Commun. Soil Sci. Plant Anal 11:835-842) or Tate and Newman (1982, Soil Biol. Biochem 14:191-196) for the description of P forms on p. 2, line 17? Why cite Turner 2008a for phosphatase and orthophosphate uptake (p. 2, line 20) – this was well-known prior to that paper. Why cite Huang et al. 2017 for vegetation and organisms (p. 10, l. 24) or mycorrhizae (p. 13, l. 16), when it presents no new information on either of these topics? The same is also true for Yu et al. 2013. There has been a lot of good research into many aspects of P cycling by a lot of good scientists; citing the review papers instead of the original research is unfair to the original researchers, and runs the risk of introducing errors into the literature if the original work was cited incorrectly in the review. Would the authors of this manuscript not prefer to have it cited directly?

   **Response: We completely agree with the reviewer. The two Newman and Tate's original papers were cited instead of Nash et al. (2014). Jackman and Black (1952) was used instead of Turner (2008a), which is one of the first studies to use enzyme additions to evaluate the availability of inositol hexaphosphate. Huang et al. 2017 was excluded from the second occasion, but it was maintained in the first one because we were not citing it for vegetation and organisms, but for the same struggle he had to drawn conclusions from the influence of vegetation and organisms on the soil P composition (as determined by $^{31}$P NMR). Yu et al (2013) was excluded from the manuscript.**

   b) Please be careful citing studies using fractionation methods (e.g. Hedley) in study discussing specific chemical forms. Fractions from fractionation methods are operationally defined, not chemically defined, especially the Hedley fractionation method.

   **Response: The citations regarding fractionation methods were reviewed. Exclusions were made when they were not appropriate**

**(when referring to soil P composition), and we changed nomenclature for "fractionation" or "fractions" when they were required on the text.**

3. Abstract:

   Lines 11-12: "nuclear magnetic resonance of NaOH-EDTA extracts" should be "nuclear magnetic resonance spectroscopy of NaOH-EDTA extracts" (something I missed in my previous review).

   **Response: Alteration made.**

   Lines 16-17: "organic and inorganic P compounds variations" should be "variations in organic and inorganic P compounds"

   **Response: Alteration made.**

   Lines 19-20: "and after altogether with plant and microbe" I still do not understand what the authors are trying to say here; something seems to be missing. Do they mean "and together with plant and microbe"?

   **Response: or clarification, "and after altogether with" was cut off and we maintained "plant and microbe" only.**

4. Introduction:
   p. 2, line 6 and elsewhere: "phosphorus" should be "P" after the first use of the abbreviation, except at the start of a sentence.

   **Response: All occurrences of "phosphorus" were changed to "P", except at the beginning of sentences, and when we described the search strategy on methods section.**

   p. 2, line 14: "composed by specific" should be "composed of specific"

   **Response: Alteration made.**

   p 2, lines 17 and 18: please replace "Nash et al. 2014" and "Cade-Menun and Preston 1996" with more appropriate references, because these compound classes were recognized prior to the publication of these papers

   **Response: The two Newman and Tate's original papers were cited instead of Nash et al. (2014).**

   p. 2, line 20: "As most enzymes, the activity of soil P cycling enzymes" should be "As with most enzymes, the activities of soil phosphatases"

   **Response: Alteration made.**

p. 2, line 21: "specific enzyme optimum" should be "specific enzyme optima"

**Response: Alteration made.**

p. 2, lines 24-25: DNA adsorption is only below 5 (the isoelectric point of DNA); also, why is a review paper (Yu et al. 2013) cited here, rather than one of the original references about pH? Please see the comments for page 12, below.

**Response: "in pH lower than 5 (the isoelectric point of DNA)" was added to the sentence. Cai et al. 2006 was used instead of Yu et al. 2013.**

p. 2, lines 27-31: studies using P fractionation (e.g. Walker and Syers 1976, Yang and Post 2011) do not give any information about Po and Pi forms and compounds, because fractionation by definition can only give information about operationally-defined pools. Please rewrite this paragraph, and cite better references that actually describe changes in Pi and Po compounds

**Response: The new references were added to support the "forms and compounds" terms. The sentence was written again adding "for P fractions" when referring to fractionation methods.**

p. 3, line 6: "While this study" should be "While the Feng et al. (2016) study"

**Response: Alteration made.**

p. 3, line 11: "Variation in" should be "Variations in"

**Response: Alteration made.**

p. 3, lines 27-28: "the NaOH-EDTA extraction does not separate Fe-, Al-and Ca-phosphate compounds (Kizewski et al. 2011)" although this might be what Kizewski et al. said, it isn't very accurate. A better way to say it is: "the high pH of the NaOH-EDTA extraction separates P species from the cations (e.g. Al, Fe, Ca) with which they were associated in soil".

**Response: Text was changed to the suggested one.**

p. 3, lines 28-30: Yes, there are other methods to study P dynamics and soil P composition, but none of these methods is perfect individually. For example, while XANES is a solid-state technique that does not require extraction, P concentrations are often below the detection limit, so it can only detect broad P species groups (e.g. Fe-P, Ca-P), but can't for example say if DNA is sorbed to Fe or Al. The most thorough studies of soil P use a combination of techniques together, and not any single technique (e.g. Liu et al. 2013 J.

Environ. Qual 42:1763-1770; Liu et al. 2015 Environ Sci Technol 49:168-176).

**Response: Our previous sentence was modified to include this explanation and references.**

p. 3, lines 30-32: "This does not mean that the results on pyrophosphate, polyphosphate and total orthophosphate concentrations are not useful, however, there are other inorganic P compounds of importance in soils". This sentence makes no sense to me. No, there are no other inorganic P compound in soil other than orthophosphate, pyrophosphate and polyphosphate. However, as noted above, extraction with NaOH-EDTA removes inorganic (and organic) P compound from the cations with which they are associated in soils. Thus, as noted below, a combination of techniques will give the most complete picture of soil P speciation and dynamics. Please rewrite this sentence.

**Response:  As we added the previous explanation we excluded this un-precise sentence.**

5. Methods: These are now clear and well-written.

6. Results:

   p. 7, line 1: "there was no pH effect on both pools" should be "there was no pH effect on either pool"

   **Response: Alteration made.**

   p. 7, line 6: "additional Appendix S3" delete "additional"

   **Response: Alteration made. All "additional" were deleted from the manuscript when referring to appendices.**

   p. 7, line 8: "additional Appendix S6" delete "additional"

   **Response: Alteration made.**

   p. 9, lines 5-6: "organic and inorganic P" should be "Po and Pi" to be consistent with the rest of the manuscript

   **Response: All occasions were changed to the abbreviation, except at the beginning of sentences.**

7. Discussion: In general, this part of the manuscript is improved compared to previous versions.

p. 10, line 5: "P organic and inorganic compounds" should be "Po and Pi compounds"

**Response: Alteration made.**

p. 10, line 8: "the decaying degree of C element is lower than the P" I do not understand what you are trying to say here. Is it "organic matter degrades more slowly than Po compounds"? Please rewrite.

**Response: decaying degree means decomposition and this term was added in parenthesis after the mentioned terms for clarity.**

p. 10, line 9: "Turner and Condron 2013" is an opening paper introducing a special issue, and does not contain data to support this statement of fact. Please cite another reference that actually contains data.

**Response: Turner and Condron 2013 was excluded, Walker 1965 was maintained.**

p. 10, lines 16-17: It is not possible to determine specific P compounds such as apatite with Hedley fractionation, and the long extraction times likely also degrade organic P. Please cite a better reference, with actual soil chemical data to support this point, and not a review article of Hedley fractionation (Yang and Post 2011).

**Response: We changed the "P compounds" to "P fractions" to fulfill the sentence meaning. Yang and Post 2011 is actually a data compilation study, and therefore was maintained since it goes beyond any other individual study.**

p. 10, lines 23-25: please cite an original study that contains data, and not a review paper (Huang et al. 2017) to  support this statement of fact

**Response: the phrase "quantitative data on the feedback between P compounds and biological communities during pedogenesis is still incipient to conclusions drawn from the influence of vegetation and organisms on the soil P composition (Huang et al. 2017)" is from the cited study, and it was originated because those authors had the same struggle as we had when evaluating other NMR studies, therefore is was maintained. There are several other instances in the manuscript where those specific responses were referenced from original NMR studies.**

p. 11, line 6: "proportion" should be "proportions".

**Response: Alteration made.**

p. 11, lines 26-28: These are also conditions under which ectomycorrhizal fungi are found. These fungi produce  hyphal mats in the forest floor, so an increase in polyphosphates  could  reflect  an  increase  in  ectomycorrhizal  hyphal mats.

**Response: This explanation was added to the paragraph in the suggested place.**

p. 12, line 5: "pH affect" should be "pH affects"

**Response:  Alteration made.**

p. 12, line 5: "organic and inorganic P compounds" should be "Po and Pi compounds"

**Response:  Alteration made.**

p. 12, line 14: "on the clay minerals", delete "the"

**Response: "the"  was deleted.**

p. 12, lines 15-18: I am pleased to see the authors citing original studies about DNA sorption at the start of this  section. However, I do not understand why they cited Yu et al. 2013 at the end of the section (line 18), because this is a review paper. Please replace this citation with a paper containing original research to support this statement of fact.

**Response:  Cai et al. 2006, the original study, was cited instead of Yu et al. 2013.**

p. 12, line 22: "for the P acquisition" delete "the"

**Response: "the"  was deleted.**

p. 12, line 25: "organic P" should be "Po"

**Response:  Alteration made.**

p. 12, line 28: "George et al. 2017" is a broad review paper; please cite a more specific reference to support this  statement of fact.

**Response:  "George  et al. 2017" was excluded and "Zimmerman et al. 2013" was maintained.**

p. 12, line 28: "Plant and microorganism" should be "Plants and microorganisms"; "diesters" should be "orthophosphate diesters" or "phosphodiesters".

**Response: Alterations were made through out the manuscript.**

p. 12, line 29: "monoesters" should be "orthophosphate monoesters", or "phosphomonoesters"; monoesters and diesters are general bond descriptions; the "orthophosphate" or "phospho" is more specific

**Response: "orthophosphate" was added in all instances throughout the manuscript.**

p. 12, line 30: Why is Turner 2008a cited to support this statement of fact about phosphatases? This was known years before the Turner paper was published (e.g. Halsted 1964, Skujins 1967, Tabatabai and Bremner 1969).

**Response: All three references were cited to support the phosphatases statement.**

p. 12, line 33: "both solubilization and hydrolysis by the phytase" As written, this implies that phytase both solubilizes and hydrolyzes inositol hexaphosphates, which is incorrect. Organic acids are required to desorb inositol phosphates, so that phytases can hydrolyze them. Please rewrite these lines, with a more appropriate reference than Turner 2008a.

**Response: Turner 2008a reference was excluded for that state of fact.**

p. 13, line 7: "effect in" should be "effect on"

**Response: Alterations were made.**

p. 13, line 8: "did not had" should be "had no"

**Response: Alterations made.**

p. 13, line 16: Huang et al. 2017 provides no direct evidence of phytate mineralization by ectomycorrhizal fungi. Please cite a better reference to support this statement of fact.

**Response: Huang et al. 2017 was excluded Chen et al. 2004 was maintained.**

p. 14, lines 12-13: Turner et al. 2002 is a review paper about inositol phosphates, so why is it being cited to support a general statement about

temperature and phosphatase activity? Please replace this with a more appropriate reference.

**Response: Hui et al. 2013 was used instead of Turner et al. 2002.**

p. 14, lines 24-29: Why are Walker and Syers 1976 and Feng et al. 2016 being cited to support discussion about decreased orthophosphate measured in NaOH-EDTA extracts by NMR? Neither of the cited references used NaOH-EDTA extraction or NMR, so they are irrelevant to the discussion here. Please replace these with better references. The authors should also mention here that caution needs to be used when discussing changes in orthophosphate extracted by NaOH-EDTA for NMR, because it will preferentially extract organic P rather than orthophosphate. As such, studies of the residuals after NaOH-EDTA show that it is mainly orthophosphate, especially for soils with low total P recovery by NaOH-EDTA.

**Response: Walker and Syers 1976 and Feng et al. 2016 references were excluded from that specific sentence. Even though they measured orthophosphate with different methods, they refer to the same orthophosphate, and therefore one should expect that the mentioned responses of increased phosphorus losses due to increased precipitation won't change. Therefore, we left the other citations in the other parts of the paragraph. The caution sentence was added to the paragraph.**

p. 15, line 9, "Even though" should be "However"

**Response: Alteration made.**

p. 15, lines 13 and 17: "Vitousek et al. 1995" is cited in the text but is not listed in the References

**Response: Vitousek et al. 1995 was added to the references.**

p. 15, lines 23-25: "phosphorus" should be abbreviated to be consistent with the rest of the manuscript.

**Response: Alteration made.**

p. 15, line 29: "why inositol hexakisphosphates have not been found" should be "why are inositol hexakisphosphates not found"

**Response: Alteration made.**

p. 16, line 15: "functional groups only (i.e. diesters and monoesters)" should be "broad compound classes only (i.e. orthophosphate diesters and monoesters"

**Response: Alteration made.**

p. 16, line 15: "when compounds concentrations" should be "when compound concentrations"

**Response: Alteration made.**

8. Conclusions:

p. 15, line 27: "as pedogenesis evolve" should be "as pedogenesis evolves"

**Response: Alteration made.**

9. Acknowledgements: "NSERC-Discover" should be "NSERC Discovery"; "from which studies" should be "from whose studies"

**Response: Alterations made.**

10. References: in general, this section is greatly improved compared to the previous version of this manuscript. However, there are still differences in formatting (e.g. Cloy et al. 2014) and some spelling mistakes (e.g. "Biophsyical" in Wang et al. 2014". Please proofread this section carefully.

**Response: Cloy et al. 2014 was formated. "Biophsyical" was replaced by "Biophysical" in Wang et al. 2014. Several other minor details were formatted.**

**Environmental drivers of soil phosphorus composition in natural ecosystems**

Leonardo Deiss[1], Anibal de Moraes[1], Vincent Maire[2]

[revised manuscript text omitted]

L D 7/14/2018 3:51 PM

L D 7/14/2018 8:03 PM

L D 7/14/2018 8:05 PM

L D 7/14/2018 8:21 PM

L D 7/14/2018 8:21 PM

L D 7/14/2018 8:21 PM

L D 7/14/2018 3:52 PM

L D 7/14/2018 8:14 PM

L D 7/14/2018 8:36 PM

L D 7/15/2018 1:22 PM

L D 7/14/2018 3:52 PM

L D 7/14/2018 3:52 PM

Engelbrecht 2011; Reitzel and Turner 2014), pyrophosphate (Figure 7, Appendix 7D) possibly mirrored the response of the total $P_o$ (Appendix 8B) to climatic variables, which may have resulted from greater soil organic matter accumulation following greater productivity (i.e., plants and organisms) in these ecosystems with greater water availability. Evaluating the P budget of the whole ecosystem, Turner et al. (2013) demonstrated the dominance of microbial P in mature soils. Wang et

5    al. (2014) found that greater $P_o$ concentrations were associated with increasing biomass production (i.e., primary production and microbial biomass) because plants and microbes incorporate P into biomass and return it to the soil. However, it is important to note that the majority of P in plant biomass is as orthophosphate (e.g. Noack et al. 2012) and not as $P_o$ compounds. However, we believe that with higher orthophosphate inputs through plant biomass, soil $P_o$ concentrations would increase altogether with orthophosphate P concentrations, and also at expense of soil orthophosphate due to the

10   greater bioavailability to plants and organisms of that latter P compound.

Changes in vegetation are expected to occur during pedogenesis, and climatic variables may govern magnitudes of these alterations along with soil changes. Vitousek et al. (1995) showed that as ecosystems develop, the pattern of P concentration in plants leaves follows a non-linear response to time, in which lower concentrations occur at either early or late stages of pedogenesis, and a maximum is reached at an intermediate stage of pedogenesis. In addition, precipitation can

15   affect the magnitude of that maximum response (intermediate stage), where the P concentration in plant leaves is higher in mesic gradients when compared to more wet gradients (Vitousek et al. 1995). Moreover, as described earlier, the soil available P, along with other climatic variables, governs maximum photosynthetic rates, but a trend that is expected to gradually decline in more weathered soils, due to a lower P availability (Maire et al. 2015). Phosphorus limitation can become sufficiently intense in the late stages of ecosystem development (also known as the retrogressive phase) to cause a

20   decline in forest biomass, and productivity (Wardle et al. 2004). The exception seems to be tropical forests (Turner et al. 2007), which exhibit very diverse tree communities on old, infertile soils (Losos and Leigh 2004). Moreover, Turner et al. (2018) showed that in lowland tropical ecosystems, P limitation affects individual species, but species-specific P limitation does not translate into a community-wide response, because some species grow rapidly on infertile soils despite extremely low P availability.

25   **4.3 Future research priorities**

Many efforts have been made to explain soil P composition during pedogenesis; however, a clear picture on how specific plant species, plant functional traits, and their communities can influence the soil P composition is still lacking, especially with results obtained with [31]P NMR. For example, why are inositol hexakisphosphates not found in tropical soils under native vegetation, i.e., is it because the rapid turnover promoted by plants and, or organisms (which one?), or exclusively due

30   to lack of inputs from plants? Does the changes in forest biomass and plant species diversity as soil P turns scarcer contribute to soil P composition in non-tropical environments, either by inputs or P compounds consumption, or the soil *per se* governs both the soil P composition and vegetation dynamics? Therefore, we point out that studies aiming to disentangle

L D 7/14/2018 3:52 PM

L D 7/14/2018 3:52 PM

L D 7/14/2018 5:09 PM

L D 7/14/2018 5:10 PM

L D 7/14/2018 8:57 PM

L D 7/14/2018 5:10 PM

L D 7/14/2018 2:29 PM

L D 7/14/2018 2:29 PM

L D 7/14/2018 2:30 PM

L D 7/14/2018 9:06 PM

[revised manuscript text omitted]

.

| Page 21: [2] Deleted | L D | 7/14/18 10:14 PM |
|---|---|---|

.

| Page 21: [3] Formatted | L D | 7/15/18 12:17 PM |
|---|---|---|

Not Highlight

| Page 21: [3] Formatted | L D | 7/15/18 12:17 PM |
|---|---|---|

Not Highlight

| Page 21: [3] Formatted | L D | 7/15/18 12:17 PM |
|---|---|---|

Not Highlight

| Page 21: [3] Formatted | L D | 7/15/18 12:17 PM |
|---|---|---|

Not Highlight

| Page 21: [3] Formatted | L D | 7/15/18 12:17 PM |
|---|---|---|

Not Highlight

| Page 21: [4] Deleted | L D | 7/14/18 10:15 PM |
|---|---|---|

, 1992.

| Page 21: [4] Deleted | L D | 7/14/18 10:15 PM |
|---|---|---|

, 1992.

| Page 21: [5] Deleted | L D | 7/14/18 10:17 PM |
|---|---|---|

R.

| Page 21: [5] Deleted | L D | 7/14/18 10:17 PM |
|---|---|---|

R.

| Page 21: [5] Deleted | L D | 7/14/18 10:17 PM |
|---|---|---|

R.

| Page 21: [5] Deleted | L D | 7/14/18 10:17 PM |
|---|---|---|

R.

| Page 21: [5] Deleted | L D | 7/14/18 10:17 PM |
|---|---|---|

R.

**Page 21: [5] Deleted** L D 7/14/18 10:17 PM

R.

**Page 21: [5] Deleted** L D 7/14/18 10:17 PM

R.

**Page 21: [5] Deleted** L D 7/14/18 10:17 PM

R.

**Page 21: [5] Deleted** L D 7/14/18 10:17 PM

R.

**Page 21: [5] Deleted** L D 7/14/18 10:17 PM

R.

**Page 21: [5] Deleted** L D 7/14/18 10:17 PM

R.

**Page 21: [5] Deleted** L D 7/14/18 10:17 PM

R.

**Page 21: [5] Deleted** L D 7/14/18 10:17 PM

R.

**Page 21: [5] Deleted** L D 7/14/18 10:17 PM

R.

**Page 21: [6] Deleted** L D 7/14/18 10:19 PM

2014

**Page 21: [6] Deleted** L D 7/14/18 10:19 PM

2014